# FedSUM Family: Efficient Federated Learning Methods under Arbitrary Client Participation

## Abstract

Federated Learning (FL) methods are often designed for specific client participation patterns, limiting their applicability in practical deployments. We introduce the FedSUM family of algorithms, which supports arbitrary client participation without additional assumptions on data heterogeneity. Our framework models participation variability with two delay metrics, the maximum delay $\tau_{\max}$ and the average delay $\tau_{\text{avg}}$. The FedSUM family comprises three variants: FedSUM-B (basic version), FedSUM (standard version), and FedSUM-CR (communication-reduced version). We provide unified convergence guarantees demonstrating the effectiveness of our approach across diverse participation patterns, thereby broadening the applicability of FL in real-world scenarios.

## 1 Introduction

Federated learning (FL) is a powerful paradigm for large-scale machine learning, especially when data and computational resources are distributed across diverse clients, such as phones, sensors, banks, and hospitals (McMahan et al., 2017; Yang et al., 2020; Kairouz et al., 2021). FL has been widely adopted in commercial applications, including autonomous vehicles (Chen et al., 2021; Zeng et al., 2022) and natural language processing (Yang et al., 2018; Ramaswamy et al., 2019). FL enhances computational efficiency by enabling parallel local training across distributed clients. It also preserves data privacy, as raw data remains on the device and is never directly transmitted to the central server.

One challenge in FL is variable **client participation**. In practice, not all clients can participate in every training round due to factors such as connectivity issues and resource constraints (Li et al., 2020). The variability has motivated the development of various models and assumptions regarding participation patterns (Karimireddy et al., 2020; Gu et al., 2021; Huang et al., 2023; Wang & Ji, 2023; Cho et al., 2023; Xiang et al., 2024), which may be either under the server's control or beyond it, and either homogeneous or heterogeneous across clients and rounds. Since different participation patterns can significantly affect convergence, quantifying and addressing their impact is essential for effective learning in practical deployments.

Another major challenge affecting FL effectiveness is **data heterogeneity**, where client data distributions are non-identical or highly personalized in operational environments (Zhao et al., 2018; Kairouz et al., 2021; Li et al., 2022). Such heterogeneity can lead to divergence between local and global models, particularly when multiple local updates are performed before aggregation (Mohri et al., 2019; Li et al., 2019). The effectiveness of classical approaches such as FedAvg (McMahan et al., 2017; Stich, 2018) has been shown to be limited in the presence of heterogeneous data and partial client participation, motivating a number of subsequent improvements (Karimireddy et al., 2020; Yang et al., 2021).

In light of these challenges, developing FL algorithms that can simultaneously mitigate data heterogeneity, support efficient local updates, and remain robust to arbitrary client participation remains a fundamental open problem in federated learning.

## 1.1 MAIN RESULTS AND CONTRIBUTIONS

In this paper, we make the following key contributions to Federated Learning:

- **Arbitrary Client Participation:** We study FL with general nonconvex objectives under arbitrary client participation, covering a wide spectrum of participation patterns, including controllable or uncontrollable, stochastic or deterministic, and homogeneous or heterogeneous. To characterize variability in participation, we consider two delay metrics, $\tau_{\max}$ (maximum delay) and $\tau_{\text{avg}}$ (average delay), which allow us to precisely quantify its impact on convergence. To the best of our knowledge, this is the first work to analyze such diverse client participation scenarios.

- **FedSUM Family of Algorithms:** We propose the FedSUM family of algorithms, including **FedSUM-B** (basic version), **FedSUM** (standard version), and **FedSUM-CR** (communication-reduced version), all designed for arbitrary client participation. These algorithms employ the **Stochastic Uplink-Merge** technique to address data heterogeneity. FedSUM achieves the same communication and memory cost as SCAFFOLD (Karimireddy et al., 2020; Huang et al., 2023) through single-variable uplink communication, whereas FedSUM-B and FedSUM-CR further achieve single-variable downlink communication to match the cost of FedAvg.

- **Unified and Novel Convergence Results:** We establish unified convergence rates for the FedSUM family, showing that, under specific participation patterns, the rates recover those of algorithms tailored to those settings. This demonstrates both the adaptability and generality of our approach. Our convergence guarantees are novel in that they hold under arbitrary client participation while incorporating the delay metrics $\tau_{\max}$ and $\tau_{\text{avg}}$. In addition, the analysis requires only smoothness and bounded variance assumptions, without imposing any additional restrictions on data heterogeneity or on the objective functions.

## 1.2 RELATED WORKS

**Client participation patterns in FL.** A variety of strategies have been proposed to model client participation in FL. Early works such as FedAvg (McMahan et al., 2017; Li et al., 2019) and SCAFFOLD (Karimireddy et al., 2020; Huang et al., 2023) assume that the server selects a small subset of clients in each round, either uniformly at random or in proportion to local data volume. Later studies address heterogeneous and time-varying response rates $p_i^t$. Some works treat these rates as known and server-controlled (e.g., determined by solving a stochastic optimization problem) (Perazzone et al., 2022), while others model them as unknown but governed by a homogeneous Markov chain (Ribero et al., 2022; Xiang et al., 2024; Wang & Ji, 2023).

Wang & Ji (2022) propose a generalized FedAvg that amplifies parameter updates every $P$ rounds, requiring additional assumptions such as equal client availability within each $P$-round window to guarantee convergence. Similarly, Crawshaw & Liu (2024) design a SCAFFOLD variant that amplifies global parameters and local gradients every $P$ rounds, but assume $p_i^t$ remains constant within each window. Yang et al. (2022) study clients participating at will, but their guarantees hold only up to a non-zero residual error. (Cho et al., 2023) consider FedAvg with cyclic sampling, decided by the server, to accelerate convergence. Gu et al. (2021) consider a more general participation pattern, but only for strongly convex objectives; under nonconvex objectives, Gu et al. (2021); Yan et al. (2024) assume strictly bounded inactive periods, a condition that often contradicts random sampling.

**Algorithm design in FL.** Following the popularity of FedAvg (McMahan et al., 2017), numerous algorithms have sought to improve performance under data heterogeneity and varying client participation. One line of research focuses on refining the FedAvg framework. For instance, FedAWE (Xiang et al., 2024) amplifies client updates to compensate for missed computation during periods of client inactivity, and FedAU (Wang & Ji, 2023) introduces a weighted aggregation of client updates to mitigate the negative effects of client non-participation.

Another line of work enhances FL performance by introducing additional control variables. For example, SCAFFOLD and its variants (Karimireddy et al., 2020; Huang et al., 2023; Crawshaw & Liu, 2024) exchanges control variables to correct the update directions for both clients and the server. In contrast, FedVARP (Jhunjhunwala et al., 2022) and MIFA (Gu et al., 2021) maintain control variables on the server to adjust its update direction, but the number of these variables scales

linearly with the number of clients. FedLaAvg (Yan et al., 2024) stores previous updates as control variables and transmits their differences, ensuring the server's update direction incorporates most recent information from all clients.

Despite these advances, many methods rely on restrictive assumptions about data distributions or objective functions, exclude local updates, or only guarantee convergence up to a non-zero residual error. For example, several works impose additional assumptions on data heterogeneity, typically by bounding the divergence between local and global gradients (Wang & Ji, 2023; Xiang et al., 2024; Yu et al., 2019; Wang et al., 2022; Yuan & Li, 2022; Wang et al., 2020). Local updates, while useful, often introduce bias under heterogeneous data; therefore, methods such as FedLaAvg (Yan et al., 2024) completely eliminate them. Other approaches impose structural conditions on the objective functions, such as bounded stochastic gradients (Perazzone et al., 2022; Yan et al., 2024) or Lipschitz Hessians (Gu et al., 2021). SCAFFOLD (Karimireddy et al., 2020; Huang et al., 2023), when paired with uniform random sampling, avoids additional assumptions on data heterogeneity, but at the cost of increased communication and memory compared to FedAvg.

### 1.3 PROBLEM SETUP

Throughout the paper, $\|\cdot\|$ denotes the $\ell_2$ vector norm, and $\mathcal{N}$ denotes the index set $\{1, \ldots, N\}$. Additionally, the expectation $\mathbb{E}[\cdot]$ is taken over the randomness of the stochastic gradient.

In FL, our goal is to solve the following optimization problem:

$$\min_{x \in \mathbb{R}^p} f(x) := \frac{1}{N} \sum_{i=1}^{N} f_i(x), \quad \text{where } f_i(x) := \mathbb{E}_{\xi_i \sim \mathcal{D}_i} F_i(x; \xi_i). \quad (1)$$

Here, $F_i(x; \xi_i)$ denotes the local loss function evaluated at model $x$ on sample $\xi_i$, and $f_i(x)$ represents the local objective under the data distribution $\mathcal{D}_i$, which may vary significantly across clients.

We introduce the standing assumptions below.

**Assumption 1.1.** *(Smoothness) Each local objective $f_i$ has L-Lipschitz continuous gradients. That is, for any $x, y \in \mathbb{R}^d$ and $1 \le i \le N$, it holds that*

$$\|\nabla f_i(x) - \nabla f_i(y)\| \le L \|x - y\|.$$

**Assumption 1.2.** *(Bounded Variance) There exists $\sigma \ge 0$ such that for any $x \in \mathbb{R}^d$ and $1 \le i \le N$, we have*

$$\mathbb{E}_{\xi_i} \left[ \nabla F_i(x; \xi_i) \right] = \nabla f_i(x), \quad \mathbb{E}_{\xi_i} \left[ \|\nabla F_i(x; \xi_i) - \nabla f_i(x)\|^2 \right] \le \sigma^2,$$

*where $\xi_i \sim \mathcal{D}_i$ are i.i.d. local samples at client $i$.*

## 2 ARBITRARY CLIENT PARTICIPATION IN FL

Understanding the impact of different client participation patterns is crucial in FL, as practical constraints often prevent all clients from joining every round. In this section, we first review several commonly studied participation patterns, then extend the discussion to the general case of arbitrary client participation, whether controllable or uncontrollable, stochastic or deterministic, homogeneous or heterogeneous. To quantify the variability among different participation patterns, we introduce two delay metrics: the **maximum delay** $\tau_{\max}$, which measures the longest inactive period of any client, and the **average delay** $\tau_{\text{avg}}$, which captures the average frequency of client participation. This broader perspective enables a unified analysis of participation heterogeneity and its effect on convergence.

### 2.1 PARTICIPATION PATTERNS

Most FL methods make simplifying assumptions about client participation in the training process. Some approaches assume uniform random sampling controlled by the server (Case 1) (Karimireddy et al., 2020; Huang et al., 2023), while others model client participation with uncertain, dynamic, and independent client unavailability (Case 2), resulting in an uncontrollable participation pattern (Li et al., 2020; Yang et al., 2022; Wang & Ji, 2023; Xiang et al., 2024). Another class of methods adopts

a scheme where each client participates for a certain number of rounds and does not participate in the other rounds of a cycle, to improve convergence performance (e.g., Case 3) (Cho et al., 2023; Ding et al., 2024). These assumptions may constrain the applicability of FL algorithms, as more structured or round-correlated schemes, such as reshuffled cyclic participation (Case 4) (Malinovsky et al., 2023) or participation modeled by Markov processes with non-independent and non-stationary client availability, fall outside these assumptions.

We formally define the participation patterns (Case 1 to 4) below:

**Case 1: Uniform Random Sampling.** In each round $t$, the server uniformly and randomly samples a subset $\mathcal{S}_t \subseteq \mathcal{N}$ with $|\mathcal{S}_t| = S$.
**Case 2: Probability-Based Independent Participation.** Each client $i$ independently participates in round $t$ with a probability $p_i^t \in (0, 1]$, where $p_i^t \geq \delta > 0$.
**Case 3: Deterministic Cyclic Participation.** Clients are arranged in a fixed order, and a block of $S$ consecutive clients is selected in each round. The process continues sequentially and wraps around at the end, cycling deterministically through all clients.
**Case 4: Reshuffled Cyclic Participation.** Clients are arranged in a random order at the beginning of each epoch, and a block of $S$ consecutive clients is selected in each round. The process continues sequentially and wraps around at the end, cycling through all clients.

The variability in participation patterns motivates a unified treatment of arbitrary client participation and its characterization.

## 2.2 ARBITRARY CLIENT PARTICIPATION AND TWO DELAY METRICS

We consider an arbitrary client participation sequence $\{\mathcal{S}_t\}_{t=0}^{T-1}$, where $\mathcal{S}_t \subseteq \mathcal{N}$ denotes the set of active clients at round $t$. This sequence may vary arbitrarily from round to round or follow a predetermined schedule. A key challenge is to quantify the variability of participation patterns. We address this by considering two metrics, the maximum delay $\tau_{\max}$ and the average delay $\tau_{\mathrm{avg}}$ (Gu et al., 2021), which offer a concise means of quantifying client participation and facilitate our theoretical analysis.

We begin with defining the **last-selection time** for each client $i \in \mathcal{N}$ at round $t$ as follows:

$$a_{i,t} := \max \{j \leq t : i \in \mathcal{S}_j\}. \tag{2}$$

In other words, $a_{i,t}$ denotes the most recent round (up to $t$) in which client $i$ was active. By convention, $a_{i,t} = -1$ if client $i$ has never participated before round $t + 1$. Equivalently, $a_{i,t}$ records the last round client $i$ was included in the active set, and can be expressed recursively as:

$$a_{i,t} = \begin{cases} a_{i,t-1} & \text{if client } i \notin \mathcal{S}_t, \\ t & \text{if client } i \in \mathcal{S}_t, \end{cases} \text{ with } a_{i,-1} = -1. \tag{3}$$

The **per-round delay** at round $t$ is then defined as:

$$\tau_t = \max_{i \in \mathcal{N}} \{t - a_{i,t}\} \geq 0. \tag{4}$$

Here, $\tau_t$ represents the largest gap between the current round and the last time any client participated. We focus on this gap because, due to data heterogeneity, a single client's behavior can significantly influence overall performance. Accordingly, we define the two delay metrics as follows:

- The **maximum delay** is given by

$$\tau_{\max} = \max_{0 \leq t \leq T} \{\tau_t\},$$

  which represents the largest per-round delay observed over the entire training process.

- The **average delay** is given by

$$\tau_{\mathrm{avg}} = \frac{1}{T} \sum_{t=0}^{T-1} \tau_t,$$

  which captures the average per-round delay across all rounds.

**Remark 2.1.** (*Intuition Behind the Delay Metrics.*) *The metrics $\tau_{\max}$ and $\tau_{avg}$ offer a concise means of quantifying client participation. Smaller values indicate more frequent participation, which generally promotes faster convergence and better global model performance. In Section 4, we evaluate the performance of the proposed algorithms in light of these metrics, demonstrating their effectiveness in addressing varying client participation patterns in FL.*

## 3 THE FEDSUM FAMILY

In this section, we introduce the FedSUM family (**Fed**erated Learning with **S**tochastic **U**plink-**M**erge), which includes **FedSUM-B** (basic version), **FedSUM** (standard version), and **FedSUM-CR** (communication-reduced version). These algorithms address various challenges in FL under arbitrary client participation, without requiring additional assumptions on data heterogeneity.

### 3.1 FEDSUM-B: A SIMPLE FL APPROACH WITHOUT LOCAL UPDATES

We first propose **FedSUM-B** (see Algorithm 1), a simple FL approach that employs a single variable for both uplink and downlink communication in each round, and omits local updates.

The algorithm maintains local control variables $\{h_i^{(t)}\}_{i=1}^N$ on clients and a global control variable $y^{(t)}$ on the server. In each round $t$, each active client $i \in \mathcal{S}_t$ computes the stochastic gradient over $K$ mini-batches $\xi_i^{(t,k)}$ evaluated at the current received model $x^{(t)}$. The subscript $i$ refers to the client index, while the superscript $(t, k)$ denotes the $t$-th round and the $k$-th mini-batch. After local computations, clients update their control variables $\{h_i^{(t)}\}_{i=1}^N$ as follows:

$$h_i^{(t+1)} = \begin{cases} \frac{1}{K} \sum_{k=0}^{K-1} \nabla F_i(x^{(t)}; \xi_i^{(t,k)}), & \text{if } i \in \mathcal{S}_t, \\ h_i^{(t)}, & \text{otherwise,} \end{cases} = \begin{cases} \frac{1}{K} \sum_{k=0}^{K-1} \nabla F_i(x^{(a_{i,t})}; \xi_i^{(a_{i,t},k)}), & \text{if } a_{i,t} \geq 0, \\ \mathbf{0}, & \text{if } a_{i,t} = -1. \end{cases} \tag{5}$$

The first expression in Equation (5) corresponds to the standard update rule in Algorithm 1, whereas the second offers a high-level interpretation: $h_i^{(t)}$ represents the aggregated stochastic gradient computed by client $i$ during its most recent selection round prior to round $t$.

The FedSUM family employs a technique called **Stochastic Uplink-Merge**, in which each active client $i \in \mathcal{S}_t$ transmits only the difference $\delta_i^{(t)}$ between its current aggregated gradient and the most recent gradient it computed. Specifically, if the client $i \in \mathcal{S}_t$ has not participated before, the sending message $\delta_i^{(t)} = \frac{1}{K} \sum_{k=0}^{K-1} \nabla F_i(x^{(t)}; \xi_i^{(t,k)})$, otherwise, $\delta_i^{(t)}$ is given by:

$$\delta_i^{(t)} = \frac{1}{K} \sum_{k=0}^{K-1} \nabla F_i(x^{(t)}; \xi_i^{(t,k)}) - \frac{1}{K} \sum_{k=0}^{K-1} \nabla F_i(x^{(a_{i,t-1})}; \xi_i^{(a_{i,t-1},k)}). \tag{6}$$

The server then updates its control variable $y^{(t)}$ by incorporating $\delta_i^{(t)}$ from all active clients. Specifically, denoting $\mathcal{A}_t := \cup_{j \leq t} \mathcal{S}_t$, the update rule for $y^{(t)}$ is given by:

$$y^{(t)} = \sum_{j=0}^{t} \sum_{i \in \mathcal{S}_j} \delta_i^{(j)} = \sum_{i \in \mathcal{A}_t} \sum_{j=0}^{t} \delta_i^{(j)} \mathbb{1}_{\{j=a_{i,j}\}} = \frac{1}{K} \sum_{i \in \mathcal{A}_t} \sum_{k=0}^{K-1} \nabla F_i(x^{(a_{i,t})}; \xi_i^{(a_{i,t},k)}). \tag{7}$$

Since $y^{(t)}$ is formed from gradients received from each client with delays, more frequent client participation keeps $y^{(t)}$ closer to the current globally aggregated gradient. This directly motivates the use of the delay metrics $\tau_{\max}$ and $\tau_{avg}$ for analyzing arbitrary client participation.

**Benefits of Stochastic Uplink-Merge.** Since $y^{(t)}$ represents the sum of aggregated gradients from the most recent participation of each previously active client and serves as the server's update direction. This design allows the FedSUM family to address data heterogeneity by incorporating information from each client's latest participation round, as illustrated in Figure 2. Similar idea has been explored in earlier works (Gu et al., 2021; Yan et al., 2024; Ying et al., 2025).

Note that although FedSUM-B operates without local updates, it achieves competitive convergence and accuracy under sufficiently large batch sizes (see Figure 11 in Appendix H).

---

**Algorithm 1** FedSUM-B: Basic FL Without Local Updates

---

1: **Input:** initial model $x^{(0)}$, control variables $y^{(-1)}$, $\{h_i^{(0)}\}_{i=1}^N$ with value $\mathbf{0}$; global learning rate $\eta_g$; local learning rate $\eta_l$; batch size $K$; client participation $\{\mathcal{S}_t\}_{t=0}^{T-1}$.
2: **for** $t = 0, 1, \cdots, T - 1$ **do**
3:     Send $x^{(t)}$ to all clients $i \in \mathcal{S}_t$.
4:     **for** client $i \in \mathcal{S}_t$ in parallel **do**
5:         Receive $x^{(t)}$ and initialize local model $x_i^{(t,0)} = x^{(t)}$.
6:         **for** $k = 0, \cdots, K - 1$ **do**
7:             Compute a mini-batch gradient $g_i^{(t,k)} = \nabla F_i(x_i^{(t,0)}; \xi_i^{(t,k)})$.
8:         **end for**
9:         Update $x_i^{(t,1)} = x_i^{(t,0)} - \eta_l \sum_{k=0}^{K-1} g_i^{(t,k)}$.
10:         Compute $\delta_i^{(t)} = \frac{x^{(t)} - x_i^{(t,1)}}{\eta_l K} - h_i^{(t)}$ and send $\delta_i^{(t)}$ to the server.
11:         Update $h_i^{(t+1)} = \frac{x^{(t)} - x_i^{(t,1)}}{\eta_l K}$ (for $i \notin \mathcal{S}_t$, $h_i^{(t+1)} = h_i^{(t)}$).
12:     **end for**
13:     Update $y^{(t)} = y^{(t-1)} + \sum_{i \in \mathcal{S}_t} \delta_i^{(t)}$ and $x^{(t+1)} = x^{(t)} - \frac{\eta_g \eta_l K}{N} y^{(t)}$.
14: **end for**
15: **Server** outputs $x^{(T)}$.

---

## 3.2 FedSUM: Enhancing FL with Local Updates

Building on FedSUM-B, we introduce the standard algorithm **FedSUM** (see Algorithm 2), which also employs the **Stochastic Uplink-Merge** technique but supports local updates. Compared to FedSUM-B, it requires additional communication of $y^{(t)}$ to compute the correction direction of local updates.

The control variables in FedSUM are similar to those in FedSUM-B, except that they correspond to client models with local updates rather than a single global model. Specifically, at each round $t$, the control variables $\{h_i^{(t)}\}_{i=1}^N$ and the global variable $y^{(t)}$ are updated as follows:

$$h_i^{(t+1)} = \frac{1}{K} \sum_{k=0}^{K-1} \nabla F_i(x_i^{(a_{i,t},k)}; \xi_i^{(a_{i,t},k)}), \text{ if } a_{i,t} \geq 0, \text{ and } \mathbf{0}, \text{ otherwise.}$$

$$y^{(t)} = \frac{1}{K} \sum_{i \in \mathcal{A}_t} \sum_{k=0}^{K-1} \nabla F_i(x_i^{(a_{i,t},k)}; \xi_i^{(a_{i,t},k)}). \tag{8}$$

Here, $x_i^{(a_{i,t},k)}$ denotes the local model of client $i$ at round $a_{i,t}$ during the $k$-th local update.

**Correction direction $y_i^{(t)}$ of local updates.** A key challenge when performing local updates is that the gradient $\nabla F_i(x_i^{(t,k)}; \xi_i^{(t,k)})$ computed on client $i$'s local data can be biased due to data heterogeneity. To mitigate this bias, we introduce the correction direction $y_i^{(t)}$ by sending the previous round's aggregated gradient $y^{(t-1)}$ from the server to each client $i \in \mathcal{S}_t$. By subtracting the client's own previous gradient, $h_i^{(t)}$, we obtain a correction direction that incorporates the gradients of other clients. Formally,

$$y_i^{(t)} = -h_i^{(t)} + y^{(t-1)} = \frac{1}{K} \sum_{j \in \mathcal{A}_t \setminus \{i\}} \sum_{k=0}^{K-1} \nabla F_j(x_j^{(a_{j,t-1},k)}; \xi_j^{(a_{j,t-1},k)}), \tag{9}$$

where $x_j^{(a_{j,t-1},k)}$ is client $j$'s local model from its last participation round $a_{i,t-1}$ at the $k$-th update step. This correction direction reflects the most recent aggregated gradients from other previously active clients, helping to align client $i$'s updates with the global descent direction, as illustrated in Figure 3.

---

**Algorithm 2** FedSUM: FL with Local Updates

---

1: **Input:** initial model $x^{(0)}$, control variables $y^{(-1)}$, $\{h_i^{(0)}\}_{i=1}^N$ with value $\mathbf{0}$; global learning rate $\eta_g$; local learning rate $\eta_l$; local steps $K$; client participation $\{\mathcal{S}_t\}_{t=0}^{T-1}$.

2: **for** $t = 0, 1, \cdots, T-1$ **do**

3:    Send $x^{(t)}$ and $y^{(t-1)}$ to all clients $i \in \mathcal{S}_t$.

4:    **for** client $i \in \mathcal{S}_t$ in parallel **do**

5:       Receive $x^{(t)}$ and $y^{(t-1)}$ and initialize local model $x_i^{(t,0)} = x^{(t)}$.

6:       Compute local update correction direction $y_i^{(t)} := -h_i^{(t)} + y^{(t-1)}$.

7:       **for** $k = 0, \cdots, K-1$ **do**

8:          Compute a mini-batch gradient $g_i^{(t,k)} = \nabla F_i(x_i^{(t,k)}; \xi_i^{(t,k)})$.

9:          Locally update $x_i^{(t,k+1)} = x_i^{(t,k)} - \frac{\eta_l}{N}\left(g_i^{(t,k)} + y_i^{(t)}\right)$.

10:       **end for**

11:       Compute $\delta_i^{(t)} = \frac{N(x^{(t)} - x_i^{(t,K)})}{\eta_l K} - y_i^{(t)} - h_i^{(t)}$ and send $\delta_i^{(t)}$ to the server.

12:       Update $h_i^{(t+1)} = \frac{N(x^{(t)} - x_i^{(t,K)})}{\eta_l K} - y_i^{(t)}$ (for $i \notin \mathcal{S}_t$, $h_i^{(t+1)} = h_i^{(t)}$).

13:    **end for**

14:    Update $y^{(t)} = y^{(t-1)} + \sum_{i \in \mathcal{S}_t} \delta_i^{(t)}$ and $x^{(t+1)} = x^{(t)} - \frac{\eta_g \eta_l K}{N} y^{(t)}$.

15: **end for**

16: **Server** outputs $x^{(T)}$.

---

### 3.3 FEDSUM-CR: REDUCING COMMUNICATION COST IN FEDSUM

We further introduce **FedSUM-CR** (see Algorithm 3 in Appendix E), to enhance the communication efficiency of FedSUM and achieve *single-variable communication for both uplink and downlink*.

In each round $t$ of FedSUM-CR, instead of computing the correction direction $y_i^{(t)}$ by receiving $y^{(t-1)}$ from the server as in FedSUM, each active client $i \in \mathcal{S}_t$ compute $y_i^{(t)}$ locally using its stored variables $a_i^{(t)}$ and $z_i^{(t)}$, thereby reducing communication overhead. Specifically, each client maintains additional control variables $a_i^{(t)}$ and $z_i^{(t)}$, updated as

$$a_i^{(t+1)} := \begin{cases} t, & \text{if } i \in \mathcal{S}_t, \\ a_i^{(t)}, & \text{otherwise,} \end{cases} = a_{i,t}, \quad z_i^{(t+1)} := \begin{cases} x^{(t)}, & \text{if } i \in \mathcal{S}_t, \\ z_i^{(t)}, & \text{otherwise,} \end{cases} = x^{(a_{i,t})}, \quad (10)$$

with $x^{(-1)} := x^{(0)}$ for convenience. Here, $a_i^{(t)} = a_{i,t-1}$ records the last round when client $i$ was selected prior to round $t$, and $z_i^{(t)} = x^{(a_{i,t-1})}$ stores the most recent model it received from the server before round $t$. Using these variables, the correction direction for client $i \in \mathcal{S}_t$ at round $t$ is given by

$$y_i^{(t)} := \frac{N}{\eta_g \eta_l K} \cdot \frac{z_i^{(t)} - x^{(t)}}{t - a_i^{(t)}} - h_i^{(t)} = \frac{1}{(t - a_{i,t-1})K} \sum_{p=a_{i,t-1}}^{t-1} \sum_{j \in \mathcal{A}_p/\{i\}} \sum_{k=0}^{K-1} \nabla F_j(x_j^{(a_{j,p},k)}; \xi_j^{(a_{j,p},k)}),$$

$$(11)$$

which represents the average aggregated gradient from other active clients between client $i$'s last selection round and round $t-1$. This correction direction aligns client $i$'s update with the global descent direction by incorporating gradients from other active clients during these intervening rounds.

The correction direction in FedSUM-CR (Equation 11) is similar to that in FedSUM (Equation 9), as both adjust local updates by incorporating the influence of other clients. Whether using the averaged gradients from intervening rounds (as in FedSUM-CR) or the most recent aggregated gradient (as in FedSUM) yields no significant theoretical distinction, since both approaches align the correction with the global descent direction. This equivalence is reflected in the convergence rates presented in Section 4.

**Comparison among the FedSUM family.** The main difference between **FedSUM-B**, **FedSUM**, and **FedSUM-CR** lies in how the correction direction $y_i^{(t)}$ is obtained. FedSUM-B omits local

updates and thus does not require $y_i^{(t)}$, resulting in reduced communication overhead. FedSUM introduces additional communication by sending the aggregated gradient $y^{(t)}$ from the server to clients for computing $y_i^{(t)}$. FedSUM-CR achieves the same correction with reduced communication by requiring extra memory to store the most recently received model $z_i^{(t)}$.

## 4 CONVERGENCE RESULTS

In this section, we present the unified convergence result for the FedSUM family, summarized in Theorem 4.1, and characterize their behavior under different client participation patterns.

**Theorem 4.1.** *Suppose Assumptions 1.1 and 1.2 hold. Under an arbitrary client participation sequence $\{\mathcal{S}_t\}_{t=0}^{T-1}$ characterized by $\tau_{\max}$ and $\tau_{avg}$, suppose the learning rates for FedSUM-B, Fed-SUM, and FedSUM-CR are set as*

$$\eta_g = \frac{1}{\sqrt{\tau_{\max}}}, \text{ and } \eta_l = \min\left\{\frac{1}{10\sqrt{\tau_{\max}}KL}, \frac{\sqrt{N\tau_{\max}\Delta_f}}{\sqrt{\max\{1,\tau_{avg}\}KTL\sigma^2}}\right\}.$$

*Then all three algorithms achieve the following convergence rate:*

$$\frac{1}{T}\sum_{t=0}^{T-1}\mathbb{E}\big[\|\nabla f(x^{(t)})\|^2\big|\,\{\mathcal{S}_t\}_{t=0}^{T-1}\big] \leq \frac{30\sqrt{(1+\tau_{avg})L\sigma^2\Delta_f}}{\sqrt{NKT}} + \frac{20\tau_{\max}\left(L\Delta_f + F_0\right)}{T}, \quad (12)$$

*where $\Delta_f := f(x^{(0)}) - f^*$ and $F_0 := \frac{1}{N}\sum_{i=1}^{N}\|\nabla f_i(x^{(0)})\|^2$.*

When the participation sequence $\{\mathcal{S}_t\}_{t=0}^{T-1}$ involves randomness, we can further take full expectation with respect to $\{\mathcal{S}_t\}_{t=0}^{T-1}$ on both sides of Equation (12). This allows us to characterize the average performance for the FedSUM family, where $\mathbb{E}[\tau_{\max}]$ and $\mathbb{E}[\tau_{\text{avg}}]$ quantify the impact of participation patterns on the convergence rate.

**Corollary 4.1.** *Under the same setting as in Theorem 4.1, suppose the participation sequence $\{\mathcal{S}_t\}_{t=0}^{T-1}$ involves randomness. Then all three algorithms achieve the following convergence rate:*

$$\frac{1}{T}\sum_{t=0}^{T-1}\mathbb{E}\big[\|\nabla f(x^{(t)})\|^2\big] \leq \frac{30\sqrt{(1+\mathbb{E}[\tau_{avg}])L\sigma^2\Delta_f}}{\sqrt{NKT}} + \frac{20\mathbb{E}[\tau_{\max}]\left(L\Delta_f + F_0\right)}{T}. \quad (13)$$

Corollary 4.1 directly follows from Theorem 4.1 noting that $\mathbb{E}[\sqrt{(1+\tau_{\text{avg}})}] \leq \sqrt{(1+\mathbb{E}[\tau_{\text{avg}}])}$.

**Remark 4.1.** *The upper bounds in Theorem 4.1 and Corollary 4.1 show that, smaller values of $\tau_{avg}$ and $\tau_{\max}$, or their expectations under a random participation pattern, lead to faster convergence rates. This result is consistent with the intuition that smaller delays that indicates more frequent client participation, improve overall convergence.*

**Remark 4.2.** *Moreover, the FedSUM family remains convergent even when the inactive period grows with $T$ (e.g., $\log(T)$), which is a significant improvement compared to Yan et al. (2024); Gu et al. (2021), where convergence requires the inactive period to be strictly bounded.*

The convergence guarantees in Theorem 4.1 apply directly to the participation patterns introduced in Section 2. As summarized in Table 1, our analysis not only unifies existing results but also extends them in scope. These results indicate that the delay metrics $\tau_{\max}$ and $\tau_{\text{avg}}$ accurately capture participation heterogeneity under arbitrary client participation schemes, while the FedSUM family achieves state-of-the-art efficiency across diverse participation regimes.

## 5 EXPERIMENTS

**Overview.** We evaluate the FedSUM family on real-world datasets to corroborate our theoretical analysis and compare against state-of-the-art baselines, including comparisons between Fed-SUM variants. Specifically, we consider a federated learning system with one parameter server and

Table 1: Convergence behavior of the FedSUM family under different participation patterns, compared with related works. Here $\tilde{\mathcal{O}}$ hides logarithmic factors.

| Pattern | Convergence Rate | Related Works |
|---------|-----------------|---------------|
| Case 1 | $\tilde{\mathcal{O}}\left(\frac{\sqrt{L\sigma^2\Delta_f}}{\sqrt{SKT}} + \frac{N(L\Delta_f+F_0)}{ST}\right)$ | Matches SCALLION without communication compression (Huang et al., 2023) (up to log factors). |
| Case 2 | $\tilde{\mathcal{O}}\left(\frac{\sqrt{L\sigma^2\Delta_f}}{\sqrt{\delta NKT}} + \frac{L\Delta_f+F_0}{\delta T}\right)$ | Matches FedAWE (Xiang et al., 2024). |
| Case 3 | $\mathcal{O}\left(\frac{\sqrt{L\sigma^2\Delta_f}}{\sqrt{SKT}} + \frac{N(L\Delta_f+F_0)}{ST}\right)$ | Not directly comparable to the CyCP framework (Cho et al., 2023), which requires the PL condition on $f$. |
| Case 4 | $\mathcal{O}\left(\frac{\sqrt{L\sigma^2\Delta_f}}{\sqrt{SKT}} + \frac{N(L\Delta_f+F_0)}{ST}\right)$ | None to the best of our knowledge. |

$N = 100$ clients, where clients become available intermittently. We consider three image classification tasks using the MNIST (LeCun et al., 2010), SVHN (Netzer et al., 2011), and CIFAR-10 (Krizhevsky et al., 2009) datasets, each containing 10 classes. For these tasks, we train convolutional neural network (CNN) models with slightly different architectures. To simulate highly heterogeneous local data distributions, the image class distribution at client $i$ follows a Dirichlet distribution with parameter $\alpha = 0.1$ (Xiang et al., 2024; Crawshaw & Liu, 2024; Wang & Ji, 2023); see Figure 4 in Appendix H for a visualization. Additional specifications and experimental results are also included in Appendix H.

**Client participation patterns.** We evaluate the FedSUM family under three participation patterns inspired by real-world FL scenarios and prior work: (i) **P1**: The server randomly selects $S = 20$ clients per round, a controllable pattern (Karimireddy et al., 2020; Huang et al., 2023). (ii) **P2**: Each client participates independently with a fixed probability $S/N$, a stationary and uncontrollable pattern (Wang & Ji, 2023; Xiang et al., 2024). (iii) **P3**: Each client participates with a time-varying probability $p_i^t$ from a sine trajectory, representing a non-stationary, uncontrollable pattern (Bonawitz et al., 2019).

**Baselines.** We compare FedSUM (standard version) against several baseline algorithms that operate without prior knowledge of client participation patterns during training. These baselines are grouped into two categorizes: (i) methods that refine the FedAvg framework, including FedAvg applied to active clients (McMahan et al., 2017), FedAU (Wang & Ji, 2023), and FedAWE (Xiang et al., 2024). (ii) methods that enhance FL performance by incorporating additional control variables, including FedVARP (Jhunjhunwala et al., 2022), MIFA (Gu et al., 2021), and SCAFFOLD (Karimireddy et al., 2020; Huang et al., 2023). For fairness, all algorithms use the same local and global learning rates, selected via grid search based on the optimal performance of FedAvg (see Appendix H for details).

Figure 1a presents the training loss and test accuracy curves for the three datasets. FedSUM and FedSUM-CR achieve faster convergence and greater stability than the baseline algorithms, which we attribute to its stochastic uplink-merge technique combined with the correction direction. Figure 1b further demonstrates that FedSUM and FedSUM-CR achieve faster convergence with respect to the communication workload. Notably, FedSUM-CR maintains the strong performance of FedSUM with improved communication efficiency, making it the most effective algorithm in this comparison. Detailed performance plots, offering clearer comparisons for specific participation patterns and last communication rounds, are provided in Appendix H.3.

## 6 CONCLUSION

This work presents the first comprehensive analysis of federated learning under arbitrary client participation. We introduce two delay metrics that quantify the impact of participation variability and propose the FedSUM family of algorithms, which achieve both efficiency and robustness through the stochastic uplink-merge technique. Our unified convergence guarantees recover known rates in special cases and extend to arbitrary participation patterns under only standard smoothness and

bounded variance assumptions. These contributions position the FedSUM family as a practical and theoretically grounded framework for federated learning across diverse participation scenarios.

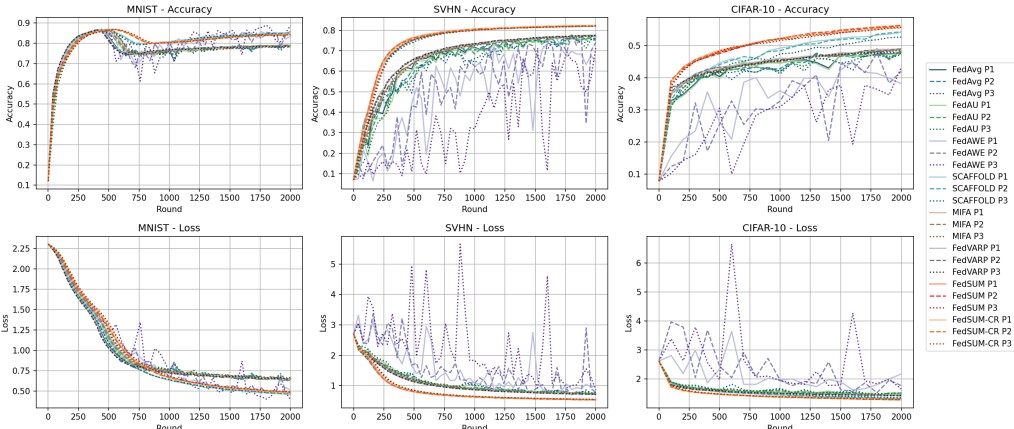

(a) Performance vs. Communication Rounds.

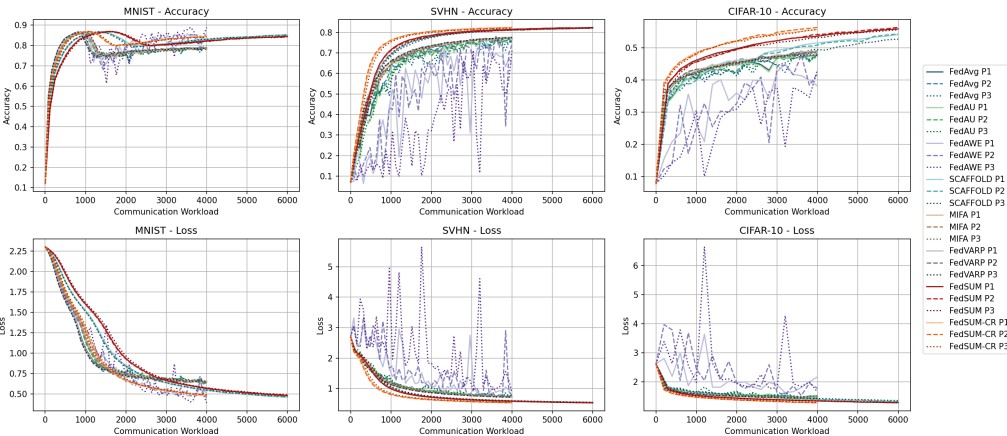

(b) Performance vs. Communication Workload.

Figure 1: Training loss and test accuracy curves for CNN models on three datasets, comparing different FL algorithms under various client participation patterns. The performance is evaluated against (a) the number of communication rounds and (b) the cumulative communication workload. For the workload, one unit corresponds to the transmission of a full-sized model.

REPRODUCIBILITY STATEMENT

We ensure reproducibility by providing all implementation details in Appendix H, including an anonymous code link (with random seeds), hyperparameters, learning rate schedules, optimizer settings, dataset splits, evaluation protocols, and hardware specifications. These details enable independent researchers to replicate our results.

ETHICS STATEMENT

This work proposes federated learning algorithms under arbitrary client participation. All experiments were conducted on publicly available datasets (e.g., MNIST, SVHN, CIFAR-10) with convolutional neural networks specified in Appendix H. We do not foresee direct risks of harm arising from our methodology and we emphasize that our contributions are intended to advance research in optimization and federated learning. We encourage responsible and ethical use of the algorithms.

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

CONTENTS

# I   LLM Usage                                                                                        54

In Appendix A, we show the visual representation of the techniques behind the FedSUM Family. In Appendix B, we provide the notations and preliminary results. In Appendix C, we derive the convergence results of FedSUM-B. In Appendix D, we present the convergence results of FedSUM. In Appendix E, we introduce the convergence results of FedSUM-CR. In Appendix F, we derive convergence bounds under various client participation patterns. In Appendix H, we present the experimental setups and additional experiments.

## A   VISUAL REPRESENTATION OF THE TECHNIQUES BEHIND THE FEDSUM FAMILY

To address data heterogeneity during model updates, we introduce the Stochastic Uplink-Merge (SUM) technique, which is illustrated in Figure 2. The figure demonstrates how the SUM technique operates across three rounds (from round $t$ to round $t + 2$) when client $i$ is selected to compute the update direction.

In this scenario, client $i$'s local minima $x_i^*$ is far from the global minima $x^*$, representing strong data heterogeneity. Despite client $i$ being the only one activated during these rounds, the SUM technique ensures that the server's model update is influenced by the gradients from other clients, even though they are not activated during the period from round $t$ to round $t + 2$. This method helps to avoid the bias that can arise from frequently activated clients, such as client $i$, and ensures that the update direction is better aligned with the global model. This approach allows the server to effectively handle data heterogeneity and reduce the negative impact of the participation bias issue discussed in Ribero et al. (2022); Sun et al. (2024).

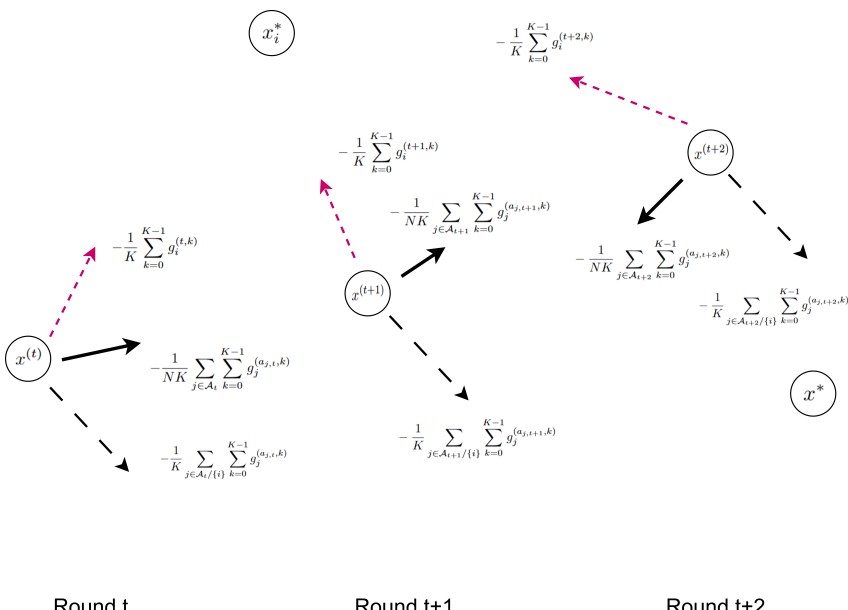

Figure 2:   Illustration of the Stochastic Uplink-Merge (SUM) technique in addressing data heterogeneity and participation bias issue during server's model updates.

In Figure 3, we illustrate the local update rule for the active client $i$ with both the (stochastic) gradient $g_i^{(t,k)}$ and the correction direction $y_i^{(t)}$, and show how this approach helps mitigate data heterogeneity. Despite the fact that the local updates initially tend to converge towards the local

minima $x_i^*$, the correction direction $y_i^{(t)}$ ensures that the update direction is guided towards the global minimum $x^*$. This process is demonstrated for local updates from $k$ to $k+3$, where the update at each step accounts for the correction direction, preventing the local update from being biased by the local minima.

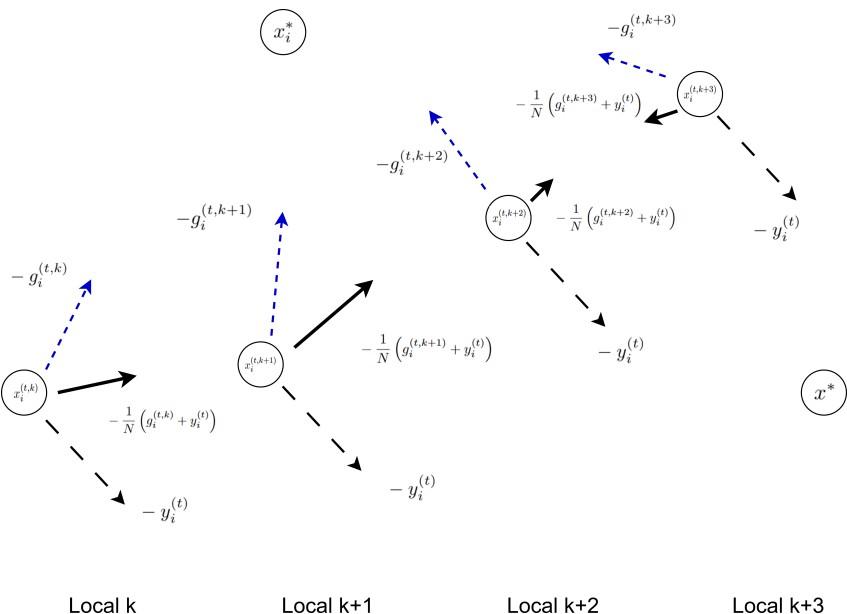

Figure 3: Illustration of the usage of the correction direction $y_i^t$ in addressing data heterogeneity during client's local model updates.

## B NOTATIONS AND PRELIMINARY RESULTS

For the simplicity of the proof, we let $\eta_l = N\eta_l'$ and $\eta_g = \frac{\eta_g'}{N}$, where $\eta_l'$ and $\eta_g'$ are the learning rates in the proposed FedSUM family (i.e., we change the learning rate symbols by letting $\eta_l \leftarrow N\eta_l$ and $\eta_g \leftarrow \frac{\eta_g}{N}$).

Define $\mathcal{F}^{(-1)} := \emptyset$, $\mathcal{F}_i^{(t,k)} := \sigma\left(\cup_{0\le j\le k}\left\{\xi_i^{(t,j)}\right\} \cup \mathcal{F}^{(t-1)}\right)$, and let $\mathcal{F}^{(t)} := \sigma\left(\cup_{i\in\mathcal{N},0\le k\le K-1}\mathcal{F}_i^{(t,k)} \cup \{\mathcal{S}_j\}_{j=0}^t\right)$ if the participation sequence $\{\mathcal{S}_t\}_{t=0}^{T-1}$ is random; otherwise, $\mathcal{F}^{(t)} := \sigma\left(\cup_{i\in\mathcal{N},0\le k\le K-1}\mathcal{F}_i^{(t,k)}\right)$, where $\sigma(\cdot)$ denotes the $\sigma$-algebra generated by the random variables within the parentheses. We use $\mathbb{E}[\cdot]$ to denote the expectation over the stochastic gradient. Additionally, each $a_{i,t}$, $\tau_t$, $\tau_{\max}$ and $\tau_{\text{avg}}$ are functions of $\mathcal{S}_t$, where $t = 0, \ldots, T-1$. These can be directly used and evaluated by taking the full expectation.

We define $S_t := |\mathcal{S}_t| \le N$ and $A_t := |\mathcal{A}_t| \le N$. The stochastic gradient of client $i \in \mathcal{S}_t$ at round $t$ and the $k$-th local update is denoted as $g_i(x_i^{(t,0)}, \xi_i^{(t,k)}) := \nabla F_i(x_i^{(t,0)}, \xi_i^{(t,k)})$ in FedSUM-B, and $g_i(x_i^{(t,k)}, \xi_i^{(t,k)}) := \nabla F_i(x_i^{(t,k)}, \xi_i^{(t,k)})$ in FedSUM and FedSUM-CR for simplicity.

Lemma B.1 provides a basic variance upper bound for the aggregated stochastic gradient, which is crucial for analyzing the convergence of the FedSUM family, and holds obviously since each client draws independent samples in every round and local update.

**Lemma B.1.** *Suppose Assumption 1.2 holds. Then, for any active set $\mathcal{A}_t$ of clients, it holds that*

$$\mathbb{E}\left\|\sum_{i\in\mathcal{A}_t}\sum_{k=0}^{K-1}\left[g_i(x^{(a_{i,t},k)}, \xi_i^{(a_{i,t},k)}) - \nabla f_i(x^{(a_{i,t},k)})\right]\right\|^2 \le NK\sigma^2.$$

*Proof.* By the independently and randomly sampled $\{\xi_i\}$, it implies that

$$\mathbb{E}\left\|\sum_{i\in\mathcal{A}_t}\sum_{k=0}^{K-1}\left[g_i(x^{(a_{i,t},k)},\xi_i^{(a_{i,t},k)})-\nabla f_i(x^{(a_{i,t},k)})\right]\right\|^2$$

$$=\sum_{i\in\mathcal{A}_t}\mathbb{E}\left\|\sum_{k=0}^{K-1}\left[g_i(x^{(a_{i,t},k)},\xi_i^{(a_{i,t},k)})-\nabla f_i(x^{(a_{i,t},k)})\right]\right\|^2$$

$$=\sum_{i\in\mathcal{A}_t}\sum_{k=0}^{K-1}\mathbb{E}\left\|\left[g_i(x^{(a_{i,t},k)},\xi_i^{(a_{i,t},k)})-\nabla f_i(x^{(a_{i,t},k)})\right]\right\|^2$$

$$\leq\sum_{i\in\mathcal{A}_t}\sum_{k=0}^{K-1}\sigma^2\leq A_tK\sigma^2\leq NK\sigma^2.$$

$\square$

Lemma B.2 establishes a basic relationship between the minimum last-selection round among clients and the maximum delay $\tau_{\max}$.

**Lemma B.2.** *For any $k\geq 0$, it holds that*

$$\min_{i\in\mathcal{N}}\{a_{i,k+\tau_{\max}}\}\geq k.$$

*Proof.* First, by the definition of $\tau_{\max}$, for any client $i\in\mathcal{N}$, it should be updated during the iteration from $k$ to $k+\tau_{\max}$. Otherwise, there exists $k'\leq T$ and client $i\in\mathcal{N}$ such that client $i$ is not selected during the iteration from $k'$ to $k'+\tau_{\max}$. Then, it implies that

$$\tau_{\max}\geq k'+\tau_{\max}-\min_{j\in\mathcal{N}}\{a_{j,k'+\tau_{\max}}\}\geq k'+\tau_{\max}-a_{i,k'+\tau_{\max}}$$

$$\geq k'+\tau_{\max}-a_{i,k'+\tau_{\max}}\geq k'+\tau_{\max}-a_{i,k'-1}$$

$$=\tau_{\max}+1+(k'-1-a_{i,k'-1})\geq\tau_{\max}+1,$$

where we use the facts that $a_{i,k'-1}=a_{i,k'}=\cdots=a_{i,k'+\tau_{\max}}$ in the fourth inequality and $t-a_{i,t}\geq 0$ in the last inequality. It conducts to a contradiction.

Thus, for any $k$ and for any client $i\in\mathcal{N}$, client $i$ should be selected at least once during the iteration from $k$ to $k+\tau_{\max}$ i.e. there exists $k^*\in\{k,k+1,\cdots,k+\tau_{\max}\}$ such that $a_{i,k^*}=k^*$. As a result, we have

$$a_{i,k+\tau_{\max}}\geq a_{i,k^*}=k^*\geq k.$$

Therefore, we have

$$\min_{i\in\mathcal{N}}\{a_{i,k+\tau_{\max}}\}\geq k.$$

$\square$

## C  CONVERGENCE ANALYSIS FOR FEDSUM-B

In this section, we get the convergence result of FedSUM-B.

We begin with the update direction $y^{(t)}$ of server in FedSUM-B shown as follows.

$$y^{(t)}=\frac{1}{K}\sum_{i\in\mathcal{A}_t}\sum_{k=0}^{K-1}g_i(x^{(a_{i,t})},\xi_i^{(a_{i,t},k)}). \tag{14}$$

**Lemma C.1.** *Suppose Assumption 1.1 and 1.2 hold. Then, we have*

$$\sum_{t=0}^{T-1}\mathbb{E}\left\|x^{(t+1)}-x^{(t)}\right\|^2\leq 2\frac{\eta_g^2\eta_l^2KT}{N}\sigma^2+2\frac{\eta_g^2\eta_l^2K^2}{N^2}\sum_{t=0}^{T-1}\mathbb{E}\left\|\sum_{i\in\mathcal{A}_t}\nabla f_i(x^{(a_{i,t})})\right\|^2.$$

*Proof.* Invoking Equation (14), we have

$$\left\| x^{(t+1)} - x^{(t)} \right\|^2 = \frac{\eta_g^2 \eta_l^2 K^2}{N^2} \left\| y^{(t)} \right\|^2 = \frac{\eta_g^2 \eta_l^2}{N^2} \left\| \sum_{i \in \mathcal{A}_t} \sum_{k=0}^{K-1} g_i(x^{(a_{i,t})}, \xi_i^{(a_{i,t},k)}) \right\|^2$$

$$\leq 2 \frac{\eta_g^2 \eta_l^2}{N^2} \left\| \sum_{i \in \mathcal{A}_t} \sum_{k=0}^{K-1} \left[ g_i(x^{(a_{i,t})}, \xi_i^{(a_{i,t},k)}) - \nabla f_i(x^{(a_{i,t})}) \right] \right\|^2 + 2 \frac{\eta_g^2 \eta_l^2 K^2}{N^2} \left\| \sum_{i \in \mathcal{A}_t} \nabla f_i(x^{a_{i,t}}) \right\|.$$

$$(15)$$

Invoking Assumption 1.2, we have, by taking expectation on both sides of Equation (15),

$$\mathbb{E} \left\| x^{(t+1)} - x^{(t)} \right\|^2 \leq 2 \frac{\eta_g^2 \eta_l^2}{N^2} \sum_{i \in \mathcal{A}_t} \sum_{k=0}^{K-1} \sigma^2 + 2 \frac{\eta_g^2 \eta_l^2 K^2}{N^2} \mathbb{E} \left\| \sum_{i \in \mathcal{A}_t} \nabla f_i(x^{(a_{i,t})}) \right\|^2$$

$$\leq 2 \frac{\eta_g^2 \eta_l^2 K}{N} \sigma^2 + 2 \frac{\eta_g^2 \eta_l^2 K^2}{N^2} \mathbb{E} \left\| \sum_{i \in \mathcal{A}_t} \nabla f_i(x^{(a_{i,t})}) \right\|^2.$$

$$(16)$$

We get the desired result by summing over $t$ from $0$ to $T-1$ on the both sides of Equation (16). □

**Lemma C.2.** *Suppose Assumption 1.1 and 1.2 hold. Then, we have*

$$\sum_{t=0}^{T-1} \sum_{p=t-\tau_t}^{t-1} \mathbb{E} \left\| \sum_{j \in \mathcal{A}_p} \nabla f_j(x^{(a_{j,p})}) \right\|^2 \leq \tau_{\max} \sum_{t=0}^{T-1} \mathbb{E} \left\| \sum_{i \in \mathcal{A}_t} \nabla f_i(x^{(a_{i,t})}) \right\|^2$$

*Proof.* Invoking Lemma B.2, we have $t - \tau_t \geq \min\{0, t - \tau_{\max}\}$ and hence,

$$\sum_{t=0}^{T-1} \sum_{p=t-\tau_t}^{t-1} \mathbb{E} \left\| \sum_{j \in \mathcal{A}_p} \nabla f_j(x^{(a_{j,p})}) \right\|^2 \leq \sum_{t=0}^{T-1} \sum_{p=\min\{0, t-\tau_{\max}\}}^{t-1} \mathbb{E} \left\| \sum_{j \in \mathcal{A}_p} \nabla f_j(x^{(a_{j,p})}) \right\|^2$$

$$\leq \sum_{p=0}^{T-1} \sum_{t=p+1}^{t=p+\tau_{\max}} \mathbb{E} \left\| \sum_{j \in \mathcal{A}_p} \nabla f_j(x^{(a_{j,p})}) \right\|^2 \leq \tau_{\max} \sum_{t=0}^{T-1} \mathbb{E} \left\| \sum_{j \in \mathcal{A}_t} \nabla f_j(x^{(a_{j,t})}) \right\|^2,$$

which gives the desired result. □

Corollary C.3 holds directly by Lemma C.2.

**Corollary C.3.** *Suppose Assumption 1.1 and 1.2 hold. Then, we have*

$$\sum_{t=0}^{T-1} \tau_t \sum_{p=t-\tau_t}^{t-1} \mathbb{E} \left\| \sum_{j \in \mathcal{A}_p} \nabla f_j(x^{(a_{j,p})}) \right\|^2 \leq \tau_{\max}^2 \sum_{t=0}^{T-1} \mathbb{E} \left\| \sum_{i \in \mathcal{A}_t} \nabla f_i(x^{(a_{i,t})}) \right\|^2$$

**Lemma C.4.** *Suppose Assumption 1.1 and 1.2 hold. Then, we have*

$$\sum_{t=0}^{T} \mathbb{E} \left\| \sum_{i=1}^{n} \nabla f_i(x^{(t)}) - \sum_{i \in \mathcal{A}_t} \nabla f_i(x^{(a_{i,t})}) \right\|^2$$

$$\leq 4 \eta_g^2 \eta_l^2 \tau_{\max} \tau_{avg} N K T L^2 \sigma^2 + 4 \eta_g^2 \eta_l^2 \tau_{\max}^2 K^2 L^2 \sum_{t=0}^{T} \mathbb{E} \left\| \sum_{i \in \mathcal{A}_t} \nabla f_i(x^{(a_{i,t})}) \right\|^2 + 2 N^2 \tau_{\max} F_0,$$

*where* $F_0 := \frac{1}{N} \sum_{i=1}^{N} \left\| \nabla f_i(x^{(0)}) \right\|^2.$

*Proof.* It holds that

$$\left\|\sum_{i\in\mathcal{N}}\nabla f_i(x^{(t)})-\sum_{i\in\mathcal{A}_t}\nabla f_i(x^{(a_{i,t})})\right\|^2$$

$$=\left\|\sum_{i\in\mathcal{N}}\left[\nabla f_i(x^{(t)})-\nabla f_i(x^{(a_{i,t})})\right]+\sum_{i\in\mathcal{N}/\mathcal{A}_t}\nabla f_i(x^{(0)})\right\|^2 \quad (17)$$

$$\leq 2N\sum_{i\in\mathcal{N}}\left\|\nabla f_i(x^{(t)})-\nabla f_i(x^{(a_{i,t})})\right\|^2+2\left(N-A_t\right)\sum_{i\in\mathcal{N}/\mathcal{A}_t}\left\|\nabla f_i(x^{(0)})\right\|^2$$

$$\leq 2NL^2\sum_{i\in\mathcal{N}}\left\|x^{(t)}-x^{(a_{i,t})}\right\|^2+2\left(N-A_t\right)\sum_{i\in\mathcal{N}/\mathcal{A}_t}\left\|\nabla f_i(x^{(0)})\right\|^2.$$

Invoking Equation (14), we have, by taking full expectation, that

$$\sum_{i\in\mathcal{N}}\mathbb{E}\left\|x^{(t)}-x^{(a_{i,t})}\right\|^2=\sum_{i\in\mathcal{N}}\mathbb{E}\left\|\sum_{p=a_{i,t}}^{t-1}\left[x^{(p+1)}-x^{(p)}\right]\right\|^2=\frac{\eta_g^2\eta_l^2K^2}{N^2}\sum_{i\in\mathcal{N}}\mathbb{E}\left\|\sum_{p=a_{i,t}}^{t-1}y^{(p)}\right\|^2$$

$$=\frac{\eta_g^2\eta_l^2}{N^2}\sum_{i\in\mathcal{N}}\mathbb{E}\left\|\sum_{p=a_{i,t}}^{t-1}\sum_{j\in\mathcal{A}_p}\sum_{k=0}^{K-1}g_j(x^{(a_{j,p})},\xi_j^{(a_{j,p},k)})\right\|^2$$

$$\leq 2\frac{\eta_g^2\eta_l^2}{N^2}\sum_{i\in\mathcal{N}}\mathbb{E}\left\|\sum_{p=a_{i,t}}^{t-1}\sum_{j\in\mathcal{A}_p}\sum_{k=0}^{K-1}\left[g_j(x^{(a_{j,p})},\xi_j^{(a_{j,p},k)})-\nabla f_j(x^{(a_{j,p})})\right]\right\|^2$$

$$+2\frac{\eta_g^2\eta_l^2K^2}{N^2}\sum_{i\in\mathcal{N}}\mathbb{E}\left\|\sum_{p=a_{i,t}}^{t-1}\sum_{j\in\mathcal{A}_p}\nabla f_j(x^{(a_{j,p})})\right\|^2$$

$$(18)$$

We can bound the first term in Equation (45) as follows:

$$2\frac{\eta_g^2\eta_l^2}{N^2}\sum_{i\in\mathcal{N}}\mathbb{E}\left\|\sum_{p=a_{i,t}}^{t-1}\sum_{j\in\mathcal{A}_p}\sum_{k=0}^{K-1}\left[g_j(x^{(a_{j,p})},\xi_j^{(a_{j,p},k)})-\nabla f_j(x^{(a_{j,p})})\right]\right\|^2$$

$$\leq 2\frac{\eta_g^2\eta_l^2}{N^2}\sum_{i\in\mathcal{N}}(t-a_{i,t})\sum_{p=a_{i,t}}^{t-1}\mathbb{E}\left\|\sum_{j\in\mathcal{A}_p}\sum_{k=0}^{K-1}\left[g_j(x^{(a_{j,p})},\xi_j^{(a_{j,p},k)})-\nabla f_j(x^{(a_{j,p})})\right]\right\|^2$$

$$\leq 2\frac{\eta_g^2\eta_l^2}{N^2}\sum_{i\in\mathcal{N}}\tau_{\max}\sum_{p=a_{i,t}}^{t-1}\sum_{j\in\mathcal{A}_p}\sum_{k=0}^{K-1}\mathbb{E}\left\|g_j(x^{(a_{j,p})},\xi_j^{(a_{j,p},k)})-\nabla f_j(x^{(a_{j,p})})\right\|^2$$

$$\leq 2\frac{\eta_g^2\eta_l^2\tau_{\max}}{N^2}\sum_{i\in\mathcal{N}}\sum_{p=a_{i,t}}^{t-1}\sum_{j\in\mathcal{A}_p}\sum_{k=0}^{K-1}\sigma^2\leq 2\frac{\eta_g^2\eta_l^2\tau_{\max}K}{N}\sum_{i\in\mathcal{N}}\sum_{p=a_{i,t}}^{t-1}\sigma^2=2\frac{\eta_g^2\eta_l^2\tau_{\max}K}{N}\sum_{i\in\mathcal{N}}(t-a_{i,t})\sigma^2$$

$$\leq 2\eta_g^2\eta_l^2K\tau_t\tau_{\max}\sigma^2.$$

$$(19)$$

For the second term in Equation (45), we have

$$2\frac{\eta_g^2\eta_l^2K^2}{N^2}\sum_{i\in\mathcal{N}}\mathbb{E}\left\|\sum_{p=a_{i,t}}^{t-1}\sum_{j\in\mathcal{A}_p}\nabla f_j(x^{(a_{j,p})})\right\|^2\leq 2\frac{\eta_g^2\eta_l^2K^2}{N^2}\sum_{i\in\mathcal{N}}(t-a_{i,t})\sum_{p=a_{i,t}}^{t-1}\mathbb{E}\left\|\sum_{j\in\mathcal{A}_p}\nabla f_j(x^{(a_{j,p})})\right\|^2$$

$$\leq 2\frac{\eta_g^2\eta_l^2K^2}{N^2}\sum_{i\in\mathcal{N}}\tau_t\sum_{p=t-\tau_t}^{t-1}\mathbb{E}\left\|\sum_{j\in\mathcal{A}_p}\nabla f_j(x^{(a_{j,p})})\right\|^2=2\frac{\eta_g^2\eta_l^2K^2}{N}\tau_t\sum_{p=t-\tau_t}^{t-1}\mathbb{E}\left\|\sum_{j\in\mathcal{A}_p}\nabla f_j(x^{(a_{j,p})})\right\|^2.$$

$$(20)$$

Summing over $t$ from $0$ to $T-1$ and invoking Lemma C.2 and Corollary C.3, we have

$$\sum_{t=0}^{T-1}\sum_{i\in\mathcal{N}}\mathbb{E}\left\|x^{(t)}-x^{(a_{i,t})}\right\|^2$$

$$\leq 2\eta_g^2\eta_l^2 K\tau_{\max}\sigma^2\sum_{t=0}^{T-1}\tau_t+2\frac{\eta_g^2\eta_l^2 K^2}{N}\sum_{t=0}^{T-1}\tau_t\sum_{p=t-\tau_t}^{t-1}\mathbb{E}\left\|\sum_{j\in\mathcal{A}_p}\nabla f_j(x^{(a_{j,p})})\right\|^2 \tag{21}$$

$$\leq 2\eta_g^2\eta_l^2\tau_{\text{avg}}\tau_{\max}KT\sigma^2+2\frac{\eta_g^2\eta_l^2 K^2}{N}\tau_{\max}^2\sum_{t=0}^{T-1}\mathbb{E}\left\|\sum_{i\in\mathcal{A}_t}\nabla f_i(x^{(a_{i,t})})\right\|^2.$$

Furthermore, we can bound the second term in Equation (17) as follows:

$$2\sum_{t=0}^{T-1}(N-A_t)\sum_{i\in\mathcal{N}/\mathcal{A}_t}\left\|\nabla f_i(x^{(0)})\right\|^2\leq 2\sum_{t=0}^{T-1}(N-A_t)\sum_{i=1}^N\left\|\nabla f_i(x^{(0)})\right\|^2$$

$$=2\sum_{t=0}^{T-1}\sum_{j\in\mathcal{N}/\mathcal{A}_t}\sum_{i=1}^N\left\|\nabla f_i(x^{(0)})\right\|^2=2\sum_{t=0}^{T-1}\sum_{j\in\mathcal{N}}\mathbb{1}_{\{j\notin\mathcal{A}_t\}}\sum_{i=1}^N\left\|\nabla f_i(x^{(0)})\right\|^2 \tag{22}$$

$$\leq 2N^2\tau_{\max}F_0,$$

where $F_0:=\frac{1}{N}\sum_{i=1}^N\left\|\nabla f_i(x^{(0)})\right\|^2$ and we use $\sum_{j\in\mathcal{N}}\sum_{t=0}^T\mathbb{1}_{\{j\notin\mathcal{A}_t\}}\leq\sum_{i=1}^N\tau_{\max}=N\tau_{\max}$ in the last inequality.

Finally, we can combine Equation (18), (19), (20), (21) and (22) to get the desired result:

$$\sum_{t=0}^{T-1}\mathbb{E}\left\|\sum_{i\in\mathcal{N}}\nabla f_i(x^{(t)})-\sum_{i\in\mathcal{A}_t}\nabla f_i(x^{(a_{i,t})})\right\|^2$$

$$\leq 4\eta_g^2\eta_l^2\tau_{\max}\tau_{\text{avg}}NKTL^2\sigma^2+4\eta_g^2\eta_l^2\tau_{\max}^2K^2L^2\sum_{t=0}^T\mathbb{E}\left\|\sum_{i\in\mathcal{A}_t}\nabla f_i(x^{(a_{i,t})})\right\|^2+2N^2\tau_{\max}F_0. \tag{23}$$

$$\square$$

**Lemma C.5.** *Suppose Assumption 1.1 and 1.2 hold. Then, we have*

$$-\frac{\eta_g\eta_l}{N}\sum_{t=0}^{T-1}\mathbb{E}\left\langle\nabla f(x^{(t)}),\sum_{i\in A_t}\sum_{k=0}^{K-1}\left[g_i(x^{(a_{i,t})},\xi_i^{(a_{i,t},k)})-\nabla f_i(x^{(a_{i,t})})\right]\right\rangle$$

$$\leq 3\frac{\eta_g^2\eta_l^2 K}{N}\tau_{avg}\sigma^2 LT+2\frac{\eta_g^2\eta_l^2 K^2}{N^2}\tau_{\max}L\sum_{t=0}^T\mathbb{E}\left\|\sum_{i\in\mathcal{A}_t}\nabla f_i(x^{(a_{i,t})})\right\|^2.$$

*Proof.* By the independently and randomly sampled $\xi_i^{(a_{i,t})}$, we have, for each $i\in\mathcal{A}_t$, and $0\leq k\leq K-1$,

$$\mathbb{E}\left\langle\nabla f(x^{(\min_{i\in\mathcal{N}}\{a_{i,t}\})}),g_i(x^{(a_{i,t},k)},\xi_i^{(a_{i,t},k)})-\nabla f_i(x^{(a_{i,t})})\right\rangle=0. \tag{24}$$

We then derive that

$$-\frac{\eta_g\eta_l}{N}\mathbb{E}\left\langle\nabla f(x^{(t)}),\sum_{i\in A_t}\sum_{k=0}^{K-1}\left[g_i(x^{(a_{i,t})},\xi_i^{(a_{i,t},k)})-\nabla f_i(x^{(a_{i,t})})\right]\right\rangle$$

$$=-\frac{\eta_g\eta_l}{N}\mathbb{E}\left\langle\nabla f(x^{(t)})-\nabla f(x^{(t-\tau_t)}),\sum_{i\in\mathcal{A}_t}\sum_{k=0}^{K-1}\left[g_i(x^{(a_{i,t})},\xi_i^{(a_{i,t},k)})-\nabla f_i(x^{(a_{i,t})})\right]\right\rangle. \tag{25}$$

Then, if $\tau_t > 0$, it holds by picking $\alpha = \frac{1}{\tau_t L}$ in the Cauchy-Schwarz inequality $\langle a, b \rangle \leq \alpha \|a\|^2 + \frac{1}{\alpha} \|b\|^2$ that,

$$
-\frac{\eta_g \eta_l}{N} \mathbb{E} \left\langle \nabla f(x^{(t)}) - \nabla f(x^{(t-\tau_t)}), \sum_{i \in \mathcal{A}_t} \sum_{k=0}^{K-1} \left[ g_i(x^{(a_{i,t})}, \xi_i^{(a_{i,t},k)}) - \nabla f_i(x^{(a_{i,t})}) \right] \right\rangle
$$

$$
\leq \frac{1}{\tau_t L} \mathbb{E} \left\| \nabla f(x^{(t)}) - \nabla f(x^{(t-\tau_t)}) \right\|^2 + \frac{\eta_g^2 \eta_l^2}{N^2} \tau_t L \mathbb{E} \left\| \sum_{i \in \mathcal{A}_t} \sum_{k=0}^{K-1} \left[ g_i(x^{(a_{i,t})}, \xi_i^{(a_{i,t},k)}) - \nabla f_i(x^{(a_{i,t})}) \right] \right\|^2.
$$

$$(26)$$

With the fact that, by taking full expectation,

$$
\tau_t \mathbb{E} \left\| \sum_{i \in \mathcal{A}_t} \sum_{k=0}^{K-1} \left[ g_i(x^{(a_{i,t})}, \xi_i^{(a_{i,t},k)}) - \nabla f_i(x^{(a_{i,t})}) \right] \right\|^2
$$

$$
\leq \tau_t \sum_{i \in \mathcal{A}_t} \sum_{k=0}^{K-1} \mathbb{E} \left\| g_i(x^{(a_{i,t})}, \xi_i^{(a_{i,t},k)}) - \nabla f_i(x^{(a_{i,t})}) \right\|^2 \leq K N \tau_t \sigma^2,
$$

and

$$
\mathbb{E} \left\| \nabla f(x^{(t)}) - \nabla f(x^{(t-\tau_t)}) \right\|^2 \leq L^2 \mathbb{E} \left\| x^{(t)} - x^{(t-\tau_t)} \right\|^2 = \frac{\eta_g^2 \eta_l^2}{N^2} L^2 \mathbb{E} \left\| \sum_{p=t-\tau_t}^{t-1} \sum_{j \in \mathcal{A}_p} \sum_{k=0}^{K-1} g_j(x^{(a_{j,p})}, \xi_j^{(a_{j,p},k)}) \right\|^2
$$

$$
\leq 2 \frac{\eta_g^2 \eta_l^2}{N^2} L^2 \mathbb{E} \left\| \sum_{p=t-\tau_t}^{t-1} \sum_{j \in \mathcal{A}_p} \sum_{k=0}^{K-1} \left[ g_j(x^{(a_{j,p})}, \xi_j^{(a_{j,p},k)}) - \nabla f_j(x^{(a_{j,p})}) \right] \right\|^2
$$

$$
+ 2 \frac{\eta_g^2 \eta_l^2 K^2}{N^2} L^2 \mathbb{E} \left\| \sum_{p=t-\tau_t}^{t-1} \sum_{j \in \mathcal{A}_p} \nabla f_j(x^{(a_{j,p})}) \right\|^2,
$$

we have

$$
\mathbb{E} \left\| \sum_{p=t-\tau_t}^{t-1} \sum_{j \in \mathcal{A}_p} \sum_{k=0}^{K-1} \left[ g_j(x^{(a_{j,p})}, \xi_j^{(a_{j,p},k)}) - \nabla f_j(x^{(a_{j,p})}) \right] \right\|^2
$$

$$
\leq \tau_t \sum_{p=t-\tau_t}^{t-1} \mathbb{E} \left\| \sum_{j \in \mathcal{A}_p} \sum_{k=0}^{K-1} \left[ g_j(x^{(a_{j,p})}, \xi_j^{(a_{j,p},k)}) - \nabla f_j(x^{(a_{j,p})}) \right] \right\|^2
$$

$$
= \tau_t \sum_{p=t-\tau_t}^{t-1} \sum_{j \in \mathcal{A}_p} \sum_{k=0}^{K-1} \mathbb{E} \left\| g_j(x^{(a_{j,p})}, \xi_j^{(a_{j,p},k)}) - \nabla f_j(x^{(a_{j,p})}) \right\|^2 \leq N K \tau_t^2 \sigma^2,
$$

and

$$
\mathbb{E} \left\| \sum_{p=t-\tau_t}^{t-1} \sum_{j \in \mathcal{A}_p} \nabla f_j(x^{(a_{j,p})}) \right\|^2 \leq \tau_t \sum_{p=t-\tau_t}^{t-1} \mathbb{E} \left\| \sum_{j \in \mathcal{A}_p} \nabla f_j(x^{(a_{j,p})}) \right\|^2.
$$

Thus, summing over $t$ from $0$ to $T$ on both sides of Equation (26) and invoking Corollary D.3, we have the following bound holds no matter whether $\tau_t < t$ or not:

$$-\frac{\eta_g \eta_l}{N} \sum_{t=0}^{T-1} \mathbb{E} \left\langle \nabla f(x^{(t)}), \sum_{i \in \mathcal{A}_t} \left[ g_i(x^{(a_{i,t})}, \xi_i^{(a_{i,t})}) - \nabla f_i(x^{(a_{i,t})}) \right] \right\rangle$$

$$\leq \frac{\eta_g^2 \eta_l^2}{N^2} \sum_{t=0}^{T-1} \tau_t K N \sigma^2 L + 2 \frac{\eta_g^2 \eta_l^2}{N^2} N K \sum_{t=0}^{T-1} \tau_t \sigma^2 L + 2 \frac{\eta_g^2 \eta_l^2 K^2}{N^2} L \sum_{t=0}^{T-1} \sum_{p=t-\tau_t}^{t-1} \mathbb{E} \left\| \sum_{j \in \mathcal{A}_p} \nabla f_j(x^{(a_{j,p})}) \right\|^2$$

$$\leq 3 \frac{\eta_g^2 \eta_l^2 K}{N} \tau_{\text{avg}} \sigma^2 L T + 2 \frac{\eta_g^2 \eta_l^2 K^2}{N^2} \tau_{\max} L \sum_{t=0}^{T} \mathbb{E} \left\| \sum_{i \in \mathcal{A}_t} \nabla f_i(x^{(a_{i,t})}) \right\|^2, \tag{27}$$

which gives the desired result. $\qquad\square$

**Lemma C.6.** *Suppose Assumption 1.1 and 1.2 hold. Then, for $\eta_g \eta_l K \leq \frac{1}{8\tau_{\max}L}$, we have*

$$\frac{1}{T} \sum_{t=0}^{T-1} \mathbb{E} \left\| \nabla f(x^{(t)}) \right\|^2 \leq \frac{2\Delta_f}{\eta_g \eta_l K T} + 10 \frac{\eta_g \eta_l}{N} \max \left\{ 1, \tau_{avg} \right\} \sigma^2 L + \frac{2\tau_{\max} F_0}{T},$$

*where $\Delta_f := f(x^{(0)}) - f^*$ and $F_0 := \frac{1}{N} \sum_{i=1}^{N} \left\| \nabla f_i(x^{(0)}) \right\|^2$.*

*Proof.* By Assumption 1.1, the function $f := \frac{1}{N} \sum_{i=1}^{N} f_i$ is $L$-smooth. Then, it holds by the descent lemma that

$$\mathbb{E} f(x^{(t+1)}) \leq \mathbb{E} f(x^{(t)}) + \mathbb{E} \left\langle \nabla f(x^{(t)}), x^{(t+1)} - x^{(t)} \right\rangle + \frac{L}{2} \left\| x^{(t+1)} - x^{(t)} \right\|^2. \tag{28}$$

For the inner product term in Equation (28), there holds that

$$\mathbb{E} \left\langle \nabla f(x^{(t)}), x^{(t+1)} - x^{(t)} \right\rangle = \mathbb{E} \left\langle \nabla f(x^{(t)}), -\frac{\eta_g \eta_l K}{N} y^{(t)} \right\rangle$$

$$= -\frac{\eta_g \eta_l}{N} \mathbb{E} \left\langle \nabla f(x^{(t)}), \sum_{i \in A_t} \sum_{k=0}^{K-1} \left[ g_i(x^{(a_{i,t})}, \xi_i^{(a_{i,t},k)}) - \nabla f_i(x^{(a_{i,t})}) \right] \right\rangle \tag{29}$$

$$- \frac{\eta_g \eta_l K}{N} \mathbb{E} \left\langle \nabla f(x^{(t)}), \sum_{i \in A_t} \nabla f_i(x^{(a_{i,t})}) \right\rangle.$$

For the second term in Equation (29), we have

$$-\frac{\eta_g \eta_l K}{N} \mathbb{E} \left\langle \nabla f(x^{(t)}), \sum_{i \in A_t} \nabla f_i(x^{(a_{i,t})}) \right\rangle$$

$$= -\frac{\eta_g \eta_l K}{N^2} \mathbb{E} \left\langle \sum_{i=1}^{N} \nabla f_i(x^{(t)}), \sum_{i \in A_t} \nabla f_i(x^{(a_{i,t})}) \right\rangle$$

$$= -\frac{\eta_g \eta_l K}{2N^2} \mathbb{E} \left[ \left\| \sum_{i=1}^{N} \nabla f_i(x^{(t)}) \right\|^2 + \left\| \sum_{i \in A_t} \nabla f_i(x^{(a_{i,t})}) \right\|^2 - \left\| \sum_{i=1}^{N} \nabla f_i(x^{(t)}) - \sum_{i \in A_t} \nabla f_i(x^{(a_{i,t})}) \right\|^2 \right]$$

$$= -\frac{1}{2} \eta_g \eta_l K \mathbb{E} \left\| \nabla f(x^{(t)}) \right\|^2 - \frac{\eta_g \eta_l K}{2N^2} \mathbb{E} \left\| \sum_{i \in A_t} \nabla f_i(x^{(a_{i,t})}) \right\|^2 + \frac{\eta_g \eta_l K}{2N^2} \mathbb{E} \left\| \sum_{i=1}^{N} \nabla f_i(x^{(t)}) - \sum_{i \in A_t} \nabla f_i(x^{(a_{i,t})}) \right\|^2. \tag{30}$$

Then, summing over $t$ from $0$ to $T-1$ on both sides of Equation (28) and taking full expectation, we have

$$
\begin{aligned}
0 \leq & \Delta_f - \frac{\eta_g \eta_l}{N} \sum_{t=0}^{T-1} \mathbb{E}\left\langle \nabla f(x^{(t)}), \sum_{i \in A_t} \sum_{k=0}^{K-1} \left[ g_i(x^{(a_{i,t})}, \xi_i^{(a_{i,t},k)}) - \nabla f_i(x^{(a_{i,t})}) \right] \right\rangle \\
& - \frac{1}{2} \eta_g \eta_l K \sum_{t=0}^{T-1} \mathbb{E}\left\| \nabla f(x^{(t)}) \right\|^2 - \frac{\eta_g \eta_l K}{2N^2} \sum_{t=0}^{T-1} \mathbb{E}\left\| \sum_{i \in A_t} \nabla f_i(x^{(a_{i,t})}) \right\|^2 \\
& + \frac{\eta_g \eta_l K}{2N^2} \sum_{t=0}^{T-1} \mathbb{E}\left\| \sum_{i=1}^{N} \nabla f_i(x^{(t)}) - \sum_{i \in A_t} \nabla f_i(x^{(a_{i,t})}) \right\|^2 + \frac{L}{2} \sum_{t=0}^{T-1} \mathbb{E}\left\| x^{(t+1)} - x^{(t)} \right\|^2,
\end{aligned}
\tag{31}
$$

where $\Delta_f := f(x^{(0)}) - f^*$. Implementing Lemma C.1 into Equation (31), we have

$$
\begin{aligned}
0 \leq & \Delta_f - \frac{\eta_g \eta_l}{N} \sum_{t=0}^{T-1} \mathbb{E}\left\langle \nabla f(x^{(t)}), \sum_{i \in A_t} \sum_{k=0}^{K-1} \left[ g_i(x^{(a_{i,t})}, \xi_i^{(a_{i,t},k)}) - \nabla f_i(x^{(a_{i,t})}) \right] \right\rangle \\
& - \frac{1}{2} \eta_g \eta_l K \sum_{t=0}^{T-1} \mathbb{E}\left\| \nabla f(x^{(t)}) \right\|^2 - \frac{\eta_g \eta_l K}{2N^2}(1 - 2\eta_g \eta_l KL) \sum_{t=0}^{T-1} \mathbb{E}\left\| \sum_{i \in A_t} \nabla f_i(x^{(a_{i,t})}) \right\|^2 \\
& + \frac{\eta_g \eta_l K}{2N^2} \sum_{t=0}^{T-1} \mathbb{E}\left\| \sum_{i=1}^{N} \nabla f_i(x^{(t)}) - \sum_{i \in A_t} \nabla f_i(x^{(a_{i,t})}) \right\|^2 + \frac{\eta_g^2 \eta_l^2 KT}{N} \sigma^2 L.
\end{aligned}
\tag{32}
$$

Implementing Lemma C.4 into Equation (32), we have

$$
\begin{aligned}
0 \leq & \Delta_f - \frac{\eta_g \eta_l}{N} \sum_{t=0}^{T-1} \mathbb{E}\left\langle \nabla f(x^{(t)}), \sum_{i \in A_t} \sum_{k=0}^{K-1} \left[ g_i(x^{(a_{i,t})}, \xi_i^{(a_{i,t},k)}) - \nabla f_i(x^{(a_{i,t})}) \right] \right\rangle \\
& - \frac{1}{2} \eta_g \eta_l K \sum_{t=0}^{T-1} \mathbb{E}\left\| \nabla f(x^{(t)}) \right\|^2 - \frac{\eta_g \eta_l K}{2N^2}(1 - 2\eta_g \eta_l KL - 4\eta_g^2 \eta_l^2 \tau_{\max}^2 K^2 L^2) \sum_{t=0}^{T-1} \mathbb{E}\left\| \sum_{i \in A_t} \nabla f_i(x^{(a_{i,t})}) \right\|^2 \\
& + \frac{\eta_g^2 \eta_l^2 KT}{N} \sigma^2 L + \frac{2\eta_g^3 \eta_l^3 \tau_{\max} \tau_{\text{avg}} K^2 T}{N} \sigma^2 L^2 + \eta_g \eta_l \tau_{\max} K F_0.
\end{aligned}
\tag{33}
$$

Implementing Lemma C.5 into Equation (33), we have

$$
\begin{aligned}
0 \leq & \Delta_f + 3\frac{\eta_g^2 \eta_l^2 K}{N} \tau_{\text{avg}} T \sigma^2 L - \frac{1}{2} \eta_g \eta_l K \sum_{t=0}^{T-1} \mathbb{E}\left\| \nabla f(x^{(t)}) \right\|^2 \\
& - \frac{\eta_g \eta_l K}{2N^2}(1 - 2\eta_g \eta_l KL - 4\eta_g^2 \eta_l^2 \tau_{\max}^2 K^2 L^2 - 4\eta_g \eta_l K \tau_{\max} L) \sum_{t=0}^{T-1} \mathbb{E}\left\| \sum_{i \in A_t} \nabla f_i(x^{(a_{i,t})}) \right\|^2 \\
& + \frac{\eta_g^2 \eta_l^2 KT}{N} \sigma^2 L + \frac{2\eta_g^3 \eta_l^3 \tau_{\max} \tau_{\text{avg}} K^2 T}{N} \sigma^2 L^2 + \eta_g \eta_l \tau_{\max} K F_0.
\end{aligned}
\tag{34}
$$

For $\eta_g \eta_l K \leq \frac{1}{8\tau_{\max}L}$, we have

$$
1 - 2\eta_g \eta_l KL - 4\eta_g^2 \eta_l^2 \tau_{\max}^2 K^2 L^2 - 4\eta_g \eta_l K \tau_{\max} L > 1 - \frac{1}{4\tau_{\max}} - \frac{1}{16} - \frac{1}{2} > \frac{1}{16} > 0.
$$

Hence, for $\eta_g \eta_l K \leq \frac{1}{8\tau_{\max}L}$, we can rearrange Equation (34) to get

$$
\frac{1}{2} \eta_g \eta_l K \sum_{t=0}^{T} \mathbb{E}\left\| \nabla f(x^{(t)}) \right\|^2 \leq \Delta_f + 5\frac{\eta_g^2 \eta_l^2 K}{N} \max\{1, \tau_{\text{avg}}\} T \sigma^2 L + \eta_g \eta_l K \tau_{\max} F_0.
\tag{35}
$$

where we use the fact that $\frac{2\eta_g^3 \eta_l^3 \tau_{\max} \tau_{\text{avg}} K^2 T}{N} \sigma^2 L^2 \leq \frac{\eta_g^2 \eta_l^2 K}{N} T \sigma^2 L$ for $\eta_g \eta_l K \leq \frac{1}{8\tau_{\max}L}$.

Dividing both sides of Equation (35) by $\frac{1}{2}\eta_g\eta_l KT$, we have

$$\frac{1}{T}\sum_{t=0}^{T-1}\mathbb{E}\left\|\nabla f(x^{(t)})\right\|^2 \leq \frac{2\Delta_f}{\eta_g\eta_l KT} + 10\frac{\eta_g\eta_l}{N}\max\{1,\tau_{\text{avg}}\}\sigma^2 L + \frac{2\tau_{\max}F_0}{T}, \tag{36}$$

which completes the proof. $\qquad\square$

**Theorem C.1.** *Suppose Assumption 1.1 and 1.2 hold. Then, for* $\eta_g\eta_l = \min\left\{\frac{\sqrt{N\Delta_f}}{\sqrt{\max\{1,\tau_{\text{avg}}\}KTL\sigma^2}}, \frac{1}{10K\tau_{\max}L}\right\}$, *we have*

$$\frac{1}{T}\sum_{t=0}^{T-1}\mathbb{E}\left\|\nabla f(x^{(t)})\right\|^2 \leq \frac{30\sqrt{\max\{1,\tau_{\text{avg}}\}L\sigma^2\Delta_f}}{\sqrt{NKT}} + \frac{20\tau_{\max}(F_0 + L\Delta_f)}{T},$$

*where* $\Delta_f := f(x^{(0)}) - f^*$ *and* $F_0 := \frac{1}{N}\sum_{i=1}^{N}\left\|\nabla f_i(x^{(0)})\right\|^2$.

*Proof.* Invoking Lemma C.6, we have, for $\eta_g\eta_l K \leq \frac{1}{8\tau_{\max}L}$,

$$\frac{1}{T}\sum_{t=0}^{T-1}\mathbb{E}\left\|\nabla f(x^{(t)})\right\|^2 \leq \frac{2\Delta_f}{\eta_g\eta_l KT} + 10\frac{\eta_g\eta_l}{N}\max\{1,\tau_{\text{avg}}\}\sigma^2 L + \frac{2\tau_{\max}F_0}{T}. \tag{37}$$

For

$$\eta_g\eta_l = \min\left\{\frac{\sqrt{N\Delta_f}}{\sqrt{\max\{1,\tau_{\text{avg}}\}KTL\sigma^2}}, \frac{1}{10K\tau_{\max}L}\right\},$$

we have

$$\frac{2\Delta_f}{\eta_g\eta_l KT} \leq \frac{2\sqrt{\max\{1,\tau_{\text{avg}}\}L\sigma^2\Delta_f}}{\sqrt{NKT}} + \frac{20\tau_{\max}L\Delta_f}{T},$$

$$10\frac{\eta_g\eta_l}{N}\max\{1,\tau_{\text{avg}}\}\sigma^2 L \leq \frac{10\sqrt{\max\{1,\tau_{\text{avg}}\}L\sigma^2\Delta_f}}{\sqrt{NKT}}.$$

Thus, we can combine the above two inequalities with Equation (37) to get

$$\frac{1}{T}\sum_{t=0}^{T-1}\mathbb{E}\left\|\nabla f(x^{(t)})\right\|^2 \leq \frac{12\sqrt{\max\{1,\tau_{\text{avg}}\}L\sigma^2\Delta_f}}{\sqrt{NKT}} + \frac{20\tau_{\max}(F_0 + L\Delta_f)}{T}, \tag{38}$$

which gives the desired result by scaling the constants appropriately. $\qquad\square$

# D CONVERGENCE ANALYSIS FOR FEDSUM

In this section, we prove the convergence result of FedSUM.

We derive the update direction $y^{(t)}$ in FedSUM as follows.

$$y^{(t)} = \frac{1}{K}\sum_{i\in\mathcal{A}_t}\sum_{k=0}^{K-1}g_i(x_i^{(a_{i,t,k})}, \xi_i^{(a_{i,t,k})}). \tag{39}$$

Furthermore, the local control variable $h_i^{(t)}$ and local update correction direction $y_i^{(t)}$ for $i \in \mathcal{A}_t$ are defined as follows:

$$h_i^{(t)} = \frac{1}{K}\sum_{k=0}^{K-1}g_i(x_i^{(a_{i,t-1,k})}, \xi_i^{(a_{i,t-1,k})}). \tag{40}$$

$$y_i^{(t)} = \frac{1}{K}\sum_{j\in\mathcal{A}_{t-1}/\{i\}}\sum_{k=0}^{K-1}g_j(x_j^{(a_{j,t-1,k})}, \xi_j^{(a_{j,t-1,k})}). \tag{41}$$

Lemma D.1 establishes a basic relationship between the difference of two consecutive model updates and the aggregated gradient, shown as follows.

**Lemma D.1.** *Suppose Assumption 1.1 and 1.2 hold. Then, it holds that*

$$\sum_{t=0}^{T-1} \mathbb{E} \left\| x^{(t+1)} - x^{(t)} \right\|^2 \leq \frac{2\eta_g^2 \eta_l^2 KT\sigma^2}{N} + \frac{2\eta_g^2 \eta_l^2}{N^2} \sum_{t=0}^{T-1} \mathbb{E} \left\| \sum_{i \in \mathcal{A}_t} \sum_{k=0}^{K-1} \nabla f_i(x_i^{(a_{i,t},k)}) \right\|^2.$$

*Proof.* Invoking Equation (8), we have, by line 17 in Algorithm 2,

$$\left\| x^{(t+1)} - x^{(t)} \right\|^2 = \frac{\eta_g^2 \eta_l^2 K^2}{N^2} \left\| y^{(t)} \right\|^2 = \frac{\eta_g^2 \eta_l^2}{N^2} \left\| \sum_{i \in \mathcal{A}_t} \sum_{k=0}^{K-1} g_i(x_i^{(a_{i,t},k)}, \xi_i^{(a_{i,t},k)}) \right\|^2$$

$$\leq \frac{2\eta_g^2 \eta_l^2}{N^2} \left\| \sum_{i \in \mathcal{A}_t} \sum_{k=0}^{K-1} \left[ g_i(x_i^{(a_{i,t},k)}, \xi_i^{(a_{i,t},k)}) - \nabla f_i(x_i^{(a_{i,t},k)}) \right] \right\|^2 + \frac{2\eta_g^2 \eta_l^2}{N^2} \left\| \sum_{i \in \mathcal{A}_t} \sum_{k=0}^{K-1} \nabla f_i(x_i^{(a_{i,t},k)}) \right\|^2. \tag{42}$$

Invoking Assumption 1.2 and Lemma B.1, we have, by taking expectation on both sides of Equation (42),

$$\mathbb{E} \left\| x^{(t+1)} - x^{(t)} \right\|^2 \leq \frac{2\eta_g^2 \eta_l^2 K}{N} \sigma^2 + \frac{2\eta_g^2 \eta_l^2}{N^2} \mathbb{E} \left\| \sum_{i \in \mathcal{A}_t} \sum_{k=0}^{K-1} \nabla f_i(x_i^{(a_{i,t},k)}) \right\|^2. \tag{43}$$

We get the desired result by summing over $t$ from $0$ to $T-1$ on the both sides of Equation (43). $\square$

As a direct consequence of Lemma B.2, we conduct the Lemma D.2 and Corollary D.3 to bound the aggregated gradient of server.

**Lemma D.2.** *Suppose Assumption 1.1 and 1.2 hold. Then, we have*

$$\sum_{t=0}^{T-1} \sum_{p=t-\tau_t}^{t-1} \mathbb{E} \left\| \sum_{j \in \mathcal{A}_p} \sum_{k=0}^{K-1} \nabla f_j(x_j^{(a_{j,p},k)}) \right\|^2 \leq \tau_{\max} \sum_{t=0}^{T-1} \mathbb{E} \left\| \sum_{i \in \mathcal{A}_t} \sum_{k=0}^{K-1} \nabla f_i(x_i^{(a_{i,t},k)}) \right\|^2.$$

*Proof.* Invoking Lemma B.2, we have

$$t - \tau_t = t - \max_{i \in \mathcal{N}} \{ t - a_{i,t} \} = \min_{i \in \mathcal{N}} \{ a_{i,t} \} \geq \min \{ 0, t - \tau_{\max} \}.$$

And hence,

$$\sum_{t=0}^{T-1} \sum_{p=t-\tau_t}^{t-1} \mathbb{E} \left\| \sum_{j \in \mathcal{A}_p} \sum_{k=0}^{K-1} \nabla f_j(x_j^{(a_{j,p},k)}) \right\|^2 \leq \sum_{t=0}^{T-1} \sum_{p=\min\{0,t-\tau_{\max}\}}^{t-1} \mathbb{E} \left\| \sum_{j \in \mathcal{A}_p} \sum_{k=0}^{K-1} \nabla f_j(x_j^{(a_{j,p},k)}) \right\|^2$$

$$\leq \sum_{p=0}^{T-1} \sum_{t=p+1}^{p+\tau_{\max}} \mathbb{E} \left\| \sum_{j \in \mathcal{A}_p} \sum_{k=0}^{K-1} \nabla f_j(x_j^{(a_{j,p},k)}) \right\|^2 = \tau_{\max} \sum_{t=0}^{T-1} \mathbb{E} \left\| \sum_{i \in \mathcal{A}_t} \sum_{k=0}^{K-1} \nabla f_i(x_i^{(a_{i,t},k)}) \right\|^2,$$

which gives the desired result. $\square$

Corollary D.3 is a direct consequence of Lemma D.2.

**Corollary D.3.** *Suppose Assumption 1.1 and 1.2 hold. Then, we have*

$$\sum_{t=0}^{T-1} \tau_t \sum_{p=t-\tau_t}^{t-1} \mathbb{E} \left\| \sum_{j \in \mathcal{A}_p} \sum_{k=0}^{K-1} \nabla f_j(x_j^{(a_{j,p},k)\,\prime}) \right\|^2 \leq \tau_{\max}^2 \sum_{t=0}^{T-1} \mathbb{E} \left\| \sum_{i \in \mathcal{A}_t} \sum_{k=0}^{K-1} \nabla f_i(x_i^{(a_{i,t},k)}) \right\|^2.$$

Lemma D.4 is the key lemma to estimate the difference between the aggregated gradient of server and the aggregated gradient of all clients.

**Lemma D.4.** *Suppose Assumption 1.1 and 1.2 hold. Then, we have, for $\eta_l \leq \frac{1}{10\sqrt{\tau_{\max}}KL}$,*

$$
\sum_{t=0}^{T-1} \mathbb{E} \left\| K \sum_{i=1}^{N} \nabla f_i(x^{(t)}) - \sum_{i \in \mathcal{A}_t} \sum_{k=0}^{K-1} \nabla f_i(x_i^{(a_{i,t},k)}) \right\|^2
$$

$$
\leq 3K^2 N^2 \tau_{\max} F_0 + 8\eta_g^2 \eta_l^2 K^3 N \tau_{\max} \max\{1, \tau_{avg}\} T\sigma^2 L^2 + 150\eta_l^2 N^3 K^3 T\sigma^2 L^2
$$

$$
+ \left(10\eta_g^2 \eta_l^2 \tau_{\max}^2 K^2 L^2 + 24\eta_l^2 K^2 N^2 \tau_{\max} L^2\right) \sum_{t=0}^{T-1} \mathbb{E} \left\| \sum_{j \in \mathcal{A}_t} \sum_{k=0}^{K-1} \nabla f_j(x_j^{(a_{j,t},k)}) \right\|^2,
$$

*where $F_0 := \frac{1}{N} \sum_{i=1}^{N} \left\| \nabla f_i(x^{(0)}) \right\|^2$.*

*Proof.* Notice that, by the decomposition of the difference between the aggregated gradient of server and the aggregated gradient of all clients, we have

$$
\left\| K \sum_{i=1}^{N} \nabla f_i(x^{(t)}) - \sum_{i \in \mathcal{A}_t} \sum_{k=0}^{K-1} \nabla f_i(x_i^{(a_{i,t},k)}) \right\|^2 \leq 3K^2 \left\| \sum_{i \in \mathcal{N}} \nabla f_i(x^{(t)}) - \sum_{i \in \mathcal{N}} \nabla f_i(x^{(a_{i,t})}) \right\|^2
$$

$$
+ 3 \left\| K \sum_{i \in \mathcal{A}_t} \nabla f_i(x^{(a_{i,t})}) - \sum_{i \in \mathcal{A}_t} \sum_{k=0}^{K-1} \nabla f_i(x_i^{(a_{i,t},k)}) \right\|^2 + 3K^2 \left\| \sum_{i \in \mathcal{N}/\mathcal{A}_t} \nabla f_i(x^{(0)}) \right\|^2.
$$

(44)

We bound the expectation of terms in Equation (44) one by one. First of all, we have

$$
\mathbb{E} \left\| \sum_{i \in \mathcal{N}} \nabla f_i(x^{(t)}) - \sum_{i \in \mathcal{N}} \nabla f_i(x^{(a_{i,t})}) \right\|^2 \leq NL^2 \sum_{i \in \mathcal{N}} \mathbb{E} \left\| x^{(t)} - x^{(a_{i,t})} \right\|^2
$$

$$
= NL^2 \sum_{i \in \mathcal{N}} \mathbb{E} \left\| \sum_{p=a_{i,t}}^{t-1} \left[ x^{(p+1)} - x^{(p)} \right] \right\|^2 = \frac{\eta_g^2 \eta_l^2}{N} L^2 \sum_{i \in \mathcal{N}} \mathbb{E} \left\| \sum_{p=a_{i,t}}^{t-1} \sum_{j \in \mathcal{A}_p} \sum_{k=0}^{K-1} g_j(x_j^{(a_{j,p},k)}, \xi_j^{(a_{j,p},k)}) \right\|^2.
$$

(45)

By Assumption 1.2, we have, according to Equation (45) and summing over $t$,

$$
3K^2 \sum_{t=0}^{T-1} \mathbb{E} \left\| \sum_{i \in \mathcal{N}} \nabla f_i(x^{(t)}) - \sum_{i \in \mathcal{N}} \nabla f_i(x^{(a_{i,t})}) \right\|^2
$$

$$
\leq 6K^2 \frac{\eta_g^2 \eta_l^2}{N} L^2 \sum_{t=0}^{T-1} \sum_{i \in \mathcal{N}} \tau_t \sum_{p=a_{i,t}}^{t-1} \sum_{j \in \mathcal{A}_p} \sum_{k=0}^{K-1} \sigma^2 + 6K^2 \frac{\eta_g^2 \eta_l^2}{N} L^2 \sum_{t=0}^{T-1} \sum_{i \in \mathcal{N}} \mathbb{E} \left\| \sum_{p=a_{i,t}}^{t-1} \sum_{j \in \mathcal{A}_p} \sum_{k=0}^{K-1} \nabla f_j(x_j^{(a_{j,p},k)}) \right\|^2
$$

$$
\leq 6\eta_g^2 \eta_l^2 K^3 N \tau_{\max} \tau_{avg} T\sigma^2 L^2 + 6\eta_g^2 \eta_l^2 K^2 \tau_{\max}^2 L^2 \sum_{t=0}^{T-1} \mathbb{E} \left\| \sum_{i \in \mathcal{A}_t} \sum_{k=0}^{K-1} \nabla f_i(x_i^{(a_{i,t},k)}) \right\|^2,
$$

(46)

where the procedure is similar to the proof of Lemma C.4.

For the second part of Equation (44), we have, by Assumption 1.1,

$$
\left\| K \sum_{i \in \mathcal{A}_t} \nabla f_i(x^{(a_{i,t})}) - \sum_{i \in \mathcal{A}_t} \sum_{k=0}^{K-1} \nabla f_i(x_i^{(a_{i,t},k)}) \right\|^2 = \left\| \sum_{i \in \mathcal{A}_t} \sum_{k=0}^{K-1} \left[ \nabla f_i(x^{(a_{i,t})}) - \nabla f_i(x_i^{(a_{i,t},k)}) \right] \right\|^2
$$

$$
\leq NKL^2 \sum_{i \in \mathcal{A}_t} \sum_{k=0}^{K-1} \left\| x^{(a_{i,t})} - x_i^{(a_{i,t},k)} \right\|^2.
$$

(47)

Note that $x^{(a_{i,t})}$ is the local model of client $i$ at the beginning of round $a_{i,t}$, and $x_i^{(a_{i,t},k)}$ is the local model of client $i$ at the beginning of round $a_{i,t}$ after $k$ local updates. Then, according to the

definition of $h_i^{(t)}$ and $y_i^{(t)}$ shown in Equation 40 and 41, it holds that

$$\sum_{i \in \mathcal{A}_t} \sum_{k=0}^{K-1} \left\| x^{(a_{i,t})} - x_i^{(a_{i,t},k)} \right\|^2 = \sum_{i \in \mathcal{A}_t} \sum_{k=0}^{K-1} \left\| \sum_{p=0}^{k-1} \left[ x_i^{(a_{i,t},p)} - x_i^{(a_{i,t},p+1)} \right] \right\|^2$$

$$= \eta_l^2 \sum_{i \in \mathcal{A}_t} \sum_{k=0}^{K-1} \left\| \sum_{p=0}^{k-1} \left[ g_i(x_i^{(a_{i,t},p)}, \xi_i^{(a_{i,t},p)}) + y_i^{(a_{i,t})} \right] \right\|^2 \tag{48}$$

$$\leq 2\eta_l^2 \sum_{i \in \mathcal{A}_t} \sum_{k=0}^{K-1} \left\| \sum_{p=0}^{k-1} \left[ g_i(x_i^{(a_{i,t},p)}, \xi_i^{(a_{i,t},p)}) - h_i^{(a_{i,t})} \right] \right\|^2 + 2\eta_l^2 \sum_{i \in \mathcal{A}_t} \sum_{k=0}^{K-1} \left\| \sum_{p=0}^{k-1} y^{(a_{i,t-1})} \right\|^2.$$

We deal with the first term in Equation (48) as follows. Notice that, by the definition of $h_i^{(t)}$ in Equation (40) that

$$h_i^{(a_{i,t})} = \frac{1}{K} \sum_{k=0}^{K-1} g_i(x_i^{(a_{i,a_{i,t}-1},k)}, \xi_i^{(a_{i,a_{i,t}-1},k)}),$$

where $a_{i,a_{i,t}-1}$ is the round when client $i$ is selected in the second last round before $t$. Then, we have

$$\left\| \sum_{p=0}^{k-1} \left[ g_i(x_i^{(a_{i,t},p)}, \xi_i^{(a_{i,t},p)}) - h_i^{(a_{i,t})} \right] \right\|^2$$

$$\leq 5 \left\| \sum_{p=0}^{k-1} \left[ g_i(x_i^{(a_{i,t},p)}, \xi_i^{(a_{i,t},p)}) - \nabla f_i(x_i^{(a_{i,t},p)}) \right] \right\|^2 + 5 \left\| \sum_{p=0}^{k-1} \left[ \nabla f_i(x_i^{(a_{i,t},p)}) - \nabla f_i(x^{(a_{i,t})}) \right] \right\|^2$$

$$+ 5k^2 \left\| \nabla f_i(x^{(a_{i,t})}) - \nabla f_i(x^{(a_{i,a_{i,t}-1})}) \right\|^2 + 5\frac{k^2}{K^2} \left\| \sum_{q=0}^{K-1} \left[ \nabla f_i(x^{(a_{i,a_{i,t}-1})}) - \nabla f_i(x_i^{(a_{i,a_{i,t}-1},q)}) \right] \right\|^2$$

$$+ 5\frac{k^2}{K^2} \left\| \sum_{q=0}^{K-1} \left[ g_i(x_i^{(a_{i,a_{i,t}-1},q)}, \xi_i^{(a_{i,a_{i,t}-1},q)}) - \nabla f_i(x^{(a_{i,a_{i,t}-1},q)}) \right] \right\|^2.$$

$$\tag{49}$$

Then, by taking expectation and invoking Equation (48), we have the following upper bound for each term in Equation (49).

**Term 1:**

$$\eta_l^2 \sum_{t=0}^{T-1} \sum_{i \in \mathcal{A}_t} \sum_{k=0}^{K-1} \mathbb{E} \left\| \sum_{p=0}^{k-1} \left[ g_i(x_i^{(a_{i,t},p)}, \xi_i^{(a_{i,t},p)}) - \nabla f_i(x_i^{(a_{i,t},p)}) \right] \right\|^2$$

$$\leq \eta_l^2 \sum_{t=0}^{T-1} \sum_{i \in \mathcal{A}_t} \sum_{k=0}^{K-1} \sum_{p=0}^{k-1} \sigma^2 \leq \eta_l^2 N K^2 T \sigma^2. \tag{50}$$

**Term 2:**

$$\eta_l^2 \sum_{t=0}^{T-1} \sum_{i \in \mathcal{A}_t} \sum_{k=0}^{K-1} \mathbb{E} \left\| \sum_{p=0}^{k-1} \left[ \nabla f_i(x_i^{(a_{i,t},p)}) - \nabla f_i(x^{(a_{i,t})}) \right] \right\|^2 \leq \eta_l^2 L^2 \sum_{t=0}^{T-1} \sum_{i \in \mathcal{A}_t} \sum_{k=0}^{K-1} k \sum_{p=0}^{k-1} \mathbb{E} \left\| x_i^{(a_{i,t},p)} - x^{(a_{i,t})} \right\|^2$$

$$\leq \eta_l^2 L^2 \sum_{t=0}^{T-1} \sum_{i \in \mathcal{A}_t} \sum_{k=0}^{K-1} k \sum_{p=0}^{K-1} \mathbb{E} \left\| x_i^{(a_{i,t},p)} - x^{(a_{i,t})} \right\|^2 \leq \eta_l^2 K^2 L^2 \sum_{t=0}^{T-1} \sum_{i \in \mathcal{A}_t} \sum_{k=0}^{K-1} \mathbb{E} \left\| x_i^{(a_{i,t},k)} - x^{(a_{i,t})} \right\|^2. \tag{51}$$

**Term 3:**

$$\eta_l^2 \sum_{t=0}^{T-1} \sum_{i \in \mathcal{A}_t} \sum_{k=0}^{K-1} k^2 \left\| \nabla f_i(x^{(a_{i,t})}) - \nabla f_i(x^{(a_{i,a_{i,t}-1})}) \right\|^2$$

$$\leq \eta_l^2 K^3 L^2 \sum_{t=0}^{T-1} \sum_{i \in \mathcal{A}_t} \mathbb{E} \left\| x^{(a_{i,t})} - x^{(a_{i,a_{i,t}-1})} \right\|^2 = \eta_l^2 K^3 L^2 \sum_{t=0}^{T-1} \sum_{i \in \mathcal{A}_t} \mathbb{E} \left\| \sum_{p=a_{i,a_{i,t}-1}}^{a_{i,t}-1} \left[ x^{(p+1)} - x^{(p)} \right] \right\|^2$$

$$\leq \frac{2\eta_g^2 \eta_l^4 K^3 L^2}{N^2} \sum_{t=0}^{T-1} \sum_{i \in \mathcal{A}_t} \mathbb{E} \left\| \sum_{p=a_{i,a_{i,t}-1}}^{a_{i,t}-1} \sum_{j \in \mathcal{A}_p} \sum_{k=0}^{K-1} \nabla f_j(x_j^{(a_{j,p},k)}) \right\|^2$$

$$+ \frac{2\eta_g^2 \eta_l^4 K^3 L^2}{N^2} \sum_{t=0}^{T-1} \sum_{i \in \mathcal{A}_t} \left[ a_{i,t} - a_{i,a_{i,t}-1} \right] \sum_{p=a_{i,a_{i,t}-1}}^{a_{i,t}-1} \sum_{j \in \mathcal{A}_p} \sum_{k=0}^{K-1} \sigma^2$$

$$\leq \frac{2\eta_g^2 \eta_l^4 K^3 L^2}{N^2} \sum_{t=0}^{T-1} \sum_{i \in \mathcal{A}_t} \left[ a_{i,t} - a_{i,a_{i,t}-1} \right] \sum_{p=a_{i,a_{i,t}-1}}^{a_{i,t}-1} \mathbb{E} \left\| \sum_{j \in \mathcal{A}_p} \sum_{k=0}^{K-1} \nabla f_j(x_j^{(a_{j,p},k)}) \right\|^2$$

$$+ \frac{2\eta_g^2 \eta_l^4 K^3 L^2}{N^2} \sum_{t=0}^{T-1} \sum_{i \in \mathcal{A}_t} \left[ a_{i,t} - a_{i,a_{i,t}-1} \right]^2 NK\sigma^2.$$

(52)

With the fact that $a_{i,a_{i,t}-1}$ is the second last round when client $i$ is selected before $t$ and $a_{i,t}$ is the last round when client $i$ is selected, it holds that $0 \leq a_{i,t} - a_{i,a_{i,t}-1} \leq \tau_{\max}$. Then,

$$\eta_l^2 \sum_{t=0}^{T-1} \sum_{i \in \mathcal{A}_t} \sum_{k=0}^{K-1} k^2 \left\| \nabla f_i(x^{(a_{i,t})}) - \nabla f_i(x^{(a_{i,a_{i,t}-1})}) \right\|^2$$

$$\leq \frac{2\eta_g^2 \eta_l^4 K^3 L^2}{N^2} \sum_{t=0}^{T-1} \sum_{i \in \mathcal{A}_t} \tau_{\max} \sum_{p=\max\{0, a_{i,t}-\tau_{\max}\}}^{a_{i,t}} \mathbb{E} \left\| \sum_{j \in \mathcal{A}_p} \sum_{k=0}^{K-1} \nabla f_j(x_j^{(a_{j,p},k)}) \right\|^2 + 2\eta_g^2 \eta_l^4 \tau_{\max}^2 K^4 T \sigma^2 L^2.$$

(53)

With the fact that $t - \tau_{\max} \leq a_{i,t} \leq t$, we have

$$\frac{2\eta_g^2 \eta_l^4 K^3 L^2}{N^2} \sum_{t=0}^{T-1} \sum_{i \in \mathcal{A}_t} \tau_{\max} \sum_{p=\max\{0, a_{i,t}-\tau_{\max}\}}^{a_{i,t}} \mathbb{E} \left\| \sum_{j \in \mathcal{A}_p} \sum_{k=0}^{K-1} \nabla f_j(x_j^{(a_{j,p},k)}) \right\|^2$$

$$\leq \frac{2\eta_g^2 \eta_l^4 \tau_{\max} K^3 L^2}{N^2} \sum_{t=0}^{T-1} \sum_{i \in \mathcal{A}_t} \sum_{p=\max\{0, t-2\tau_{\max}\}}^{t} \mathbb{E} \left\| \sum_{j \in \mathcal{A}_p} \sum_{k=0}^{K-1} \nabla f_j(x_j^{(a_{j,p},k)}) \right\|^2$$

$$\leq \frac{2\eta_g^2 \eta_l^4 \tau_{\max} K^3 L^2}{N} \sum_{t=0}^{T-1} \sum_{p=\max\{0, t-2\tau_{\max}\}}^{t} \mathbb{E} \left\| \sum_{j \in \mathcal{A}_p} \sum_{k=0}^{K-1} \nabla f_j(x_j^{(a_{j,p},k)}) \right\|^2 \qquad (54)$$

$$\leq \frac{2\eta_g^2 \eta_l^4 \tau_{\max} K^3 L^2}{N} \sum_{p=0}^{T-1} \sum_{t=p}^{\min\{T-1, p+2\tau_{\max}\}} \mathbb{E} \left\| \sum_{j \in \mathcal{A}_p} \sum_{k=0}^{K-1} \nabla f_j(x_j^{(a_{j,p},k)}) \right\|^2$$

$$\leq \frac{4\eta_g^2 \eta_l^4 \tau_{\max}^2 K^3 L^2}{N} \sum_{t=0}^{T-1} \mathbb{E} \left\| \sum_{j \in \mathcal{A}_t} \sum_{k=0}^{K-1} \nabla f_j(x_j^{(a_{j,t},k)}) \right\|^2.$$

**Term 4:**

$$\eta_l^2 \sum_{t=0}^{T-1} \sum_{i \in \mathcal{A}_t} \sum_{k=0}^{K-1} \frac{k^2}{K^2} \mathbb{E} \left\| \sum_{q=0}^{K-1} \left[ \nabla f_i(x^{(a_{i,a_{i,t}-1})}) - \nabla f_i(x_i^{(a_{i,a_{i,t}-1},q)}) \right] \right\|^2$$

$$\leq \eta_l^2 L^2 \sum_{t=0}^{T-1} \sum_{i \in \mathcal{A}_t} \sum_{k=0}^{K-1} \frac{k^2}{K} \sum_{q=0}^{K-1} \mathbb{E} \left\| x^{(a_{i,a_{i,t}-1})} - x_i^{(a_{i,a_{i,t}-1},q)} \right\|^2 \tag{55}$$

$$\leq \eta_l^2 \tau_{\max} K^2 L^2 \sum_{t=0}^{T-1} \sum_{i \in \mathcal{A}_t} \sum_{k=0}^{K-1} \mathbb{E} \left\| x_i^{(a_{i,t},k)} - x^{(a_{i,t})} \right\|^2,$$

where the last inequality is due to the fact that $a_{i,a_{i,t}-1}$ is the second last round when client $i$ is selected before $t$ and $a_{i,t}$ is the last round when client $i$ is selected. Then, it holds that $a_{i,a_{i,t}+1} = a_{i,t}$. Hence, $\sum_{t=0}^{T-1} \sum_{i \in \mathcal{A}_t} \sum_{q=0}^{K-1} \mathbb{E} \left\| x^{(a_{i,a_{i,t}-1})} - x_i^{(a_{i,a_{i,t}-1},q)} \right\|^2 \leq \tau_{\max} \sum_{t=0}^{T-1} \sum_{i \in \mathcal{A}_t} \sum_{k=0}^{K-1} \mathbb{E} \left\| x_i^{(a_{i,t},k)} - x^{(a_{i,t})} \right\|^2$.

**Term 5:**

$$\eta_l^2 \sum_{t=0}^{T-1} \sum_{i \in \mathcal{A}_t} \sum_{k=0}^{K-1} \frac{k^2}{K^2} \mathbb{E} \left\| \sum_{q=0}^{K-1} \left[ g_i(x_i^{(a_{i,a_{i,t}-1},q)}, \xi_i^{(a_{i,a_{i,t}-1},q)}) - \nabla f_i(x^{(a_{i,a_{i,t}-1},q)}) \right] \right\|^2$$

$$\leq \eta_l^2 \sum_{t=0}^{T-1} \sum_{i \in \mathcal{A}_t} \sum_{k=0}^{K-1} \frac{k^2}{K^2} \sum_{q=0}^{K-1} \sigma^2 \leq \eta_l^2 N K^2 T \sigma^2. \tag{56}$$

For the second term in Equation (48), we have

$$2\eta_l^2 \sum_{i \in \mathcal{A}_t} \sum_{k=0}^{K-1} \left\| \sum_{p=0}^{k-1} y^{(a_{i,t}-1)} \right\|^2 = 2\eta_l^2 \sum_{i \in \mathcal{A}_t} \sum_{k=0}^{K-1} k^2 \left\| y^{(a_{i,t}-1)} \right\|^2 \leq 2\eta_l^2 K^3 \sum_{i \in \mathcal{A}_t} \left\| y^{(a_{i,t}-1)} \right\|^2. \tag{57}$$

Consequently, it implies that

$$2\eta_l^2 K^3 \sum_{t=0}^{T-1} \sum_{i \in \mathcal{A}_t} \mathbb{E} \left\| y^{(a_{i,t}-1)} \right\|^2 = 2\eta_l^2 K \sum_{t=0}^{T-1} \sum_{i \in \mathcal{A}_t} \mathbb{E} \left\| \sum_{j \in \mathcal{A}_{a_{i,t}-1}} \sum_{k=0}^{K-1} g_j(x_j^{(a_{j,a_{i,t}},k)}, \xi_j^{(a_{j,a_{i,t}},k)}) \right\|^2$$

$$\leq 4\eta_l^2 K \sum_{t=0}^{T-1} \sum_{i \in \mathcal{A}_t} \mathbb{E} \left\| \sum_{j \in \mathcal{A}_{a_{i,t}-1}} \sum_{k=0}^{K-1} \nabla f_j(x_j^{(a_{j,a_{i,t}},k)}) \right\|^2 + 4\eta_l^2 K^2 N^2 \sigma^2 T$$

$$\leq 4\eta_l^2 K N \tau_{\max} \sum_{t=0}^{T-1} \mathbb{E} \left\| \sum_{i \in \mathcal{A}_t} \sum_{k=0}^{K-1} \nabla f_i(x_i^{(a_{i,t},k)}) \right\|^2 + 4\eta_l^2 K^2 N^2 \sigma^2 T, \tag{58}$$

where the last inequality is due to the fact that $\mathbb{E} \left\| \sum_{i \in \mathcal{A}_t} \sum_{k=0}^{K-1} \nabla f_i(x_i^{(a_{i,t},k)}) \right\|^2$ appears at most $N\tau_{\max}$ times in the previous sum.

Finally, after summing $t$ from 0 to $T-1$ in Equation (48), and implementing the upper bound for each term in Equation (49) as above, it holds that

$$\sum_{t=0}^{T-1} \sum_{i\in\mathcal{A}_t} \sum_{k=0}^{K-1} \mathbb{E}\left\|x^{(a_{i,t})} - x_i^{(a_{i,t},k)}\right\|^2 \le 10\eta_l^2 NK^2T\sigma^2 + 10\eta_l^2 K^2 L^2 \sum_{t=0}^{T-1} \sum_{i\in\mathcal{A}_t} \sum_{k=0}^{K-1} \mathbb{E}\left\|x_i^{(a_{i,t},k)} - x^{(a_{i,t})}\right\|^2$$

$$+ \frac{40\eta_g^2\eta_l^4\tau_{\max}^2 K^3 L^2}{N} \sum_{t=0}^{T-1} \mathbb{E}\left\|\sum_{j\in\mathcal{A}_t} \sum_{k=0}^{K-1} \nabla f_j(x_j^{(a_{j,t},k)})\right\|^2 + 20\eta_g^2\eta_l^4\tau_{\max}^2 K^4 T\sigma^2 L^2$$

$$+ 10\eta_l^2\tau_{\max}K^2 L^2 \sum_{t=0}^{T-1} \sum_{i\in\mathcal{A}_t} \sum_{k=0}^{K-1} \mathbb{E}\left\|x_i^{(a_{i,t},k)} - x^{(a_{i,t})}\right\|^2 + 10\eta_l^2 NK^2 T\sigma^2$$

$$+ 4\eta_l^2 KN\tau_{\max} \sum_{t=0}^{T-1} \mathbb{E}\left\|\sum_{i\in\mathcal{A}_t} \sum_{k=0}^{K-1} \nabla f_i(x_i^{(a_{i,t},k)})\right\|^2 + 4\eta_l^2 K^2 N^2\sigma^2 T.$$

$$(59)$$

Then, for $\eta_l \le \frac{1}{10\sqrt{\tau_{\max}}KL}$, by rearranging the terms in Equation (59), we have

$$\sum_{t=0}^{T-1} \sum_{i\in\mathcal{A}_t} \sum_{k=0}^{K-1} \mathbb{E}\left\|x^{(a_{i,t})} - x_i^{(a_{i,t},k)}\right\|^2 \le 50\eta_l^2 N^2 K^2 T\sigma^2 + 40\eta_g^2\eta_l^4\tau_{\max}^2 K^4 T\sigma^2 L^2$$

$$+ \left(\frac{80\eta_g^2\eta_l^4\tau_{\max}^2 K^3 L^2}{N} + 8\eta_l^2 KN\tau_{\max}\right) \sum_{t=0}^{T-1} \mathbb{E}\left\|\sum_{j\in\mathcal{A}_t} \sum_{k=0}^{K-1} \nabla f_j(x_j^{(a_{j,t},k)})\right\|^2.$$

$$(60)$$

For the third part of Equation (44), we have, by summing over $t$,

$$3K^2 \sum_{t=0}^{T-1} \left\|\sum_{i\in\mathcal{N}/\mathcal{A}_t} \nabla f_i(x^{(0)})\right\|^2 \le 3K^2 \sum_{t=0}^{T-1} (N - A_t) \sum_{i=1}^{N} \left\|\nabla f_i(x^{(0)})\right\|^2$$

$$= 3K^2 \sum_{t=0}^{T-1} \sum_{j\in\mathcal{N}} \mathbb{1}_{j\notin\mathcal{A}_t} \sum_{i=1}^{N} \left\|\nabla f_i(x^{(0)})\right\|^2 \le 3K^2 N^2\tau_{\max}F_0,$$

$$(61)$$

where $F_0 := \frac{1}{N} \sum_{i=1}^{N} \left\|\nabla f_i(x^{(0)})\right\|^2$.

Thus, by summing over $t$ and taking expectation in Equation (44), implementing the three parts, we have, for $\eta_l \le 1/\left(10\sqrt{\tau_{\max}}KL\right)$, by $3NKL^2 \cdot 40\eta_g^2\eta_l^4\tau_{\max}^2 K^4 T\sigma^2 L^4 \le 2\eta_g^2\eta_l^2 K^3 N\tau_{\max}T\sigma^2 L^2$,

$$\sum_{t=0}^{T-1} \mathbb{E}\left\|K\sum_{i=1}^{N} \nabla f_i(x^{(t)}) - \sum_{i\in\mathcal{A}_t} \sum_{k=0}^{K-1} \nabla f_i(x_i^{(a_{i,t},k)})\right\|^2$$

$$\le 3K^2 N^2\tau_{\max}F_0 + 8\eta_g^2\eta_l^2 K^3 N\tau_{\max}\max\{1, \tau_{\mathrm{avg}}\}T\sigma^2 L^2 + 6\eta_g^2\eta_l^2 K^2\tau_{\max}^2 L^2 \sum_{t=0}^{T-1} \mathbb{E}\left\|\sum_{i\in\mathcal{A}_t} \sum_{k=0}^{K-1} \nabla f_i(x_i^{(a_{i,t},k)})\right\|^2$$

$$+ 150\eta_l^2 N^3 K^3 T\sigma^2 L^2 + \left(240\eta_g^2\eta_l^4\tau_{\max}^2 K^4 L^4 + 24\eta_l^2 K^2 N^2\tau_{\max}L^2\right) \sum_{t=0}^{T-1} \mathbb{E}\left\|\sum_{j\in\mathcal{A}_t} \sum_{k=0}^{K-1} \nabla f_j(x_j^{(a_{j,t},k)})\right\|^2$$

$$\le 3K^2 N^2\tau_{\max}F_0 + 8\eta_g^2\eta_l^2 K^3 N\tau_{\max}\max\{1, \tau_{\mathrm{avg}}\}T\sigma^2 L^2 + 150\eta_l^2 N^3 K^3 T\sigma^2 L^2$$

$$+ \left(10\eta_g^2\eta_l^2\tau_{\max}^2 K^2 L^2 + 24\eta_l^2 K^2 N^2\tau_{\max}L^2\right) \sum_{t=0}^{T-1} \mathbb{E}\left\|\sum_{j\in\mathcal{A}_t} \sum_{k=0}^{K-1} \nabla f_j(x_j^{(a_{j,t},k)})\right\|^2.$$

$$(62)$$

$$\square$$

**Lemma D.5.** *Suppose Assumption 1.1 and 1.2 hold. Then, we have*

$$-\frac{\eta_g \eta_l}{N} \sum_{t=0}^{T-1} \mathbb{E} \left\langle \nabla f(x^{(t)}), \sum_{i \in \mathcal{A}_t} \sum_{k=0}^{K-1} \left[ g_i(x_i^{(a_{i,t},k)}, \xi_i^{(a_{i,t},k)}) - \nabla f_i(x_i^{(a_{i,t},k)}) \right] \right\rangle$$

$$\leq 3\frac{\eta_g^2 \eta_l^2 K}{N} \tau_{avg} \sigma^2 LT + 2\frac{\eta_g^2 \eta_l^2}{N^2} \tau_{\max} L \sum_{t=0}^{T} \mathbb{E} \left\| \sum_{i \in \mathcal{A}_t} \sum_{k=0}^{K-1} \nabla f_i(x_i^{(a_{i,t},k)}) \right\|^2.$$

*Proof.* By the independently and randomly sampled $\xi_i^{(a_{i,t})}$, we have, for each $i \in \mathcal{A}_t$ and $0 \leq k \leq K-1$,

$$\mathbb{E} \left\langle \nabla f(x^{(\min_{i \in \mathcal{N}}\{a_{i,t}\})}), g_i(x_i^{(a_{i,t},k)}, \xi_i^{(a_{i,t},k)}) - \nabla f_i(x_i^{(a_{i,t},k)}) \right\rangle = 0. \tag{63}$$

By $t - \tau_t = \min_{i \in \mathcal{N}}\{a_{i,t}\}$, we then derive that

$$-\frac{\eta_g \eta_l}{N} \mathbb{E} \left\langle \nabla f(x^{(t)}), \sum_{i \in \mathcal{A}_t} \sum_{k=0}^{K-1} \left[ g_i(x_i^{(a_{i,t},k)}, \xi_i^{(a_{i,t},k)}) - \nabla f_i(x_i^{(a_{i,t},k)}) \right] \right\rangle$$

$$= -\frac{\eta_g \eta_l}{N} \mathbb{E} \left\langle \nabla f(x^{(t)}) - \nabla f(x^{(t-\tau_t)}), \sum_{i \in \mathcal{A}_t} \sum_{k=0}^{K-1} \left[ g_i(x_i^{(a_{i,t},k)}, \xi_i^{(a_{i,t},k)}) - \nabla f_i(x_i^{(a_{i,t},k)}) \right] \right\rangle. \tag{64}$$

Then, if $\tau_t > 0$, it holds by picking $\alpha = \frac{1}{\tau_t L}$ in the Cauchy-Schwarz inequality $\langle a, b \rangle \leq \alpha \|a\|^2 + \frac{1}{\alpha} \|b\|^2$ that,

$$-\frac{\eta_g \eta_l}{N} \mathbb{E} \left\langle \nabla f(x^{(t)}) - \nabla f(x^{(t-\tau_t)}), \sum_{i \in \mathcal{A}_t} \sum_{k=0}^{K-1} \left[ g_i(x_i^{(a_{i,t},k)}, \xi_i^{(a_{i,t},k)}) - \nabla f_i(x_i^{(a_{i,t},k)}) \right] \right\rangle$$

$$\leq \frac{1}{\tau_t L} \mathbb{E} \left\| \nabla f(x^{(t)}) - \nabla f(x^{(t-\tau_t)}) \right\|^2 + \frac{\eta_g^2 \eta_l^2}{N^2} \tau_t L \mathbb{E} \left\| \sum_{i \in \mathcal{A}_t} \sum_{k=0}^{K-1} \left[ g_i(x_i^{(a_{i,t},k)}, \xi_i^{(a_{i,t},k)}) - \nabla f_i(x_i^{(a_{i,t},k)}) \right] \right\|^2. \tag{65}$$

With the fact that, by taking full expectation, we derive that

$$\mathbb{E} \left\| \sum_{i \in \mathcal{A}_t} \sum_{k=0}^{K-1} \left[ g_i(x_i^{(a_{i,t},k)}, \xi_i^{(a_{i,t},k)}) - \nabla f_i(x_i^{(a_{i,t},k)}) \right] \right\|^2$$

$$\leq \sum_{i \in \mathcal{A}_t} \sum_{k=0}^{K-1} \mathbb{E} \left\| g_i(x_i^{(a_{i,t},k)}, \xi_i^{(a_{i,t},k)}) - \nabla f_i(x_i^{(a_{i,t},k)}) \right\|^2 \leq NK\sigma^2,$$

and

$$\mathbb{E} \left\| \nabla f(x^{(t)}) - \nabla f(x^{(t-\tau_t)}) \right\|^2 \leq L^2 \mathbb{E} \left\| x^{(t)} - x^{(t-\tau_t)} \right\|^2 = \frac{\eta_g^2 \eta_l^2}{N^2} L^2 \mathbb{E} \left\| \sum_{p=t-\tau_t}^{t-1} \sum_{j \in \mathcal{A}_p} \sum_{k=0}^{K-1} g_j(x_j^{(a_{j,p},k)}, \xi_j^{(a_{j,p},k)}) \right\|^2$$

$$\leq 2\frac{\eta_g^2 \eta_l^2}{N^2} L^2 \mathbb{E} \left\| \sum_{p=t-\tau_t}^{t-1} \sum_{j \in \mathcal{A}_p} \sum_{k=0}^{K-1} \left[ g_j(x_j^{(a_{j,p},k)}, \xi_j^{(a_{j,p},k)}) - \nabla f_j(x_j^{(a_{j,p},k)}) \right] \right\|^2$$

$$+ 2\frac{\eta_g^2 \eta_l^2}{N^2} L^2 \mathbb{E} \left\| \sum_{p=t-\tau_t}^{t-1} \sum_{j \in \mathcal{A}_p} \sum_{k=0}^{K-1} \nabla f_j(x_j^{(a_{j,p},k)}) \right\|^2.$$

Then, we have

$$
\mathbb{E} \left\| \sum_{p=t-\tau_t}^{t-1} \sum_{j \in \mathcal{A}_p} \sum_{k=0}^{K-1} \left[ g_j(x_j^{(a_{j,p},k)}, \xi_j^{(a_{j,p},k)}) - \nabla f_j(x_j^{(a_{j,p},k)}) \right] \right\|^2
$$

$$
= \tau_t \sum_{p=t-\tau_t}^{t-1} \sum_{j \in \mathcal{A}_p} \sum_{k=0}^{K-1} \mathbb{E} \left\| g_j(x_j^{(a_{j,p},k)}, \xi_j^{(a_{j,p},k)}) - \nabla f_j(x_j^{(a_{j,p},k)}) \right\|^2
$$

$$
\leq \tau_t \sum_{p=t-\tau_t}^{t-1} NK\sigma^2 = NK\tau_t^2\sigma^2,
$$

and

$$
\mathbb{E} \left\| \sum_{p=t-\tau_t}^{t-1} \sum_{j \in \mathcal{A}_p} \sum_{k=0}^{K-1} \nabla f_j(x^{(a_{j,p},k)}) \right\|^2 \leq \tau_t \sum_{p=t-\tau_t}^{t-1} \mathbb{E} \left\| \sum_{j \in \mathcal{A}_p} \sum_{k=0}^{K-1} \nabla f_j(x^{(a_{j,p},k)}) \right\|^2.
$$

Thus, summing over $t$ from 0 to $T-1$ on both sides of Equation (65) and invoking Lemma D.2 and Corollary D.3, we have the following bound holds no matter whether $\tau_t > 0$ or not:

$$
- \frac{\eta_g \eta_l}{N} \sum_{t=0}^{T-1} \mathbb{E} \left\langle \nabla f(x^{(t)}), \sum_{i \in \mathcal{A}_t} \sum_{k=0}^{K-1} \left[ g_i(x^{(a_{i,t},k)}, \xi_i^{(a_{i,t},k)}) - \nabla f_i(x^{(a_{i,t},k)}) \right] \right\rangle
$$

$$
\leq \sum_{t=0}^{T-1} \tau_t \frac{\eta_g^2 \eta_l^2}{N^2} NK\sigma^2 L + 2\frac{\eta_g^2 \eta_l^2}{N^2} NK\sigma^2 L \sum_{t=0}^{T-1} \tau_t + 2\frac{\eta_g^2 \eta_l^2}{N^2} L \sum_{t=0}^{T-1} \sum_{p=t-\tau_t}^{t-1} \mathbb{E} \left\| \sum_{j \in \mathcal{A}_p} \sum_{k=0}^{K-1} \nabla f_j(x^{(a_{j,p},k)}) \right\|^2
$$

$$
\leq 3\frac{\eta_g^2 \eta_l^2 K}{N} \tau_{\text{avg}} \sigma^2 LT + 2\frac{\eta_g^2 \eta_l^2}{N^2} \tau_{\text{max}} L \sum_{t=0}^{T-1} \mathbb{E} \left\| \sum_{i \in \mathcal{A}_t} \sum_{k=0}^{K-1} \nabla f_i(x^{(a_{i,t},k)}) \right\|^2,
$$

(66)

which gives the desired result. $\qquad\square$

**Lemma D.6.** *Suppose Assumption 1.1 and 1.2 hold. Then, for $\eta_l \leq \frac{1}{10\sqrt{\tau_{\text{max}} NKL}}$, and $\eta_g \eta_l \leq \frac{1}{10\tau_{\text{max}} KL}$, we have*

$$
\frac{1}{T} \sum_{t=0}^{T-1} \mathbb{E} \left\| \nabla f(x^{(t)}) \right\|^2 \leq \frac{2\Delta_f}{\eta_g \eta_l KT} + \frac{9\eta_g \eta_l \max\{1, \tau_{avg}\} \sigma^2 L}{N} + \frac{3\tau_{\text{max}} F_0}{T} + 150\eta_l^2 NK\sigma^2 L^2.
$$

*where $\Delta_f := f(x^{(0)}) - f^*$ and $F_0 := \frac{1}{n} \sum_{i=1}^{n} \left\| \nabla f_i(x^{(0)}) \right\|^2$.*

*Proof.* By Assumption 1.1, the function $f := \frac{1}{n} \sum_{i=1}^{n} f_i$ is $L$-smooth. Then, it holds by the descent lemma that

$$
\mathbb{E}f(x^{(t+1)}) \leq \mathbb{E}f(x^{(t)}) + \mathbb{E} \left\langle \nabla f(x^{(t)}), x^{(t+1)} - x^{(t)} \right\rangle + \frac{L}{2} \left\| x^{(t+1)} - x^{(t)} \right\|^2. \qquad (67)
$$

For the inner product term in Equation (67), there holds that

$$
\mathbb{E} \left\langle \nabla f(x^{(t)}), x^{(t+1)} - x^{(t)} \right\rangle = \mathbb{E} \left\langle \nabla f(x^{(t)}), -\frac{\eta_g \eta_l K}{N} y^{(t)} \right\rangle
$$

$$
= -\frac{\eta_g \eta_l}{N} \mathbb{E} \left\langle \nabla f(x^{(t)}), \sum_{i \in A_t} \sum_{k=0}^{K-1} \left[ g_i(x_i^{(a_{i,t},k)}, \xi_i^{(a_{i,t},k)}) - \nabla f_i(x_i^{(a_{i,t},k)}) \right] \right\rangle
$$

(68)

$$
- \frac{\eta_g \eta_l}{N^2} \mathbb{E} \left\langle \sum_{i=1}^{N} \nabla f_i(x^{(t)}), \sum_{i \in A_t} \sum_{k=0}^{K-1} \nabla f_i(x_i^{(a_{i,t},k)}) \right\rangle.
$$

We bound the first term after summing over $t$ in Equation (68) according to Lemma D.5. Then, for the second term in Equation (68), we have

$$
-\frac{\eta_g\eta_l}{N^2}\mathbb{E}\left\langle \sum_{i=1}^{N}\nabla f_i(x^{(t)}), \sum_{i\in A_t}\sum_{k=0}^{K-1}\nabla f_i(x_i^{(a_{i,t},k)})\right\rangle = -\frac{\eta_g\eta_l}{N^2K}\mathbb{E}\left\langle K\sum_{i=1}^{N}\nabla f_i(x^{(t)}), \sum_{i\in A_t}\sum_{k=0}^{K-1}\nabla f_i(x_i^{(a_{i,t},k)})\right\rangle
$$

$$
= -\frac{1}{2}\eta_g\eta_l K\mathbb{E}\left\|\nabla f(x^{(t)})\right\|^2 - \frac{\eta_g\eta_l}{2N^2K}\mathbb{E}\left\|\sum_{i\in A_t}\sum_{k=0}^{K-1}\nabla f_i(x_i^{(a_{i,t},k)})\right\|^2
$$

$$
+ \frac{\eta_g\eta_l}{2N^2K}\mathbb{E}\left\|K\sum_{i=1}^{N}\nabla f_i(x^{(t)}) - \sum_{i\in A_t}\sum_{k=0}^{K-1}\nabla f_i(x_i^{(a_{i,t},k)})\right\|^2.
$$

$$(69)$$

Then, summing over $t$ from 0 to $T-1$ on both sides of Equation (67) and taking full expectation, we have

$$
0 \leq \Delta_f - \frac{\eta_g\eta_l}{N}\sum_{t=0}^{T-1}\mathbb{E}\left\langle \nabla f(x^{(t)}), \sum_{i\in A_t}\sum_{k=0}^{K-1}\left[g_i(x_i^{(a_{i,t},k)}, \xi_i^{(a_{i,t},k)}) - \nabla f_i(x_i^{(a_{i,t},k)})\right]\right\rangle
$$

$$
- \frac{1}{2}\eta_g\eta_l K\sum_{t=0}^{T-1}\mathbb{E}\left\|\nabla f(x^{(t)})\right\|^2 - \frac{\eta_g\eta_l}{2N^2K}\sum_{t=0}^{T}\mathbb{E}\left\|\sum_{i\in A_t}\sum_{k=0}^{K-1}\nabla f_i(x_i^{(a_{i,t},k)})\right\|^2
$$

$$
+ \frac{\eta_g\eta_l}{2N^2K}\sum_{t=0}^{T-1}\mathbb{E}\left\|K\sum_{i=1}^{N}\nabla f_i(x^{(t)}) - \sum_{i\in A_t}\sum_{k=0}^{K-1}\nabla f_i(x_i^{(a_{i,t},k)})\right\|^2 + \frac{L}{2}\sum_{t=0}^{T}\mathbb{E}\left\|x^{(t+1)} - x^{(t)}\right\|^2.
$$

$$(70)$$

where $\Delta_f := f(x^{(0)}) - f^*$. Implementing Lemma D.1 into Equation (70), we have

$$
0 \leq \Delta_f - \frac{\eta_g\eta_l}{N}\sum_{t=0}^{T-1}\mathbb{E}\left\langle \nabla f(x^{(t)}), \sum_{i\in A_t}\sum_{k=0}^{K-1}\left[g_i(x_i^{(a_{i,t},k)}, \xi_i^{(a_{i,t},k)}) - \nabla f_i(x_i^{(a_{i,t},k)})\right]\right\rangle
$$

$$
- \frac{1}{2}\eta_g\eta_l K\sum_{t=0}^{T-1}\mathbb{E}\left\|\nabla f(x^{(t)})\right\|^2 - \frac{\eta_g\eta_l}{2N^2K}\left(1 - \eta_g\eta_l KL\right)\sum_{t=0}^{T}\mathbb{E}\left\|\sum_{i\in A_t}\sum_{k=0}^{K-1}\nabla f_i(x_i^{(a_{i,t},k)})\right\|^2
$$

$$
+ \frac{\eta_g\eta_l}{2N^2K}\sum_{t=0}^{T-1}\mathbb{E}\left\|K\sum_{i=1}^{N}\nabla f_i(x^{(t)}) - \sum_{i\in A_t}\sum_{k=0}^{K-1}\nabla f_i(x_i^{(a_{i,t},k)})\right\|^2 + \frac{\eta_g^2\eta_l^2 KT\sigma^2 L}{N}.
$$

$$(71)$$

Implementing Lemma D.5 into Equation (71), we have

$$
0 \leq \Delta_f - \frac{1}{2}\eta_g\eta_l K\sum_{t=0}^{T-1}\mathbb{E}\left\|\nabla f(x^{(t)})\right\|^2 - \frac{\eta_g\eta_l}{2N^2K}\left(1 - \eta_g\eta_l KL - 4\eta_g\eta_l \tau_{\max}KL\right)\sum_{t=0}^{T}\mathbb{E}\left\|\sum_{i\in A_t}\sum_{k=0}^{K-1}\nabla f_i(x_i^{(a_{i,t},k)})\right\|^2
$$

$$
+ \frac{\eta_g\eta_l}{2N^2K}\sum_{t=0}^{T-1}\mathbb{E}\left\|K\sum_{i=1}^{N}\nabla f_i(x^{(t)}) - \sum_{i\in A_t}\sum_{k=0}^{K-1}\nabla f_i(x_i^{(a_{i,t},k)})\right\|^2 + \frac{4\eta_g^2\eta_l^2 \max\{1, \tau_{\text{avg}}\}KT\sigma^2 L}{N}.
$$

$$(72)$$

Implementing Lemma D.4 into Equation (72), we have, for $\eta_l \leq \frac{1}{10\sqrt{\tau_{\max}}KL}$,

$$
\begin{aligned}
0 \leq &\Delta_f - \frac{1}{2}\eta_g\eta_l K \sum_{t=0}^{T-1} \mathbb{E}\left\|\nabla f(x^{(t)})\right\|^2 \\
&- \frac{\eta_g\eta_l}{2N^2 K}\left(1 - \eta_g\eta_l KL - 4\eta_g\eta_l\tau_{\max}KL - 10\eta_g^2\eta_l^2\tau_{\max}^2 K^2 L^2 - 24\eta_l^2 K^2 N^2\tau_{\max}L^2\right)\cdot \\
&\qquad \sum_{t=0}^{T} \mathbb{E}\left\|\sum_{i\in A_t}\sum_{k=0}^{K-1}\nabla f_i(x_i^{(a_{i,t},k)})\right\|^2 + \frac{3\eta_g\eta_l\tau_{\max}KF_0}{2} \\
&+ \frac{4\eta_g^2\eta_l^2\max\{1,\tau_{\mathrm{avg}}\}KT\sigma^2 L}{N} + \frac{4\eta_g^3\eta_l^3\tau_{\max}\max\{1,\tau_{\mathrm{avg}}\}K^2 T\sigma^2 L^2}{N} + 75\eta_g\eta_l^3 NK^2 T\sigma^2 L^2.
\end{aligned}
$$
(73)

Then, for $\eta_l \leq \frac{1}{10\sqrt{\tau_{\max}}NKL}$, and $\eta_g\eta_l \leq \frac{1}{10\tau_{\max}KL}$, we have

$$
\begin{aligned}
&1 - \eta_g\eta_l KL - 4\eta_g\eta_l\tau_{\max}KL - 10\eta_g^2\eta_l^2\tau_{\max}^2 K^2 L^2 - 24\eta_l^2 K^2 N^2\tau_{\max}L^2 \\
&\geq 1 - \frac{1}{10} - \frac{2}{5} - \frac{1}{10} - \frac{6}{25} > 0.
\end{aligned}
$$
(74)

Therefore, for $\eta_l \leq \frac{1}{10\sqrt{\tau_{\max}}NKL}$, and $\eta_g\eta_l \leq \frac{1}{10\tau_{\max}KL}$, by rearranging Equation (73) and noticing

$$
\frac{4\eta_g^3\eta_l^3\tau_{\max}\max\{1,\tau_{\mathrm{avg}}\}K^2 T\sigma^2 L^2}{N} \leq \frac{\eta_g^2\eta_l^2\max\{1,\tau_{\mathrm{avg}}\}KT\sigma^2 L}{2N},
$$

it implies that

$$
\frac{1}{T}\sum_{t=0}^{T-1}\mathbb{E}\left\|\nabla f(x^{(t)})\right\|^2 \leq \frac{2\Delta_f}{\eta_g\eta_l KT} + \frac{9\eta_g\eta_l\max\{1,\tau_{\mathrm{avg}}\}\sigma^2 L}{N} + \frac{3\tau_{\max}F_0}{T} + 150\eta_l^2 NK\sigma^2 L^2.
$$
(75)

$\square$

**Theorem D.1.** *Suppose Assumption 1.1 and 1.2 hold. Then, for*

$$
\eta_g = \frac{N}{\sqrt{\tau_{\max}}}, \text{ and } \eta_l = \min\left\{\frac{1}{10\sqrt{\tau_{\max}}NKL}, \frac{\sqrt{\tau_{\max}\Delta_f}}{\sqrt{N\max\{1,\tau_{avg}\}KTL\sigma^2}}\right\},
$$

*we have*

$$
\frac{1}{T}\sum_{t=0}^{T-1}\mathbb{E}\left\|\nabla f(x^{(t)})\right\|^2 \leq \frac{30\sqrt{\max\{1,\tau_{avg}\}L\sigma^2\Delta_f}}{\sqrt{NKT}} + \frac{20\tau_{\max}(L\Delta_f + F_0)}{T},
$$

*where $\Delta_f := f(x^{(0)}) - f^*$ and $F_0 := \frac{1}{N}\sum_{i=1}^{N}\left\|\nabla f_i(x^{(0)})\right\|^2$.*

*Proof.* Invoking Lemma D.6, we have, for $\eta_l \leq \frac{1}{10\sqrt{\tau_{\max}}NKL}$, and $\eta_g\eta_l \leq \frac{1}{10\tau_{\max}KL}$,

$$
\frac{1}{T}\sum_{t=0}^{T-1}\mathbb{E}\left\|\nabla f(x^{(t)})\right\|^2 \leq \frac{2\Delta_f}{\eta_g\eta_l KT} + \frac{9\eta_g\eta_l\max\{1,\tau_{\mathrm{avg}}\}\sigma^2 L}{N} + \frac{3\tau_{\max}F_0}{T} + 150\eta_l^2 NK\sigma^2 L^2.
$$
(76)

Choosing

$$
\eta_g = \frac{N}{\sqrt{\tau_{\max}}}, \quad \eta_l = \min\left\{\frac{1}{10\sqrt{\tau_{\max}}NKL}, \frac{\sqrt{\tau_{\max}\Delta_f}}{\sqrt{N\max\{1,\tau_{\mathrm{avg}}\}KTL\sigma^2}}\right\},
$$

we have

$$\eta_g \eta_l = \min \left\{ \frac{\sqrt{N\Delta_f}}{\sqrt{\max\{1, \tau_{\text{avg}}\} KTL\sigma^2}}, \frac{1}{10\tau_{\max}KL} \right\},$$

$$\frac{2\Delta_f}{\eta_g \eta_l KT} \leq \frac{2\sqrt{\max\{1, \tau_{\text{avg}}\} L\sigma^2 \Delta_f}}{\sqrt{NKT}} + \frac{20\tau_{\max}L\Delta_f}{T},$$

$$9\frac{\eta_g \eta_l}{N} \max\{1, \tau_{\text{avg}}\} \sigma^2 L \leq \frac{9\sqrt{\max\{1, \tau_{\text{avg}}\} L\sigma^2 \Delta_f}}{\sqrt{NKT}},$$

$$150\eta_l^2 NK\sigma^2 L^2 \leq 150\frac{1}{10\sqrt{\tau_{\max}}NKL} \frac{\sqrt{\tau_{\max}\Delta_f}}{\sqrt{N\max\{1, \tau_{\text{avg}}\} KTL\sigma^2}} NK\sigma^2 L^2 \leq \frac{15\sqrt{L\sigma^2\Delta_f}}{\sqrt{NKT}}.$$

Thus, we can combine the above bounds with Equation (76) to get

$$\frac{1}{T} \sum_{t=0}^{T-1} \mathbb{E}\left\| \nabla f(x^{(t)}) \right\|^2 \leq \frac{26\sqrt{\max\{1, \tau_{\text{avg}}\} L\sigma^2 \Delta_f}}{\sqrt{NKT}} + \frac{20\tau_{\max}(L\Delta_f + F_0)}{T},$$

which gives the desired result by appropriately magnifying the constants. $\qquad \square$

# E  CONVERGENCE ANALYSIS FOR FEDSUM-CR

In this section, we prove the convergence result of FedSUM-CR.

---

**Algorithm 3** FedSUM-CR: Enhancing Communication Efficiency in FedSUM

---

1: **Input:** initial model $x^{(0)}$, control variables $y^{(-1)}$, $\{h_i^{(0)}\}_{i=1}^N$ with value $\mathbf{0}$, $\{z_i^{(0)}\}_{i=1}^N$ with value $x^{(0)}$ and $\{a_i^{(0)}\}_{i=1}^N$ with value $-1$; global learning rate $\eta_g$; local learning rate $\eta_l$; local steps $K$; client participation $\{\mathcal{S}_t\}_{t=0}^{T-1}$

2: **for** $t = 0, 1, \cdots, T-1$ **do**

3:     Send $x^{(t)}$ to all clients $i \in \mathcal{S}_t$.

4:     **for** client $i \in \mathcal{S}_t$ in parallel **do**

5:         Receive $x^{(t)}$ and initialize local model $x_i^{(t,0)} = x^{(t)}$.

6:         Compute local update correction direction $y_i^{(t)} = \frac{N}{\eta_g \eta_l K} \frac{z_i^{(t)} - x^{(t)}}{t - a_i^{(t)}} - h_i^{(t)}$.

7:         **for** $k = 0, \cdots, K-1$ **do**

8:             Compute a mini-batch gradient $g_i^{(t,k)} = \nabla F_i(x_i^{(t,k)}; \xi_i^{(t,k)})$.

9:             Locally update $x_i^{(t,k+1)} = x_i^{(t,k)} - \frac{\eta_l}{N}\left(g_i^{(t,k)} + y_i^{(t)}\right)$.

10:         **end for**

11:         Compute $\delta_i^{(t)} = \frac{N\left(x^{(t)} - x_i^{(t,K)}\right)}{\eta_l K} - y_i^{(t)} - h_i^{(t)}$ and send $\delta_i^{(t)}$ to the server.

12:         Update $a_i^{(t+1)} = t$, $z_i^{(t+1)} = x^{(t)}$, and $h_i^{(t+1)} = \frac{N\left(x^{(t)} - x_i^{(t,K)}\right)}{\eta_l K} - y_i^{(t)}$

13:             (for $i \notin \mathcal{S}_t$, $z_i^{(t+1)} = z_i^{(t)}$, $a_i^{(t+1)} = a_i^{(t)}$ and $h_i^{(t+1)} = h_i^{(t)}$).

14:     **end for**

15:     Update $y^{(t)} = y^{(t-1)} + \sum_{i \in \mathcal{S}_t} \delta_i^{(t)}$ and $x^{(t+1)} = x^{(t)} - \frac{\eta_g \eta_l K}{N} y^{(t)}$.

16: **end for**

17: **Server** outputs $x^{(T)}$.

---

We derive the update direction $y^{(t)}$ in FedSUM-CR as follows.

$$y^{(t)} = \frac{1}{K} \sum_{i \in \mathcal{A}_t} \sum_{k=0}^{K-1} g_i(x_i^{(a_{i,t},k)}, \xi_i^{(a_{i,t},k)}). \tag{77}$$

For the convenience of notation, we define that $a_{i,-1} = -1$ for all $i \in \mathcal{N}$ and $x^{(-1)} = x^{(0)}$. Then, we have the control variables:

$$h_i^{(t)} = \frac{1}{K} \sum_{k=0}^{K-1} g_i(x_i^{(a_{i,t-1},k)}, \xi_i^{(a_{i,t-1},k)}),$$

$$a_i^{(t)} = a_{i,t-1}, z_i^{(t)} = x^{(a_{i,t-1})}.$$

(78)

Furthermore, the local update corrected direction $y_i^{(t)}$ is derived as follows:

$$
\begin{aligned}
y_i^{(t)} &= \frac{N}{\eta_g \eta_l K} \frac{z_i^{(t)} - x^{(t)}}{t - a_i^{(t)}} - h_i^{(t)} = \frac{N}{\eta_g \eta_l K} \frac{x^{(a_{i,t-1})} - x^{(t)}}{t - a_{i,t-1}} - h_i^{(t)} \\
&= \frac{N}{\eta_g \eta_l K} \frac{1}{t - a_{i,t-1}} \sum_{p=a_{i,t-1}}^{t-1} \left( x^{(p)} - x^{(p+1)} \right) - h_i^{(t)} \\
&= \frac{1}{t - a_{i,t-1}} \sum_{p=a_{i,t-1}}^{t-1} y^{(p)} - h_i^{(t)}.
\end{aligned}
$$

(79)

Lemma E.1 establishes a basic relationship between the difference of two consecutive model updates and the aggregated gradient, shown as follows.

**Lemma E.1.** *Suppose Assumption 1.1 and 1.2 hold. Then, it holds that*

$$\sum_{t=0}^{T-1} \mathbb{E} \left\| x^{(t+1)} - x^{(t)} \right\|^2 \leq \frac{2\eta_g^2 \eta_l^2 KT\sigma^2}{N} + \frac{2\eta_g^2 \eta_l^2}{N^2} \sum_{t=0}^{T-1} \mathbb{E} \left\| \sum_{i \in \mathcal{A}_t} \sum_{k=0}^{K-1} \nabla f_i(x_i^{(a_{i,t},k)}) \right\|^2.$$

*Proof.* Similar to the proof of Lemma D.1. $\square$

As a direct consequence of Lemma B.2, we conduct the Lemma E.2 and Corollary E.3 to bound the aggregated gradient of server.

**Lemma E.2.** *Suppose Assumption 1.1 and 1.2 hold. Then, we have*

$$\sum_{t=0}^{T-1} \sum_{p=t-\tau_t}^{t-1} \mathbb{E} \left\| \sum_{j \in \mathcal{A}_p} \sum_{k=0}^{K-1} \nabla f_j(x_j^{(a_{j,p},k)}) \right\|^2 \leq \tau_{\max} \sum_{t=0}^{T-1} \mathbb{E} \left\| \sum_{i \in \mathcal{A}_t} \sum_{k=0}^{K-1} \nabla f_i(x_i^{(a_{i,t},k)}) \right\|^2.$$

*Proof.* The proof is similar to the proof of Lemma D.2. $\square$

Corollary E.3 is a direct consequence of Lemma E.2.

**Corollary E.3.** *Suppose Assumption 1.1 and 1.2 hold. Then, we have*

$$\sum_{t=0}^{T-1} \tau_t \sum_{p=t-\tau_t}^{t-1} \mathbb{E} \left\| \sum_{j \in \mathcal{A}_p} \sum_{k=0}^{K-1} \nabla f_j(x_j^{(a_{j,p},k)}) \right\|^2 \leq \tau_{\max}^2 \sum_{t=0}^{T-1} \mathbb{E} \left\| \sum_{i \in \mathcal{A}_t} \sum_{k=0}^{K-1} \nabla f_i(x_i^{(a_{i,t},k)}) \right\|^2.$$

Lemma E.4 is the key lemma to estimate the difference between the aggregated gradient of server and the aggregated gradient of all clients.

**Lemma E.4.** *Suppose Assumption 1.1 and 1.2 hold. Then, we have, for $\eta_l \leq \frac{1}{10\sqrt{\tau_{\max}}KL}$,*

$$
\begin{aligned}
&\sum_{t=0}^{T-1} \mathbb{E} \left\| K \sum_{i=1}^{N} \nabla f_i(x^{(t)}) - \sum_{i \in \mathcal{A}_t} \sum_{k=0}^{K-1} \nabla f_i(x_i^{(a_{i,t},k)}) \right\|^2 \\
&\leq 3K^2 N^2 \tau_{\max} F_0 + 8\eta_g^2 \eta_l^2 K^3 N \tau_{\max} \max\{1, \tau_{avg}\} T\sigma^2 L^2 + 150\eta_l^2 N^3 K^3 \tau_{\max} T\sigma^2 L^2 \\
&\quad + \left( 10\eta_g^2 \eta_l^2 \tau_{\max}^2 K^2 L^2 + 24\eta_l^2 K^2 N^2 \tau_{\max} L^2 \right) \sum_{t=0}^{T-1} \mathbb{E} \left\| \sum_{j \in \mathcal{A}_t} \sum_{k=0}^{K-1} \nabla f_j(x_j^{(a_{j,t},k)}) \right\|^2,
\end{aligned}
$$

*where $F_0 := \frac{1}{N} \sum_{i=1}^{N} \left\| \nabla f_i(x^{(0)}) \right\|^2$.*

*Proof.* Notice that, by the decomposition of the difference between the aggregated gradient of the server and the aggregated gradient of all clients, we have

$$
\left\| K \sum_{i=1}^{N} \nabla f_i(x^{(t)}) - \sum_{i \in \mathcal{A}_t} \sum_{k=0}^{K-1} \nabla f_i(x_i^{(a_{i,t},k)}) \right\|^2 \leq 3K^2 \left\| \sum_{i \in \mathcal{N}} \nabla f_i(x^{(t)}) - \sum_{i \in \mathcal{N}} \nabla f_i(x^{(a_{i,t})}) \right\|^2
$$

$$
+ 3 \left\| K \sum_{i \in \mathcal{A}_t} \nabla f_i(x^{(a_{i,t})}) - \sum_{i \in \mathcal{A}_t} \sum_{k=0}^{K-1} \nabla f_i(x_i^{(a_{i,t},k)}) \right\|^2 + 3K^2 \left\| \sum_{i \in \mathcal{N}/\mathcal{A}_t} \nabla f_i(x^{(0)}) \right\|^2.
$$

$$(80)$$

We bound the expectation of terms in Equation (80) one by one. First of all, we have

$$
\mathbb{E} \left\| \sum_{i \in \mathcal{N}} \nabla f_i(x^{(t)}) - \sum_{i \in \mathcal{N}} \nabla f_i(x^{(a_{i,t})}) \right\|^2 \leq NL^2 \sum_{i \in \mathcal{N}} \mathbb{E} \left\| x^{(t)} - x^{(a_{i,t})} \right\|^2
$$

$$
= NL^2 \sum_{i \in \mathcal{N}} \mathbb{E} \left\| \sum_{p=a_{i,t}}^{t-1} \left[ x^{(p+1)} - x^{(p)} \right] \right\|^2 = \frac{\eta_g^2 \eta_l^2}{N} L^2 \sum_{i \in \mathcal{N}} \mathbb{E} \left\| \sum_{p=a_{i,t}}^{t-1} \sum_{j \in \mathcal{A}_p} \sum_{k=0}^{K-1} g_j(x_j^{(a_{j,p},k)}, \xi_j^{(a_{j,p},k)}) \right\|^2.
$$

$$(81)$$

By Assumption 1.2, we have, according to Equation (81) and summing over $t$,

$$
3K^2 \sum_{t=0}^{T-1} \mathbb{E} \left\| \sum_{i \in \mathcal{N}} \nabla f_i(x^{(t)}) - \sum_{i \in \mathcal{N}} \nabla f_i(x^{(a_{i,t})}) \right\|^2
$$

$$
\leq 6K^2 \frac{\eta_g^2 \eta_l^2}{N} L^2 \sum_{t=0}^{T-1} \sum_{i \in \mathcal{N}} \tau_{\max} \sum_{p=a_{i,t}}^{t-1} \sum_{j \in \mathcal{A}_p} \sum_{k=0}^{K-1} \sigma^2 + 6K^2 \frac{\eta_g^2 \eta_l^2}{N} L^2 \sum_{t=0}^{T-1} \sum_{i \in \mathcal{N}} \mathbb{E} \left\| \sum_{p=a_{i,t}}^{t-1} \sum_{j \in \mathcal{A}_p} \sum_{k=0}^{K-1} \nabla f_j(x_j^{(a_{j,p},k)}) \right\|^2
$$

$$
\leq 6\eta_g^2 \eta_l^2 K^3 N \tau_{\max} \tau_{\mathrm{avg}} T \sigma^2 L^2 + 6\eta_g^2 \eta_l^2 K^2 \tau_{\max}^2 L^2 \sum_{t=0}^{T-1} \mathbb{E} \left\| \sum_{i \in \mathcal{A}_t} \sum_{k=0}^{K-1} \nabla f_i(x_i^{(a_{i,t},k)}) \right\|^2,
$$

$$(82)$$

where the procedure is similar to the proof of Lemma C.4.

For the second part of Equation (44), we have, by Assumption 1.1,

$$
\left\| K \sum_{i \in \mathcal{A}_t} \nabla f_i(x^{(a_{i,t})}) - \sum_{i \in \mathcal{A}_t} \sum_{k=0}^{K-1} \nabla f_i(x_i^{(a_{i,t},k)}) \right\|^2 = \left\| \sum_{i \in \mathcal{A}_t} \sum_{k=0}^{K-1} \left[ \nabla f_i(x^{(a_{i,t})}) - \nabla f_i(x_i^{(a_{i,t},k)}) \right] \right\|^2
$$

$$
\leq NKL^2 \sum_{i \in \mathcal{A}_t} \sum_{k=0}^{K-1} \left\| x^{(a_{i,t})} - x_i^{(a_{i,t},k)} \right\|^2.
$$

$$(83)$$

Note that $x^{(a_{i,t})}$ is the local model of client $i$ at the beginning of round $a_{i,t}$, and $x_i^{(a_{i,t},k)}$ is the local model of client $i$ at the beginning of round $a_{i,t}$ after $k$ local updates. Then, according to the

definition of $h_i^{(t)}$ and $y_i^{(t)}$ shown in Equation (78) and (79), it holds that

$$
\begin{aligned}
\sum_{i \in \mathcal{A}_t} \sum_{k=0}^{K-1} \left\| x^{(a_{i,t})} - x_i^{(a_{i,t},k)} \right\|^2 &= \sum_{i \in \mathcal{A}_t} \sum_{k=0}^{K-1} \left\| \sum_{p=0}^{k-1} \left[ x_i^{(a_{i,t},p)} - x_i^{(a_{i,t},p+1)} \right] \right\|^2 \\
&= \eta_l^2 \sum_{i \in \mathcal{A}_t} \sum_{k=0}^{K-1} \left\| \sum_{p=0}^{k-1} \left[ g_i(x_i^{(a_{i,t},p)}, \xi_i^{(a_{i,t},p)}) + y_i^{(a_{i,t})} \right] \right\|^2 \\
&\leq 2\eta_l^2 \sum_{i \in \mathcal{A}_t} \sum_{k=0}^{K-1} \left\| \sum_{p=0}^{k-1} \left[ g_i(x_i^{(a_{i,t},p)}, \xi_i^{(a_{i,t},p)}) - h_i^{(a_{i,t})} \right] \right\|^2 \\
&\quad + 2\eta_l^2 \sum_{i \in \mathcal{A}_t} \sum_{k=0}^{K-1} \frac{1}{(a_{i,t} - a_{i,a_{i,t}-1})^2} \left\| \sum_{p=0}^{k-1} \sum_{q=a_{i,a_{i,t}-1}}^{a_{i,t}-1} y^{(q)} \right\|^2 .
\end{aligned}
\tag{84}
$$

We deal with the first term in Equation (84) as follows. Notice that, by the definition of $h_i^{(t)}$ in Equation (78) that

$$
h_i^{(a_{i,t})} = \frac{1}{K} \sum_{k=0}^{K-1} g_i(x_i^{(a_{i,a_{i,t}-1},k)}, \xi_i^{(a_{i,a_{i,t}-1},k)}),
$$

where $a_{i,a_{i,t}-1}$ is the round when client $i$ is selected in the second last round before $t$. Then, we have

$$
\begin{aligned}
&\left\| \sum_{p=0}^{k-1} \left[ g_i(x_i^{(a_{i,t},p)}, \xi_i^{(a_{i,t},p)}) - h_i^{(a_{i,t})} \right] \right\|^2 \\
&\leq 5 \left\| \sum_{p=0}^{k-1} \left[ g_i(x_i^{(a_{i,t},p)}, \xi_i^{(a_{i,t},p)}) - \nabla f_i(x_i^{(a_{i,t},p)}) \right] \right\|^2 + 5 \left\| \sum_{p=0}^{k-1} \left[ \nabla f_i(x_i^{(a_{i,t},p)}) - \nabla f_i(x^{(a_{i,t})}) \right] \right\|^2 \\
&\quad + 5k^2 \left\| \nabla f_i(x^{(a_{i,t})}) - \nabla f_i(x^{(a_{i,a_{i,t}-1})}) \right\|^2 + 5 \frac{k^2}{K^2} \left\| \sum_{q=0}^{K-1} \left[ \nabla f_i(x^{(a_{i,a_{i,t}-1})}) - \nabla f_i(x_i^{(a_{i,a_{i,t}-1},q)}) \right] \right\|^2 \\
&\quad + 5 \frac{k^2}{K^2} \left\| \sum_{q=0}^{K-1} \left[ g_i(x_i^{(a_{i,a_{i,t}-1},q)}, \xi_i^{(a_{i,a_{i,t}-1},q)}) - \nabla f_i(x^{(a_{i,a_{i,t}-1},q)}) \right] \right\|^2 .
\end{aligned}
\tag{85}
$$

Then, by taking expectation and invoking Equation (84), we have the following upper bound for each term in Equation (85).

**Term 1:**

$$
\begin{aligned}
&\eta_l^2 \sum_{t=0}^{T-1} \sum_{i \in \mathcal{A}_t} \sum_{k=0}^{K-1} \mathbb{E} \left\| \sum_{p=0}^{k-1} \left[ g_i(x_i^{(a_{i,t},p)}, \xi_i^{(a_{i,t},p)}) - \nabla f_i(x_i^{(a_{i,t},p)}) \right] \right\|^2 \\
&\leq \eta_l^2 \sum_{t=0}^{T-1} \sum_{i \in \mathcal{A}_t} \sum_{k=0}^{K-1} \sum_{p=0}^{k-1} \sigma^2 \leq \eta_l^2 N K^2 T \sigma^2 .
\end{aligned}
\tag{86}
$$

**Term 2:**

$$
\begin{aligned}
&\eta_l^2 \sum_{t=0}^{T-1} \sum_{i \in \mathcal{A}_t} \sum_{k=0}^{K-1} \mathbb{E} \left\| \sum_{p=0}^{k-1} \left[ \nabla f_i(x_i^{(a_{i,t},p)}) - \nabla f_i(x^{(a_{i,t})}) \right] \right\|^2 \leq \eta_l^2 L^2 \sum_{t=0}^{T-1} \sum_{i \in \mathcal{A}_t} \sum_{k=0}^{K-1} k \sum_{p=0}^{k-1} \mathbb{E} \left\| x_i^{(a_{i,t},p)} - x^{(a_{i,t})} \right\|^2 \\
&\leq \eta_l^2 L^2 \sum_{t=0}^{T-1} \sum_{i \in \mathcal{A}_t} \sum_{k=0}^{K-1} k \sum_{p=0}^{k-1} \mathbb{E} \left\| x_i^{(a_{i,t},p)} - x^{(a_{i,t})} \right\|^2 \leq \eta_l^2 K^2 L^2 \sum_{t=0}^{T-1} \sum_{i \in \mathcal{A}_t} \sum_{k=0}^{K-1} \mathbb{E} \left\| x_i^{(a_{i,t},k)} - x^{(a_{i,t})} \right\|^2 .
\end{aligned}
\tag{87}
$$

**Term 3:**

$$\eta_l^2 \sum_{t=0}^{T-1} \sum_{i \in \mathcal{A}_t} \sum_{k=0}^{K-1} k^2 \left\| \nabla f_i(x^{(a_{i,t})}) - \nabla f_i(x^{(a_{i,a_{i,t}-1})}) \right\|^2$$

$$\leq \eta_l^2 K^3 L^2 \sum_{t=0}^{T-1} \sum_{i \in \mathcal{A}_t} \mathbb{E} \left\| x^{(a_{i,t})} - x^{(a_{i,a_{i,t}-1})} \right\|^2 = \eta_l^2 K^3 L^2 \sum_{t=0}^{T-1} \sum_{i \in \mathcal{A}_t} \mathbb{E} \left\| \sum_{p=a_{i,a_{i,t}-1}}^{a_{i,t}-1} \left[ x^{(p+1)} - x^{(p)} \right] \right\|^2$$

$$\leq \frac{2\eta_g^2 \eta_l^4 K^3 L^2}{N^2} \sum_{t=0}^{T-1} \sum_{i \in \mathcal{A}_t} \left[ a_{i,t} - a_{i,a_{i,t}-1} \right] \sum_{p=a_{i,a_{i,t}-1}}^{a_{i,t}} \mathbb{E} \left\| \sum_{j \in \mathcal{A}_p} \sum_{k=0}^{K-1} \nabla f_j(x_j^{(a_{j,p},k)}) \right\|^2$$

$$+ \frac{2\eta_g^2 \eta_l^4 K^3 L^2}{N^2} \sum_{t=0}^{T-1} \sum_{i \in \mathcal{A}_t} \left[ a_{i,t} - a_{i,a_{i,t}-1} \right]^2 N K \sigma^2.$$

(88)

With the fact that $a_{i,a_{i,t}-1}$ is the second last round when client $i$ is selected before $t$ and $a_{i,t}$ is the last round when client $i$ is selected, it holds that $a_{i,t} - a_{i,a_{i,t}-1} \leq \tau_{\max}$. Then,

$$\eta_l^2 \sum_{t=0}^{T-1} \sum_{i \in \mathcal{A}_t} \sum_{k=0}^{K-1} k^2 \left\| \nabla f_i(x^{(a_{i,t})}) - \nabla f_i(x^{(a_{i,a_{i,t}-1})}) \right\|^2$$

$$\leq \frac{2\eta_g^2 \eta_l^4 K^3 L^2}{N^2} \sum_{t=0}^{T-1} \sum_{i \in \mathcal{A}_t} \tau_{\max} \sum_{p=\max\{0, a_{i,t}-\tau_{\max}\}}^{a_{i,t}} \mathbb{E} \left\| \sum_{j \in \mathcal{A}_p} \sum_{k=0}^{K-1} \nabla f_j(x_j^{(a_{j,p},k)}) \right\|^2 + 2\eta_g^2 \eta_l^4 \tau_{\max}^2 K^4 T \sigma^2 L^2.$$

(89)

With the fact that $t - \tau_{\max} \leq a_{i,t} \leq t$, we have

$$\frac{2\eta_g^2 \eta_l^4 K^3 L^2}{N^2} \sum_{t=0}^{T-1} \sum_{i \in \mathcal{A}_t} \tau_{\max} \sum_{p=\max\{0, a_{i,t}-\tau_{\max}\}}^{a_{i,t}} \mathbb{E} \left\| \sum_{j \in \mathcal{A}_p} \sum_{k=0}^{K-1} \nabla f_j(x_j^{(a_{j,p},k)}) \right\|^2$$

$$\leq \frac{2\eta_g^2 \eta_l^4 \tau_{\max} K^3 L^2}{N^2} \sum_{t=0}^{T-1} \sum_{i \in \mathcal{A}_t} \sum_{p=\max\{0, t-2\tau_{\max}\}}^{t} \mathbb{E} \left\| \sum_{j \in \mathcal{A}_p} \sum_{k=0}^{K-1} \nabla f_j(x_j^{(a_{j,p},k)}) \right\|^2$$

$$\leq \frac{2\eta_g^2 \eta_l^4 \tau_{\max} K^3 L^2}{N} \sum_{t=0}^{T-1} \sum_{p=\max\{0, t-2\tau_{\max}\}}^{t} \mathbb{E} \left\| \sum_{j \in \mathcal{A}_p} \sum_{k=0}^{K-1} \nabla f_j(x_j^{(a_{j,p},k)}) \right\|^2$$

(90)

$$\leq \frac{2\eta_g^2 \eta_l^4 \tau_{\max} K^3 L^2}{N} \sum_{p=0}^{T-1} \sum_{t=p}^{\min\{T-1, p+2\tau_{\max}\}} \mathbb{E} \left\| \sum_{j \in \mathcal{A}_p} \sum_{k=0}^{K-1} \nabla f_j(x_j^{(a_{j,p},k)}) \right\|^2$$

$$\leq \frac{4\eta_g^2 \eta_l^4 \tau_{\max}^2 K^3 L^2}{N} \sum_{t=0}^{T-1} \mathbb{E} \left\| \sum_{j \in \mathcal{A}_t} \sum_{k=0}^{K-1} \nabla f_j(x_j^{(a_{j,t},k)}) \right\|^2.$$

**Term 4:**

$$\eta_l^2 \sum_{t=0}^{T-1} \sum_{i \in \mathcal{A}_t} \sum_{k=0}^{K-1} \frac{k^2}{K^2} \mathbb{E} \left\| \sum_{q=0}^{K-1} \left[ \nabla f_i(x^{(a_{i,a_{i,t}-1})}) - \nabla f_i(x_i^{(a_{i,a_{i,t}-1},q)}) \right] \right\|^2$$

$$\leq \eta_l^2 L^2 \sum_{t=0}^{T-1} \sum_{i \in \mathcal{A}_t} \sum_{k=0}^{K-1} \frac{k^2}{K} \sum_{q=0}^{K-1} \mathbb{E} \left\| x^{(a_{i,a_{i,t}-1})} - x_i^{(a_{i,a_{i,t}-1},q)} \right\|^2$$

(91)

$$\leq \eta_l^2 \tau_{\max} K^2 L^2 \sum_{t=0}^{T-1} \sum_{i \in \mathcal{A}_t} \sum_{k=0}^{K-1} \mathbb{E} \left\| x_i^{(a_{i,t},k)} - x^{(a_{i,t})} \right\|^2,$$

where the last inequality is due to the fact that $a_{i,a_{i,t}-1}$ is the second last round when client $i$ is selected before $t$ and $a_{i,t}$ is the last round when client $i$ is selected. Then, it holds that $a_{i,a_{i,t}+1} = a_{i,t}$. Hence, $\sum_{t=0}^{T-1} \sum_{i \in \mathcal{A}_t} \sum_{q=0}^{K-1} \mathbb{E} \left\| x^{(a_{i,a_{i,t}-1})} - x_i^{(a_{i,a_{i,t}-1},q)} \right\|^2 \leq$ $\tau_{\max} \sum_{t=0}^{T-1} \sum_{i \in \mathcal{A}_t} \sum_{k=0}^{K-1} \mathbb{E} \left\| x_i^{(a_{i,t},k)} - x^{(a_{i,t})} \right\|^2$.

**Term 5:**

$$\eta_l^2 \sum_{t=0}^{T-1} \sum_{i \in \mathcal{A}_t} \sum_{k=0}^{K-1} \frac{k^2}{K^2} \mathbb{E} \left\| \sum_{q=0}^{K-1} \left[ g_i(x_i^{(a_{i,a_{i,t}-1},q)}, \xi_i^{(a_{i,a_{i,t}-1},q)}) - \nabla f_i(x^{(a_{i,a_{i,t}-1},q)}) \right] \right\|^2$$

$$\leq \eta_l^2 \sum_{t=0}^{T-1} \sum_{i \in \mathcal{A}_t} \sum_{k=0}^{K-1} \frac{k^2}{K^2} \sum_{q=0}^{K-1} \sigma^2 \leq \eta_l^2 N K^2 T \sigma^2. \tag{92}$$

For the second term in Equation (84), we have

$$2\eta_l^2 \sum_{i \in \mathcal{A}_t} \sum_{k=0}^{K-1} \frac{1}{(a_{i,t} - a_{i,a_{i,t}-1})^2} \left\| \sum_{p=0}^{k-1} \sum_{q=a_{i,a_{i,t}-1}}^{a_{i,t}-1} y^{(q)} \right\|^2 \leq 2\eta_l^2 K^3 \sum_{i \in \mathcal{A}_t} \frac{1}{(a_{i,t} - a_{i,a_{i,t}-1})^2} \left\| \sum_{q=a_{i,a_{i,t}-1}}^{a_{i,t}-1} y^{(q)} \right\|^2. \tag{93}$$

Consequently, it implies by the fact $a_{i,t} - a_{i,a_{i,t}-1} \leq \tau_{\max}$ that

$$2\eta_l^2 K^3 \sum_{t=0}^{T-1} \sum_{i \in \mathcal{A}_t} \frac{1}{(a_{i,t} - a_{i,a_{i,t}-1})^2} \mathbb{E} \left\| \sum_{q=a_{i,a_{i,t}-1}}^{a_{i,t}-1} y^{(q)} \right\|^2$$

$$= 2\eta_l^2 K \sum_{t=0}^{T-1} \sum_{i \in \mathcal{A}_t} \frac{1}{(a_{i,t} - a_{i,a_{i,t}-1})^2} \mathbb{E} \left\| \sum_{q=a_{i,a_{i,t}-1}}^{a_{i,t}-1} \sum_{j \in \mathcal{A}_q} \sum_{k=0}^{K-1} g_j(x_j^{(a_{j,q},k)}, \xi_j^{(a_{j,q},k)}) \right\|^2$$

$$\leq 4\eta_l^2 K \sum_{t=0}^{T-1} \sum_{i \in \mathcal{A}_t} \frac{1}{a_{i,t} - a_{i,a_{i,t}-1}} \sum_{q=a_{i,a_{i,t}-1}}^{a_{i,t}-1} \mathbb{E} \left\| \sum_{j \in \mathcal{A}_q} \sum_{k=0}^{K-1} \nabla f_j(x_j^{(a_{j,q},k)}) \right\|^2 + 4\eta_l^2 K^2 N^2 \sigma^2 T$$

$$\leq 4\eta_l^2 K N \tau_{\max} \sum_{t=0}^{T-1} \mathbb{E} \left\| \sum_{i \in \mathcal{A}_t} \sum_{k=0}^{K-1} \nabla f_i(x_i^{(a_{i,t},k)}) \right\|^2 + 4\eta_l^2 K^2 N^2 \sigma^2 T, \tag{94}$$

where the last equality holds by Equation (90).

Finally, after summing $t$ from 0 to $T-1$ in Equation (84), and implementing the upper bound for each term in Equation (85) as above, it holds that

$$\sum_{t=0}^{T-1} \sum_{i \in \mathcal{A}_t} \sum_{k=0}^{K-1} \mathbb{E} \left\| x^{(a_{i,t})} - x_i^{(a_{i,t},k)} \right\|^2 \leq 10\eta_l^2 N K^2 T \sigma^2 + 10\eta_l^2 K^2 L^2 \sum_{t=0}^{T-1} \sum_{i \in \mathcal{A}_t} \sum_{k=0}^{K-1} \mathbb{E} \left\| x_i^{(a_{i,t},k)} - x^{(a_{i,t})} \right\|^2$$

$$+ \frac{40\eta_g^2 \eta_l^4 \tau_{\max}^2 K^3 L^2}{N} \sum_{t=0}^{T-1} \mathbb{E} \left\| \sum_{j \in \mathcal{A}_t} \sum_{k=0}^{K-1} \nabla f_j(x_j^{(a_{j,t},k)}) \right\|^2 + 20\eta_g^2 \eta_l^4 \tau_{\max}^2 K^4 T \sigma^2 L^2$$

$$+ 10\eta_l^2 \tau_{\max} K^2 L^2 \sum_{t=0}^{T-1} \sum_{i \in \mathcal{A}_t} \sum_{k=0}^{K-1} \mathbb{E} \left\| x_i^{(a_{i,t},k)} - x^{(a_{i,t})} \right\|^2 + 10\eta_l^2 N K^2 T \sigma^2$$

$$+ 4\eta_l^2 K N \tau_{\max}^2 \sum_{t=0}^{T-1} \mathbb{E} \left\| \sum_{i \in \mathcal{A}_t} \sum_{k=0}^{K-1} \nabla f_i(x_i^{(a_{i,t},k)}) \right\|^2 + 4\eta_l^2 K^2 N^2 \sigma^2 T. \tag{95}$$

Then, for $\eta_l \leq \frac{1}{10\sqrt{\tau_{\max}}KL}$, by rearranging the terms in Equation (95), we have

$$
\sum_{t=0}^{T-1} \sum_{i \in \mathcal{A}_t} \sum_{k=0}^{K-1} \mathbb{E}\left\|x^{(a_{i,t})} - x_i^{(a_{i,t,k})}\right\|^2 \leq 50\eta_l^2 N^2 K^2 \tau_{\max} T\sigma^2 + 40\eta_g^2 \eta_l^4 \tau_{\max}^2 K^4 T\sigma^2 L^2
$$

$$
+ \left(\frac{80\eta_g^2 \eta_l^4 \tau_{\max}^2 K^3 L^2}{N} + 8\eta_l^2 KN\tau_{\max}^2\right) \sum_{t=0}^{T-1} \mathbb{E}\left\|\sum_{j \in \mathcal{A}_t} \sum_{k=0}^{K-1} \nabla f_j(x_j^{(a_{j,t,k})})\right\|^2. \tag{96}
$$

For the third part of Equation (80), we have, by summing over $t$,

$$
3K^2 \sum_{t=0}^{T-1}\left\|\sum_{i \in \mathcal{N}/\mathcal{A}_t} \nabla f_i(x^{(0)})\right\|^2 \leq 3K^2 \sum_{t=0}^{T-1}(N - A_t)\sum_{i=1}^{N}\left\|\nabla f_i(x^{(0)})\right\|^2
$$

$$
= 3K^2 \sum_{t=0}^{T-1}\sum_{j \in \mathcal{N}} \mathbb{1}_{j \notin \mathcal{A}_t}\sum_{i=1}^{N}\left\|\nabla f_i(x^{(0)})\right\|^2 \leq 3K^2 N^2 \tau_{\max} F_0, \tag{97}
$$

where $F_0 := \frac{1}{N}\sum_{i=1}^{N}\left\|\nabla f_i(x^{(0)})\right\|^2$.

Thus, by summing over $t$ and taking expectation in Equation (80), implementing the three parts, we have, for $\eta_l \leq 1/\left(10\sqrt{\tau_{\max}}KL\right)$, by $3NKL^2 \cdot 40\eta_g^2\eta_l^4\tau_{\max}^2 K^4 T\sigma^2 L^4 \leq 2\eta_g^2\eta_l^2 K^3 N\tau_{\max}T\sigma^2 L^2$,

$$
\sum_{t=0}^{T-1} \mathbb{E}\left\|K\sum_{i=1}^{N}\nabla f_i(x^{(t)}) - \sum_{i \in \mathcal{A}_t}\sum_{k=0}^{K-1}\nabla f_i(x_i^{(a_{i,t,k})})\right\|^2
$$

$$
\leq 3K^2 N^2 \tau_{\max}F_0 + 8\eta_g^2\eta_l^2 K^3 N\tau_{\max}\max\{1, \tau_{avg}\}T\sigma^2 L^2 + 6\eta_g^2\eta_l^2 K^2\tau_{\max}^2 L^2\sum_{t=0}^{T-1}\mathbb{E}\left\|\sum_{i \in \mathcal{A}_t}\sum_{k=0}^{K-1}\nabla f_i(x_i^{(a_{i,t,k})})\right\|^2
$$

$$
+ 150\eta_l^2 N^3 K^3\tau_{\max}T\sigma^2 L^2 + \left(240\eta_g^2\eta_l^4\tau_{\max}^2 K^4 L^4 + 24\eta_l^2 K^2 N^2\tau_{\max}^2 L^2\right)\sum_{t=0}^{T-1}\mathbb{E}\left\|\sum_{j \in \mathcal{A}_t}\sum_{k=0}^{K-1}\nabla f_j(x_j^{(a_{j,t,k})})\right\|^2
$$

$$
\leq 3K^2 N^2\tau_{\max}F_0 + 8\eta_g^2\eta_l^2 K^3 N\tau_{\max}\max\{1, \tau_{avg}\}T\sigma^2 L^2 + 150\eta_l^2 N^3 K^3\tau_{\max}T\sigma^2 L^2
$$

$$
+ \left(10\eta_g^2\eta_l^2\tau_{\max}^2 K^2 L^2 + 24\eta_l^2 K^2 N^2\tau_{\max}^2 L^2\right)\sum_{t=0}^{T-1}\mathbb{E}\left\|\sum_{j \in \mathcal{A}_t}\sum_{k=0}^{K-1}\nabla f_j(x_j^{(a_{j,t,k})})\right\|^2. \tag{98}
$$

$\square$

**Lemma E.5.** *Suppose Assumption 1.1 and 1.2 hold. Then, we have*

$$
-\frac{\eta_g\eta_l}{N}\sum_{t=0}^{T-1}\mathbb{E}\left\langle\nabla f(x^{(t)}), \sum_{i \in \mathcal{A}_t}\sum_{k=0}^{K-1}\left[g_i(x_i^{(a_{i,t,k})}, \xi_i^{(a_{i,t,k})}) - \nabla f_i(x_i^{(a_{i,t,k})})\right]\right\rangle
$$

$$
\leq 3\frac{\eta_g^2\eta_l^2 K}{N}\tau_{avg}\sigma^2 LT + 2\frac{\eta_g^2\eta_l^2}{N^2}\tau_{\max}L\sum_{t=0}^{T}\mathbb{E}\left\|\sum_{i \in \mathcal{A}_t}\sum_{k=0}^{K-1}\nabla f_i(x_i^{(a_{i,t,k})})\right\|^2.
$$

*Proof.* The proof is similar to the proof of Lemma D.5. $\square$

**Lemma E.6.** *Suppose Assumption 1.1 and 1.2 hold. Then, for $\eta_l \leq \frac{1}{10\tau_{\max}NKL}$, and $\eta_g\eta_l \leq \frac{1}{10\tau_{\max}KL}$, we have*

$$
\frac{1}{T}\sum_{t=0}^{T-1}\mathbb{E}\left\|\nabla f(x^{(t)})\right\|^2 \leq \frac{2\Delta_f}{\eta_g\eta_l KT} + \frac{9\eta_g\eta_l\max\{1, \tau_{avg}\}\sigma^2 L}{N} + \frac{3\tau_{\max}F_0}{T} + 150\eta_l^2 NK\sigma^2 L^2.
$$

*where $\Delta_f := f(x^{(0)}) - f^*$ and $F_0 := \frac{1}{n}\sum_{i=1}^{n}\left\|\nabla f_i(x^{(0)})\right\|^2$.*

*Proof.* By Assumption 1.1, the function $f := \frac{1}{n} \sum_{i=1}^{n} f_i$ is $L$-smooth. Then, it holds by the descent lemma that

$$\mathbb{E}f(x^{(t+1)}) \leq \mathbb{E}f(x^{(t)}) + \mathbb{E}\left\langle \nabla f(x^{(t)}), x^{(t+1)} - x^{(t)} \right\rangle + \frac{L}{2}\left\| x^{(t+1)} - x^{(t)} \right\|^2. \tag{99}$$

For the inner product term in Equation (99), there holds that

$$
\begin{aligned}
\mathbb{E}\left\langle \nabla f(x^{(t)}), x^{(t+1)} - x^{(t)} \right\rangle &= \mathbb{E}\left\langle \nabla f(x^{(t)}), -\frac{\eta_g \eta_l K}{N} y^{(t)} \right\rangle \\
&= -\frac{\eta_g \eta_l}{N} \mathbb{E}\left\langle \nabla f(x^{(t)}), \sum_{i \in A_t} \sum_{k=0}^{K-1} \left[ g_i(x_i^{(a_{i,t,k})}, \xi_i^{(a_{i,t,k})}) - \nabla f_i(x_i^{(a_{i,t,k})}) \right] \right\rangle \\
&\quad - \frac{\eta_g \eta_l}{N^2} \mathbb{E}\left\langle \sum_{i=1}^{N} \nabla f_i(x^{(t)}), \sum_{i \in A_t} \sum_{k=0}^{K-1} \nabla f_i(x_i^{(a_{i,t,k})}) \right\rangle.
\end{aligned}
\tag{100}
$$

We bound the first term after summing over $t$ in Equation (100) according to Lemma E.5. Then, for the second term in Equation (100), we have

$$
\begin{aligned}
&-\frac{\eta_g \eta_l}{N^2} \mathbb{E}\left\langle \sum_{i=1}^{N} \nabla f_i(x^{(t)}), \sum_{i \in A_t} \sum_{k=0}^{K-1} \nabla f_i(x_i^{(a_{i,t,k})}) \right\rangle \\
&= -\frac{\eta_g \eta_l}{N^2 K} \mathbb{E}\left\langle K \sum_{i=1}^{N} \nabla f_i(x^{(t)}), \sum_{i \in A_t} \sum_{k=0}^{K-1} \nabla f_i(x_i^{(a_{i,t,k})}) \right\rangle \\
&= -\frac{1}{2}\eta_g \eta_l K \mathbb{E}\left\| \nabla f(x^{(t)}) \right\|^2 - \frac{\eta_g \eta_l}{2N^2 K}\mathbb{E}\left\| \sum_{i \in A_t} \sum_{k=0}^{K-1} \nabla f_i(x_i^{(a_{i,t,k})}) \right\|^2 \\
&\quad + \frac{\eta_g \eta_l}{2N^2 K}\mathbb{E}\left\| K \sum_{i=1}^{N} \nabla f_i(x^{(t)}) - \sum_{i \in A_t} \sum_{k=0}^{K-1} \nabla f_i(x_i^{(a_{i,t,k})}) \right\|^2.
\end{aligned}
\tag{101}
$$

Then, summing over $t$ from 0 to $T-1$ on both sides of Equation (99) and taking full expectation, we have

$$
\begin{aligned}
0 \leq &\Delta_f - \frac{\eta_g \eta_l}{N} \sum_{t=0}^{T-1} \mathbb{E}\left\langle \nabla f(x^{(t)}), \sum_{i \in A_t} \sum_{k=0}^{K-1} \left[ g_i(x_i^{(a_{i,t,k})}, \xi_i^{(a_{i,t,k})}) - \nabla f_i(x_i^{(a_{i,t,k})}) \right] \right\rangle \\
&- \frac{1}{2}\eta_g \eta_l K \sum_{t=0}^{T-1} \mathbb{E}\left\| \nabla f(x^{(t)}) \right\|^2 - \frac{\eta_g \eta_l}{2N^2 K} \sum_{t=0}^{T} \mathbb{E}\left\| \sum_{i \in A_t} \sum_{k=0}^{K-1} \nabla f_i(x_i^{(a_{i,t,k})}) \right\|^2 \\
&+ \frac{\eta_g \eta_l}{2N^2 K} \sum_{t=0}^{T-1} \mathbb{E}\left\| K \sum_{i=1}^{N} \nabla f_i(x^{(t)}) - \sum_{i \in A_t} \sum_{k=0}^{K-1} \nabla f_i(x_i^{(a_{i,t,k})}) \right\|^2 + \frac{L}{2} \sum_{t=0}^{T} \mathbb{E}\left\| x^{(t+1)} - x^{(t)} \right\|^2.
\end{aligned}
\tag{102}
$$

where $\Delta_f := f(x^{(0)}) - f^*$. Implementing Lemma E.1 into Equation (102), we have

$$
\begin{aligned}
0 \leq &\Delta_f - \frac{\eta_g \eta_l}{N} \sum_{t=0}^{T-1} \mathbb{E}\left\langle \nabla f(x^{(t)}), \sum_{i \in A_t} \sum_{k=0}^{K-1} \left[ g_i(x_i^{(a_{i,t,k})}, \xi_i^{(a_{i,t,k})}) - \nabla f_i(x_i^{(a_{i,t,k})}) \right] \right\rangle \\
&- \frac{1}{2}\eta_g \eta_l K \sum_{t=0}^{T-1} \mathbb{E}\left\| \nabla f(x^{(t)}) \right\|^2 - \frac{\eta_g \eta_l}{2N^2 K}\left(1 - \eta_g \eta_l K L\right) \sum_{t=0}^{T} \mathbb{E}\left\| \sum_{i \in A_t} \sum_{k=0}^{K-1} \nabla f_i(x_i^{(a_{i,t,k})}) \right\|^2 \\
&+ \frac{\eta_g \eta_l}{2N^2 K} \sum_{t=0}^{T-1} \mathbb{E}\left\| K \sum_{i=1}^{N} \nabla f_i(x^{(t)}) - \sum_{i \in A_t} \sum_{k=0}^{K-1} \nabla f_i(x_i^{(a_{i,t,k})}) \right\|^2 + \frac{\eta_g^2 \eta_l^2 K T \sigma^2 L}{N}.
\end{aligned}
\tag{103}
$$

Implementing Lemma E.5 into Equation (103), we have

$$
\begin{aligned}
0 \leq & \Delta_f - \frac{1}{2}\eta_g\eta_l K \sum_{t=0}^{T-1} \mathbb{E}\left\|\nabla f(x^{(t)})\right\|^2 + \frac{4\eta_g^2\eta_l^2 \max\{1, \tau_{\text{avg}}\} KT\sigma^2 L}{N} \\
& - \frac{\eta_g\eta_l}{2N^2K}\left(1 - \eta_g\eta_l KL - 4\eta_g\eta_l\tau_{\max}KL\right) \sum_{t=0}^{T} \mathbb{E}\left\|\sum_{i\in A_t}\sum_{k=0}^{K-1}\nabla f_i(x_i^{(a_{i,t},k)})\right\|^2 \\
& + \frac{\eta_g\eta_l}{2N^2K}\sum_{t=0}^{T-1}\mathbb{E}\left\|K\sum_{i=1}^{N}\nabla f_i(x^{(t)}) - \sum_{i\in A_t}\sum_{k=0}^{K-1}\nabla f_i(x_i^{(a_{i,t},k)})\right\|^2.
\end{aligned}
\tag{104}
$$

Implementing Lemma E.4 into Equation (104), we have, for $\eta_l \leq \frac{1}{10\sqrt{\tau_{\max}}KL}$,

$$
\begin{aligned}
0 \leq & \Delta_f - \frac{1}{2}\eta_g\eta_l K \sum_{t=0}^{T-1}\mathbb{E}\left\|\nabla f(x^{(t)})\right\|^2 + 75\eta_g\eta_l^3 NK^2\tau_{\max}T\sigma^2 L^2 \\
& - \frac{\eta_g\eta_l}{2N^2K}\left(1 - \eta_g\eta_l KL - 4\eta_g\eta_l\tau_{\max}KL - 10\eta_g^2\eta_l^2\tau_{\max}^2 K^2L^2 - 24\eta_l^2 K^2 N^2\tau_{\max}L^2\right)\cdot \\
& \qquad \sum_{t=0}^{T}\mathbb{E}\left\|\sum_{i\in A_t}\sum_{k=0}^{K-1}\nabla f_i(x_i^{(a_{i,t},k)})\right\|^2 + \frac{3\eta_g\eta_l\tau_{\max}KF_0}{2} \\
& + \frac{4\eta_g^2\eta_l^2\max\{1, \tau_{\text{avg}}\}KT\sigma^2 L}{N} + \frac{3\eta_g^3\eta_l^3\tau_{\max}\max\{1, \tau_{\text{avg}}\}K^2T\sigma^2 L^2}{N}.
\end{aligned}
\tag{105}
$$

Then, for $\eta_l \leq \frac{1}{10\sqrt{\tau_{\max}}NKL}$, and $\eta_g\eta_l \leq \frac{1}{10\tau_{\max}KL}$, we have

$$
\begin{aligned}
& 1 - \eta_g\eta_l KL - 4\eta_g\eta_l\tau_{\max}KL - 10\eta_g^2\eta_l^2\tau_{\max}^2 K^2L^2 - 24\eta_l^2 K^2 N^2\tau_{\max}^2 L^2 \\
& \geq 1 - \frac{1}{10} - \frac{2}{5} - \frac{1}{10} - \frac{6}{25} > 0.
\end{aligned}
\tag{106}
$$

Therefore, for $\eta_l \leq \frac{1}{10\tau_{\max}NKL}$, and $\eta_g\eta_l \leq \frac{1}{10\tau_{\max}KL}$, by rearranging Equation (105) and noticing

$$
\frac{4\eta_g^3\eta_l^3\tau_{\max}\max\{1, \tau_{\text{avg}}\}K^2T\sigma^2 L^2}{N} \leq \frac{\eta_g^2\eta_l^2\max\{1, \tau_{\text{avg}}\}KT\sigma^2 L}{2N},
$$

it implies that

$$
\frac{1}{T}\sum_{t=0}^{T-1}\mathbb{E}\left\|\nabla f(x^{(t)})\right\|^2 \leq \frac{2\Delta_f}{\eta_g\eta_l KT} + \frac{9\eta_g\eta_l\max\{1, \tau_{\text{avg}}\}\sigma^2 L}{N} + \frac{3\tau_{\max}F_0}{T} + 150\eta_l^2 NK\sigma^2 L^2.
\tag{107}
$$

$\square$

**Theorem E.1.** *Suppose Assumption 1.1 and 1.2 hold. Then, for*

$$
\eta_g = \frac{N}{\sqrt{\tau_{\max}}}, \text{ and } \eta_l = \min\left\{\frac{1}{10\sqrt{\tau_{\max}}NKL}, \frac{\sqrt{\tau_{\max}\Delta_f}}{\sqrt{N\max\{1, \tau_{avg}\}KTL\sigma^2}}\right\},
$$

*we have*

$$
\frac{1}{T}\sum_{t=0}^{T-1}\mathbb{E}\left\|\nabla f(x^{(t)})\right\|^2 \leq \frac{30\sqrt{\max\{1, \tau_{avg}\}L\sigma^2\Delta_f}}{\sqrt{NKT}} + \frac{20\tau_{\max}(L\Delta_f + F_0)}{T},
$$

*where $\Delta_f := f(x^{(0)}) - f^*$ and $F_0 := \frac{1}{N}\sum_{i=1}^{N}\left\|\nabla f_i(x^{(0)})\right\|^2$.*

*Proof.* Invoking Lemma D.6, we have, for $\eta_l \leq \frac{1}{10\sqrt{\tau_{\max}}NKL}$, and $\eta_g\eta_l \leq \frac{1}{10\tau_{\max}KL}$,

$$
\frac{1}{T}\sum_{t=0}^{T-1}\mathbb{E}\left\|\nabla f(x^{(t)})\right\|^2 \leq \frac{2\Delta_f}{\eta_g\eta_l KT} + \frac{9\eta_g\eta_l\max\{1, \tau_{\text{avg}}\}\sigma^2 L}{N} + \frac{3\tau_{\max}F_0}{T} + 150\eta_l^2 NK\sigma^2 L^2.
\tag{108}
$$

Choosing

$$\eta_g = \frac{N}{\sqrt{\tau_{\max}}}, \quad \eta_l = \min\left\{\frac{1}{10\sqrt{\tau_{\max}}NKL}, \frac{\sqrt{\tau_{\max}\Delta_f}}{\sqrt{N\max\{1,\tau_{\text{avg}}\}KTL\sigma^2}}\right\},$$

we have

$$\eta_g\eta_l = \min\left\{\frac{\sqrt{N\Delta_f}}{\sqrt{\max\{1,\tau_{\text{avg}}\}KTL\sigma^2}}, \frac{1}{10\tau_{\max}KL}\right\},$$

$$\frac{2\Delta_f}{\eta_g\eta_l KT} \leq \frac{2\sqrt{\max\{1,\tau_{\text{avg}}\}L\sigma^2\Delta_f}}{\sqrt{NKT}} + \frac{20\tau_{\max}L\Delta_f}{T},$$

$$9\frac{\eta_g\eta_l}{N}\max\{1,\tau_{\text{avg}}\}\sigma^2 L \leq \frac{9\sqrt{\max\{1,\tau_{\text{avg}}\}L\sigma^2\Delta_f}}{\sqrt{NKT}},$$

$$150\eta_l^2 NK\sigma^2 L^2 \leq 150\frac{1}{10\sqrt{\tau_{\max}}NKL}\frac{\sqrt{\tau_{\max}\Delta_f}}{\sqrt{N\max\{1,\tau_{\text{avg}}\}KTL\sigma^2}}NK\sigma^2 L^2 \leq \frac{15\sqrt{L\sigma^2\Delta_f}}{\sqrt{NKT}}.$$

Thus, we can combine the above bounds with Equation (108) to get

$$\frac{1}{T}\sum_{t=0}^{T-1}\mathbb{E}\left\|\nabla f(x^{(t)})\right\|^2 \leq \frac{26\sqrt{\max\{1,\tau_{\text{avg}}\}L\sigma^2\Delta_f}}{\sqrt{NKT}} + \frac{20\tau_{\max}(L\Delta_f + F_0)}{T},$$

which gives the desired result by appropriately magnifying the constants. $\qquad\square$

# F   CLIENT PARTICIPATION PATTERNS

In this section, we provide rough estimates of client participation characteristics for the four cases introduced in Section 2. Since both $\tau_{\text{avg}}$ and $\tau_{\max}$ are functions of $\{\mathcal{S}_t\}_{t=0}^{T-1}$, it follows directly from the definition that $\tau_{\text{avg}} \leq \tau_{\max}$. Hence, we focus on bounding $\tau_{\max}$ (or $\mathbb{E}[\tau_{\max}]$) as a rough measure of different client participation patterns.

**Lemma F.1.** *Suppose $\mathcal{S}_t$ are sampled uniformly at random from $\mathcal{N}$ with $|\mathcal{S}_t| = S < N$ for $t = 0,\ldots,T$, as described in Case 1 of Section 2. Then, we have*

$$\mathbb{E}[\tau_{\max}] \leq \frac{4N}{S}\ln(NT).$$

*Proof.* For each client $i \in \mathcal{N}$ at the iteration $t$, define the indicator variable:

$$X_{i,t} := \begin{cases} 1, & \text{if } i \in \mathcal{S}_t, \\ 0, & \text{otherwise.} \end{cases}$$

Then, for each client $i$, define its longest-run random variable as

$$L_i := \max\{l \geq 1 : X_{i,t} = X_{i,t+1} = \cdots = X_{i,t+l-1} = 0 \text{ for some } t\}.$$

Then,

$$\tau_{\max} = \max_{i \in \mathcal{N}}\{L_i\}.$$

By uniformly and randomly sampling $\mathcal{S}_t$ from $\mathcal{N}$, it holds that for fixed $i \in \mathcal{N}$,

$$p := \mathbb{P}(X_{i,t} = 0) = \mathbb{P}(i \notin \mathcal{S}_t) = 1 - \frac{\binom{N-1}{S-1}}{\binom{N}{S}} = 1 - \frac{S}{N}.$$

The event that client $i$ appears in iterations $t, t+1, \cdots, t+l-1$ has probability $p^l$ and there are $T - l + 1$ possible starting points for such a run. Thus, a union bound gives

$$\mathbb{P}(\tau_{\max} \geq l) = \mathbb{P}\left(\cup_{i=1}^N \cup_{t=1}^{T-l+1}\{X_{i,t} = X_{i,t+1} = \cdots = X_{i,t+l-1} = 0\}\right)$$

$$\leq \sum_{i=1}^N \sum_{t=1}^{T-l+1}\mathbb{P}(X_{i,t} = X_{i,t+1} = \cdots = X_{i,t+l-1} = 0) = \sum_{i=1}^N \sum_{t=1}^{T-l+1} p^l \leq NTp^l.$$

Using the tail-sum formular for the expected value, we have

$$\mathbb{E}\tau_{\max} = \sum_{l=1}^{T} \mathbb{P}\left(\tau_{\max} \geq l\right) \leq \sum_{l=1}^{T} \min\left\{1, NTp^l\right\}. \tag{109}$$

Picking integer $m = \lceil \log_p \frac{(1-p)}{NTp} \rceil$, we have

$$\mathbb{E}\tau_{\max} \leq \sum_{l=1}^{m} 1 + \sum_{l=m+1}^{T} NTp^l \leq m + \frac{NTp^{m+1}}{1-p}. \tag{110}$$

Besides, implementing into Equation (110) with the fact that

$$\frac{NTp}{1-p}p^m \leq \frac{NTp}{1-p}p^{\log_p \frac{(1-p)}{NTp}} = \frac{NTp}{1-p}\frac{(1-p)}{NTp} = 1, \text{ and } \frac{p}{1-p} = \frac{N-S}{S} \leq N,$$

we have

$$\mathbb{E}\tau_{\max} \leq \log_p \frac{(1-p)}{NTp} + 1 + 1 = \frac{\ln\left(\frac{1-p}{NTp}\right)}{\ln p} + 2 \leq \frac{4\ln(NT)}{\ln(\frac{1}{p})} \leq \frac{4\ln(NT)}{1-p} = \frac{4N}{S}\ln(NT),$$

which completes the proof. $\qquad\square$

**Lemma F.2.** *Suppose each client independently participates in each round with a fixed probability $p_i \geq \delta \in (0,1]$, as described in Case 2 of Section 2. Then, we have*

$$\mathbb{E}[\tau_{\max}] \leq \frac{4}{\delta}\max\left\{\ln(NT), \ln(\frac{1}{\delta})\right\}.$$

*Proof.* For each client $i \in \mathcal{N}$ at the iteration $t$, define the indicator variable:

$$X_{i,t} := \begin{cases} 1, & \text{if } i \in \mathcal{S}_t, \\ 0, & \text{otherwise.} \end{cases}$$

Then, for each client $i$, define its longest-run random variable as

$$L_i := \max\left\{l \geq 1 : X_{i,t} = X_{i,t+1} = \cdots = X_{i,t+l-1} = 0 \text{ for some } t\right\}.$$

Then,

$$\tau_{\max} = \max_{i \in \mathcal{N}}\left\{L_i\right\}.$$

By the assumption that each client independently participates in each round with a fixed probability $p_i \geq \delta \in (0,1]$, it holds that for any client $i$,

$$p := \mathbb{P}(X_{i,t} = 0) = 1 - p_i \leq 1 - \delta.$$

Using the tail-sum formular for the expected value, we have

$$\mathbb{E}\tau_{\max} = \sum_{l=1}^{T} \mathbb{P}\left(\tau_{\max} \geq l\right) \leq \sum_{l=1}^{T} \min\left\{1, NTp^l\right\}. \tag{111}$$

Picking integer $m = \lceil \log_p \frac{(1-p)}{NTp} \rceil$, we have

$$\mathbb{E}\tau_{\max} \leq \sum_{l=1}^{m} 1 + \sum_{l=m+1}^{T} NTp^l \leq m + \frac{NTp^{m+1}}{1-p}. \tag{112}$$

Besides, implementing into Equation (112) with the fact that

$$\frac{NTp}{1-p}p^m \leq \frac{NTp}{1-p}p^{\log_p \frac{(1-p)}{NTp}} = \frac{NTp}{1-p}\frac{(1-p)}{NTp} = 1, \text{ and } \frac{p}{1-p} \leq \frac{1}{\delta},$$

we have

$$\mathbb{E}\tau_{\max} \leq \log_p \frac{(1-p)}{NTp} + 1 + 1 = \frac{\ln\left(\frac{1-p}{NTp}\right)}{\ln p} + 2 \leq \frac{\ln(NT) + \ln(\frac{1}{\delta})}{\ln(\frac{1}{p})} + 2$$

$$\leq \frac{\ln(NT) + \ln(\frac{1}{\delta})}{1-p} + 2 \leq \frac{\ln(NT)}{\delta} + \frac{\ln\frac{1}{\delta}}{\delta} + 2 \leq \frac{4}{\delta}\max\left\{\ln(NT), \ln(\frac{1}{\delta})\right\},$$

which completes the proof. $\qquad\square$

**Lemma F.3.** *Suppose clients are selected in reshuffled cyclic order, as described in Case 3 of Section 2. Then, we have,*

$$\mathbb{E}\tau_{\max} \leq \frac{4N}{S}.$$

*Proof.* Since the client order is randomly reshuffled at the beginning of each epoch, the interval between two consecutive selections of any client $i$ cannot exceed the rounds spanning from the start of one epoch to the end of the next.

As one epoch consists of $\lfloor N/S \rfloor$ rounds, the gap between two consecutive selections of any client $i$ is at most $2\lfloor N/S \rfloor$. Hence,

$$\tau_{\max} \leq \max_{0 \leq t \leq T-1} \max_{i \in \mathcal{N}} \{ t - a_{i,t} \} \leq 2\lfloor N/S \rfloor \leq \frac{4N}{S}.$$

Taking expectation on both sides yields the desired result. $\qquad\square$

**Lemma F.4.** *Suppose clients are selected in deterministic cyclic order, as described in Case 4 of Section 2. Then, we have,*

$$\tau_{\max} \leq \frac{2N}{S}.$$

*Proof.* Since the client order is deterministic of each epoch, ther interval between two consecutive selections of any client $i$ cannot exceed the rounds spanning from the start of one epoch to the end of it, which implies

$$\tau_{\max} \leq \frac{2N}{S}.$$

$\square$

# G CONVERGENCE RESULTS OF THE FEDSUM FAMILY

In this section, we present the convergence results of the FedSUM family and analyze their behavior under different client participation patterns. Since we take the full expectation over $\tau_{\mathrm{avg}}$ and $\tau_{\max}$, both functions of $\mathcal{S}_t, t = 0, \ldots, T-1$, in Lemmas C.6, D.6, and E.6, it suffices to use their expected values in the final theorem.

Combining Theorem C.1, D.1 and E.1, we obtain the following unified convergence result.

**Theorem G.1.** *Under Assumptions 1.1 and 1.2, and for arbitrary client participation characterized by $\tau_{\max}$ and $\tau_{avg}$, if the learning rates for FedSUM-B, FedSUM, and FedSUM-CR are set as*

$$\eta_g = \frac{N}{\sqrt{\tau_{\max}}}, \text{ and } \eta_l = \min\left\{ \frac{1}{10\sqrt{\tau_{\max}}NKL}, \frac{\sqrt{\tau_{\max}\Delta_f}}{\sqrt{N\max\{1, \tau_{avg}\}KTL\sigma^2}} \right\},$$

*then all three algorithms achieve the following convergence rate:*

$$\frac{1}{T} \sum_{t=0}^{T-1} \mathbb{E}\big[\|\nabla f(x^{(t)})\|^2\big] \leq \frac{30\sqrt{\max\{1, \tau_{avg}\}L\sigma^2\Delta_f}}{\sqrt{NKT}} + \frac{20\tau_{\max}\big(L\Delta_f + F_0\big)}{T}, \qquad (113)$$

*where $\Delta_f := f(x^{(0)}) - f^*$ and $F_0 := \frac{1}{N}\sum_{i=1}^{N}\|\nabla f_i(x^{(0)})\|^2$.*

The following corollary is direct consequences of Lemmas F.1, F.2, F.3, and F.4. By substituting the bounds on $\tau_{\max}$ (and $\tau_{\mathrm{avg}}$ when applicable) from each lemma into Theorem 4.1, we obtain the convergence results of FedSUM-B, FedSUM, and FedSUM-CR under the four client participation schemes described in Section 2.

**Corollary G.1** (Convergence under Case 1–4). *Under Assumptions 1.1 and 1.2, and with appropriately chosen learning rates $\eta_g$ and $\eta_l$, the FedSUM-B, FedSUM, and FedSUM-CR algorithms satisfy the following bounds under the participation schemes in Section 2:*

- *Case 1 (Uniform Random Sampling):*

$$\frac{1}{T}\sum_{t=0}^{T-1}\mathbb{E}\left\|\nabla f(x^{(t)})\right\|^2 \leq \frac{60\sqrt{L\sigma^2\Delta_f\log(NT)}}{\sqrt{SKT}} + \frac{80\big(L\Delta_f + F_0\big)N\log(NT)}{ST}.$$

- *Case 2 (Probability-Based Independent Sampling):*

$$\frac{1}{T}\sum_{t=0}^{T-1}\mathbb{E}\left\|\nabla f(x^{(t)})\right\|^2 \leq \frac{60\sqrt{L\sigma^2\Delta_f}\max\big\{\log(NT),\log(\frac{1}{\delta})\big\}}{\sqrt{\delta NKT}}$$
$$+ \frac{80\big(L\Delta_f + F_0\big)\max\big\{\log(NT),\log(\frac{1}{\delta})\big\}}{\delta T}.$$

- *Case 3 (Reshuffled Cyclic Participation):*

$$\frac{1}{T}\sum_{t=0}^{T-1}\mathbb{E}\left\|\nabla f(x^{(t)})\right\|^2 \leq \frac{60\sqrt{L\sigma^2\Delta_f}}{\sqrt{SKT}} + \frac{80N\big(L\Delta_f + F_0\big)}{ST}.$$

- *Case 4 (Cyclic Participation):*

$$\frac{1}{T}\sum_{t=0}^{T-1}\mathbb{E}\left\|\nabla f(x^{(t)})\right\|^2 \leq \frac{60\sqrt{L\sigma^2\Delta_f}}{\sqrt{SKT}} + \frac{80N\big(L\Delta_f + F_0\big)}{ST}.$$

## H   NUMERICAL EXPERIMENTS

### H.1   CODE

The code for reproducing our experiments is available at `https://anonymous.4open.science/r/FedSUM-0658`.

### H.2   EXPERIMENTAL SETUPS

**Hardware and software Setups.**

- Hardware. The experiments are performed on a private cluster with eight Nvidia RTX 3090 GPU cards.
- Software. We code the experiments based on Pytorch 2.0.1 and Python 3.11.4.

**Neural network and hyper-parameter specifications.**

Table 2 details the models and training setup. The initial local learning rate $\eta_0$ and global learning rate $\eta_g$ are optimized over a grid search, with $\eta_0 \in \{0.01, 0.005, 0.001, 0.0005\}$ and $\eta_0\eta_g = 0.01$, based on the best performance after 500 global rounds of FedAvg. Consequently, we select $\eta_g = 1.0$ and $\eta_0 = 0.01$ for all algorithms across various models and datasets. Furthermore, we choose $K = 50$ in FedAU as suggested in the original paper (Wang & Ji, 2023).

**Datasets and data heterogeneity.**

All the datasets we evaluate contain 10 classes of images, detailed as follows.

- **MNIST (LeCun et al., 2010).** The dataset contains $28 \times 28$ grayscale images of 10 different handwritten digits. In total, there are 60000 train images and 10000 test images.
- **SVHN (Netzer et al., 2011).** The dataset contains $32 \times 32$ colored images of 10 different number digits. In total, there are 73257 train images and 26032 test images.
- **CIFAR-10 (Krizhevsky et al., 2009).** The dataset contains $32 \times 32$ colored images of 10 different objects. In total, there are 50000 train images and 10000 test images.

Figure 4 shows the data distribution across 100 clients in training over MNIST. The x-axis represents the client index, and the y-axis the number of data samples per client. The color bars in each histogram show the proportions of different labels. The Dirichlet parameter $\alpha = 0.1$ controls data heterogeneity: smaller $\alpha$ values lead to more non-i.i.d. distributions, while larger $\alpha$ values result in more homogeneous data.

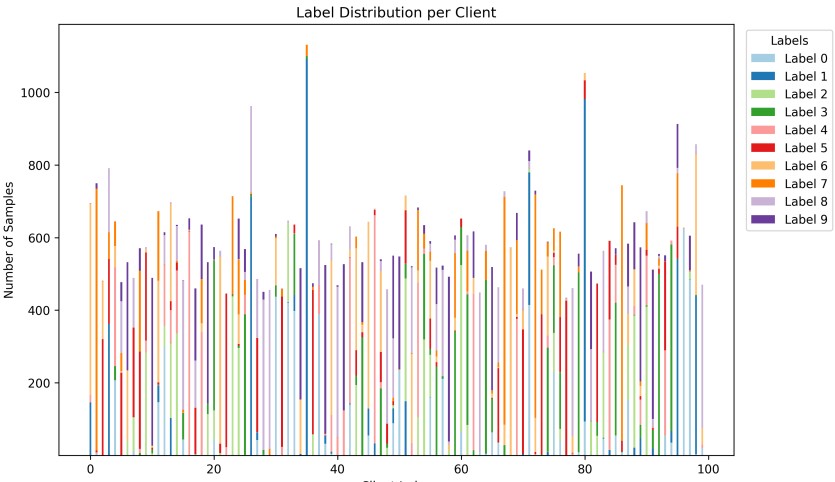

Figure 4: Data heterogeneity with Dirichlet($\alpha = 0.1$) distribution across 100 clients. The x-axis is the client index, and the y-axis is the number of samples. The color bars represent the proportion of each label. Smaller $\alpha$ leads to more non-i.i.d. data.

**Client participation patterns.**

As described in Section 2, we consider three client participation patterns:

- **P1 (Uniform random sampling):** At each round $t$, $S = 20$ clients are randomly and uniformly selected to participate.

- **P2 (Stationary probability participation):** At each round $t$, each client has a participation probability of $\frac{S}{N}$, where $S = 20$, $N = 100$.

- **P3 (Non-stationary with sine trajectory):** At each round $t$, the participation probability of each client is $p_i^t := \frac{S}{N}\left(0.3sin(\frac{\pi t}{5}) + 0.7\right)$, where $S = 20$, $N = 100$.

### H.3 DETAILED EXPERIMENTAL RESULTS

To provide a more granular analysis of the experimental results presented in Section 5 of the main paper, this part offers more detailed performance comparisons. Specifically, Figures 5, 6, and 7 isolate the performance of the evaluated algorithms under each of the three client participation patterns (P1, P2, and P3), respectively. This allows for a clearer assessment of how each algorithm responds to different participation schemes. Furthermore, to offer a clearer view of the final convergence behavior, Figure 8 presents the training dynamics over the last 160 rounds.

### H.4 ADDITIONAL EXPERIMENTAL RESULTS

We present the performance of FedSUM across all the client participation patterns outlined in Section 2, specifically for Cases 1 to 4. The results shown in Figure 9 confirm the robust performance of FedSUM under different patterns.

In addition, we illustrate the performance of FedSUM evaluated under the client participation pattern Case 2 (Probability-Based Independent Participation, with varying probabilities $p$), as shown in Figure 10. The results imply that larger $p$ (and thus smaller $\tau_{\text{avg}}$ and $\tau_{\text{max}}$) generally leads to better performance, which agrees with the theoretical findings.

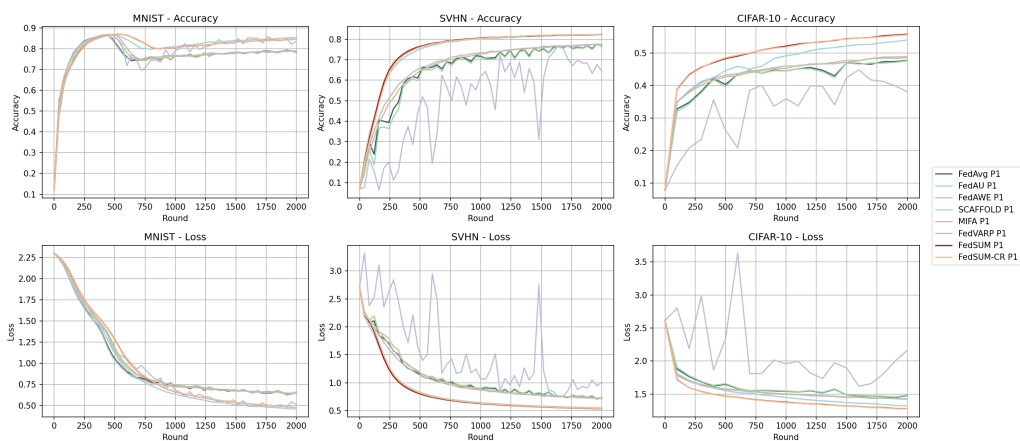

Figure 5: Performance of the evaluated algorithms using CNN models on three datasets under client participation pattern P1.

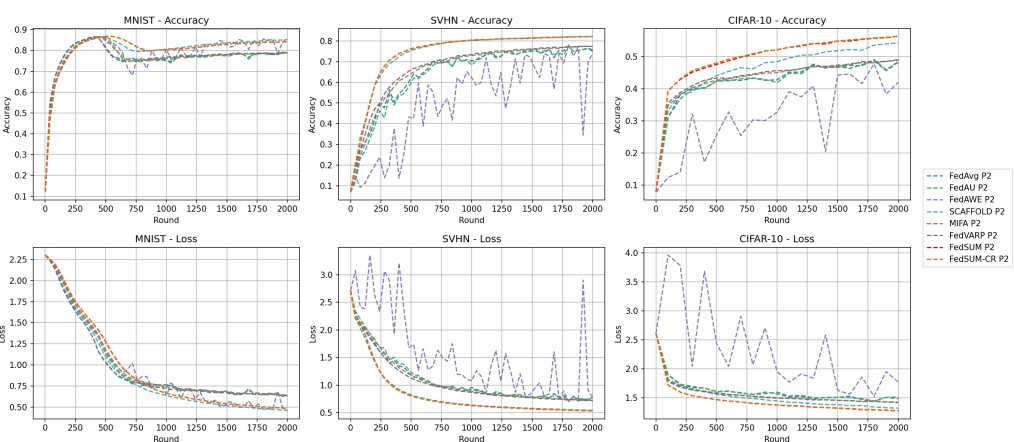

Figure 6: Performance of the evaluated algorithms using CNN models on three datasets under client participation pattern P2.

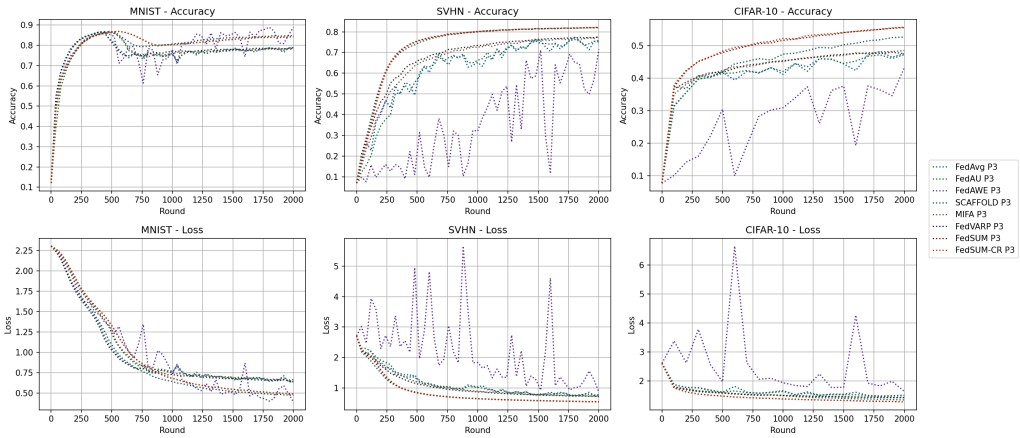

Figure 7: Performance of the evaluated algorithms using CNN models on three datasets under client participation pattern P3.

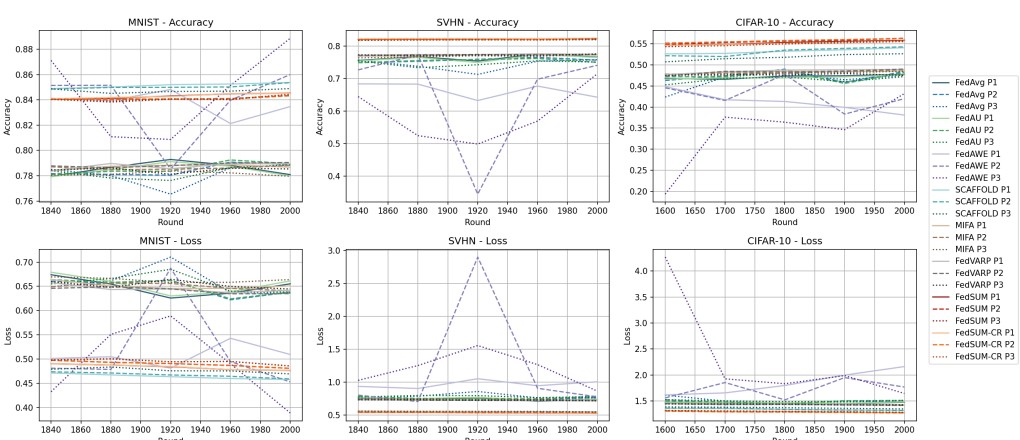

Figure 8: Detailed view of convergence behavior during the final 160 rounds of training.

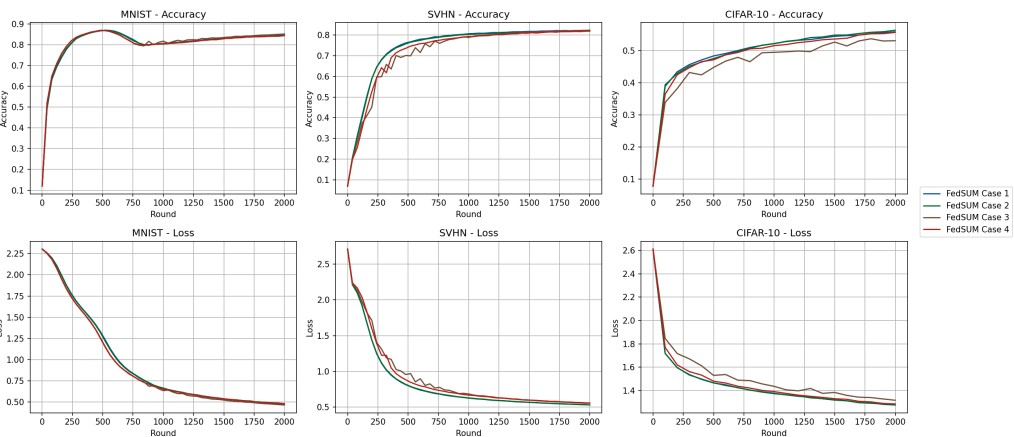

Figure 9: Performance of the FedSUM algorithm using CNN models on three datasets, evaluated across client participation patterns (Case 1 to Case 4) as described in Section 2.

Table 2: Neural network architectures, loss function, learning rate scheduling, training steps and batch size specifications.

| Datasets | MNIST | SVHN | CIFAR-10 |
|---|---|---|---|
| Neural network | CNN | CNN | CNN |
| Model architecture | C(1,10) - R - M - C(10,20) - D - R - M - L(50) - R - D - L(10) | C(3,32)-R -M- C(32,32)-R-M -L(128)-R-L(10) | C(3,32)-R -M- C(32,32)-R-M -L(256)-R-L(64) -R-L(10) |
| Loss function | Cross-entropy loss | | |
| Local learning rate $\eta_l$ Scheduling | $\eta_l = \frac{\eta_0}{\sqrt{t/10+1}}$, where $t$ denotes the global round. | | |
| Number of clients $N$ | 100 | | |
| Number of local updates $K$ | 10 | | |
| Number of global rounds $T$ | 2000 | | |
| Batch size | 128 | | |

* C(# in-channel, # out-channel): a 2D convolution layer (kernel size 3, stride 1, padding 1); R: ReLU activation function; M: a 2D max-pool layer (kernel size 2, stride 2); L: (# outputs): a fully-connected linear layer; D: a dropout layer (probability 0.2).

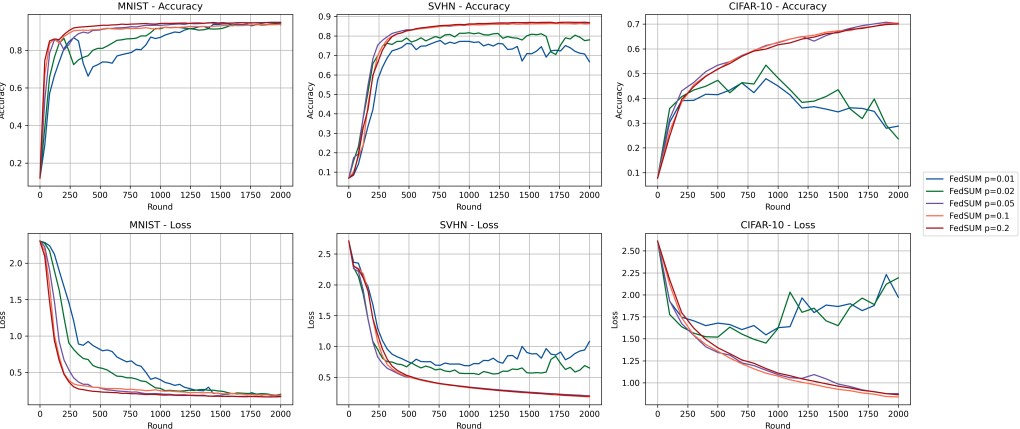

Figure 10: Performance of the FedSUM algorithm using CNN models on three datasets with data heterogeneity ($\alpha = 0.1$) and constant learning rates $\eta_g = 1.0$ and $\eta_l = 0.01$, evaluated under the client participation pattern Case 2 (Probability-Based Independent Participation, with varying probabilities $p$).

## H.5 COMPARISON AMONG THE FEDSUM FAMILY

Figure 11 further demonstrates that FedSUM-B and FedSUM-CR achieve performance comparable to or even better than FedSUM when further setting the constant and bigger local learning rate $\eta_l = 0.1$. This is mainly because the batch size is set to 128, which is relatively large given that each client has only about 600 data samples. Under a smaller batch size (e.g., 8 or 16), the performance of FedSUM-B degrades while FedSUM takes the lead.

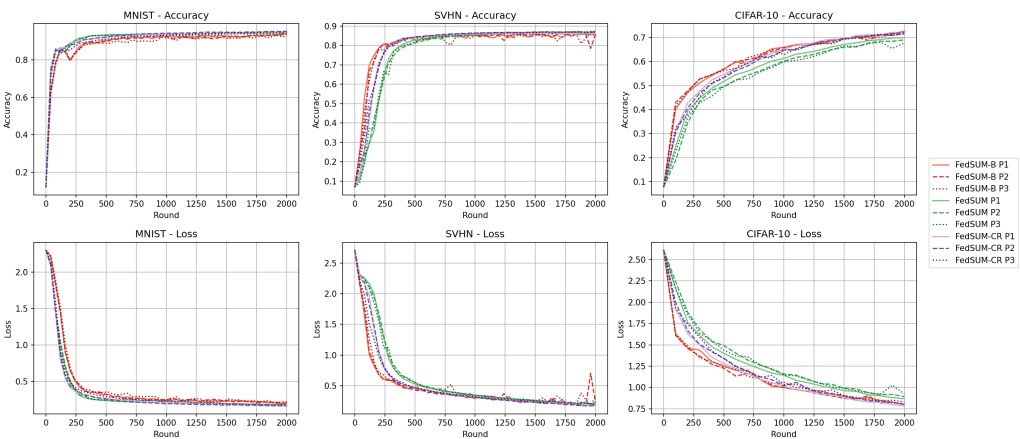

Figure 11: Training loss and test accuracy curves for CNN models trained using FedSUM family on three datasets under different client participation patterns.

## H.6 ADDITIONAL EXPERIMENT: COMPARISON OF FEDSUM WITH FEDAVG AND SCAFFOLD ON TRAINING RESNET-18

We conduct additional experiments by training a ResNet-18 model on the CIFAR-10 and CIFAR-100 datasets to further evaluate the performance of our proposed methods.

In both experiments, we simulate a federated environment with 50 total clients, from which 10 are randomly selected in each communication round. To model data heterogeneity, the training data is partitioned among clients using a Dirichlet distribution with $\alpha = 0.1$. The optimization and training parameters are configured as follows:

- **Optimizer:** We use SGD without weight decay. The global learning rate is set to $\eta_g = 1.0$.

- **CIFAR-10:** Each client performs 10 local updates with a batch size of 32 and a local learning rate of $\eta_l = 0.01$.

- **CIFAR-100:** Each client performs 10 local updates with a batch size of 32 and a local learning rate of $\eta_l = 0.001$.

The results, showing the training loss and test accuracy curves for these experiments, are presented in Figure 12a and Figure 12b. It can be seen that FedSUM achieves the strongest performance in both cases.

## H.7 ADDITIONAL EXPERIMENT: COMPARISON OF FEDSUM WITH FEDAVG ON SST-2 DATASET

We further evaluate the performance of the proposed FedSUM algorithm against the standard Federated Averaging (FedAvg) baseline (McMahan et al., 2017) on the Stanford Sentiment Treebank (SST-2) dataset (Socher et al., 2013), a benchmark for binary sentiment classification.

The base model for the experiment is "bert-base-uncased" (Devlin et al., 2019), which we fine-tune for the downstream task. We utilize the pre-trained weights and implementation from the Hugging Face transformers library (Wolf et al., 2020) and then append a single fully-connected linear layer which serves as the classification head. To simulate a federated environment, we partition the SST-2 training set into 8 non-overlapping, equal-sized subsets, creating an I.I.D. data distribution across the clients.

During the federated training process, 2 clients are randomly sampled to participate in each communication round. Each selected client performs 3 local updates of training on its data partition with a batch size of 4. We use SGD optimizer without weight decay, setting the local learning rate to

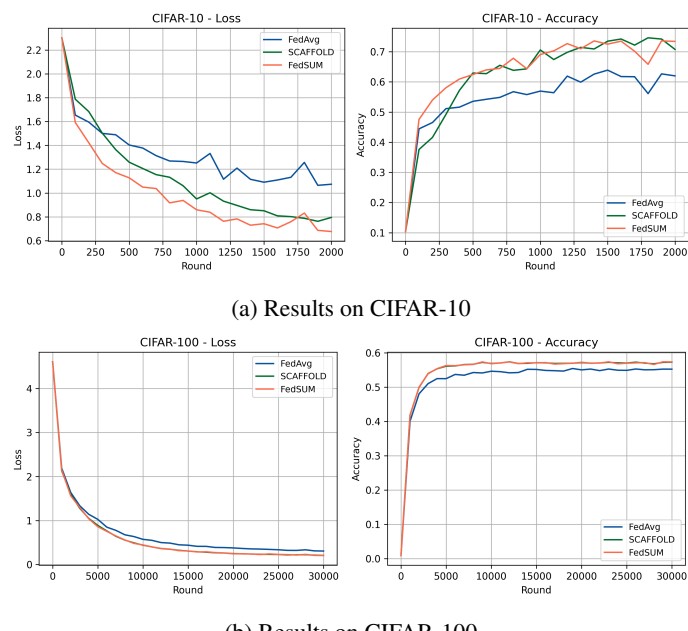

(a) Results on CIFAR-10

(b) Results on CIFAR-100

Figure 12: Training loss and test accuracy for a ResNet-18 model on CIFAR-10 and CIFAR-100. For both datasets, data is partitioned heterogeneously among 50 clients using a Dirichlet distribution ($\alpha = 0.1$).

$\eta_l = 1 \times 10^{-4}$ and the global learning rate to $\eta_g = 1.0$. This setup allows for a direct comparison of the convergence properties and final accuracy of the two algorithms under a controlled IID setting.

The results of this experiment, shown in Figure 13, compare the loss and accuracy curves for the two algorithms: FedAvg and FedSUM. The FedSUM algorithm demonstrates competitive performance in terms of both loss reduction and accuracy improvement compared to FedAvg. Note that both SCAFFOLD (Karimireddy et al., 2020) and FedSUM-CR do not converge in this experiment and thus the corresponding results are not depicted.

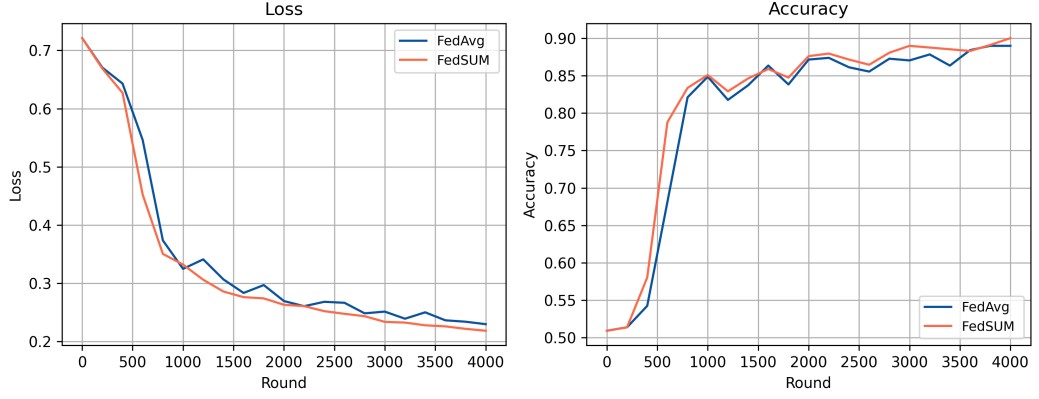

Figure 13: Comparison of FedSUM and FedAvg on SST-2 dataset using BERT model. The left plot shows the loss curve, while the right plot shows the accuracy curve for each algorithm over 4000 rounds.

## H.8 Additional Experiment: Influence of Biased Sampling

We further evaluate the performance of the proposed FedSUM algorithm against the standard Federated Averaging (FedAvg) baseline (McMahan et al., 2017) under a biased sampling client participation pattern.

In this biased sampling scenario, clients are assigned different probabilities of participation based on their indices. Specifically, clients 1 to 11 have a 0.5 probability of participating, clients 12 to 22 have a 0.45 probability, clients 23 to 33 have a 0.4 probability, and so on, with the probability decreasing as client indices increase.

The results in Figures 14a and 14b indicate that the bias issue discussed in Ribero et al. (2022); Sun et al. (2024) does not affect the performance of FedSUM, which is consistent with the benefits of the Stochastic Uplink-Merge (SUM) technique outlined in Appendix A.

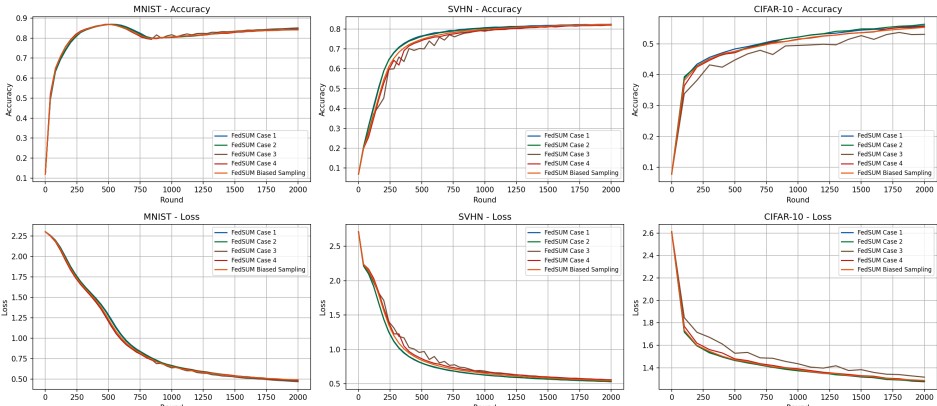

(a) Performance comparison of the FedSUM algorithm using CNN models across different client participation patterns (Case 1 to Case 4), including the biased sampling case.

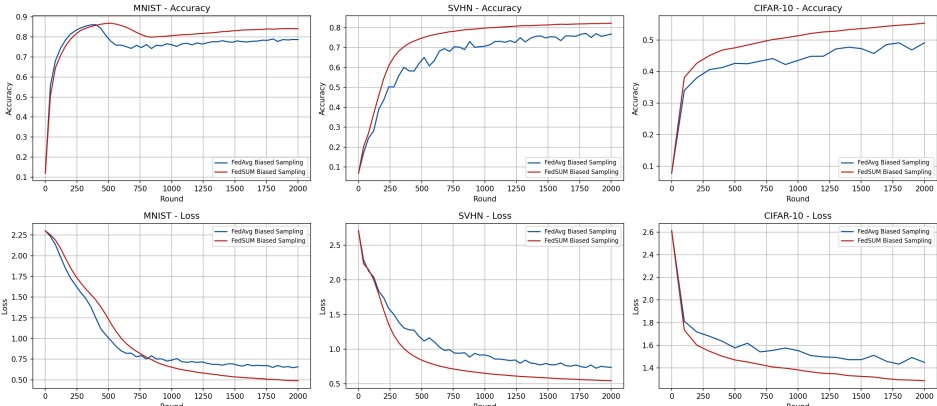

(b) Performance comparison between the FedSUM and FedAvg algorithms using CNN models on three datasets, evaluated under the biased sampling case.

Figure 14: Influence of the biased sampling case.

## I    LLM USAGE

In preparing this manuscript, we made limited use of Large Language Models (LLMs) solely for minor text polishing. The LLM was used only to improve grammar, clarity, and readability. All conceptual development, theoretical analysis, experimental design, and interpretation of results were conducted entirely by the authors, and the scientific content is the authors' original work.

