# OpenReview forum: "FedSUM Family: Efficient Federated Learning Methods under Arbitrary Client Participation"
_ICLR.cc/2026/Conference — Submitted to ICLR 2026_

### Official Review · Reviewer_Z1o5 · 2025-10-27

**Soundness:** 3
**Presentation:** 3
**Contribution:** 3
**Rating:** 6
**Confidence:** 3

**Summary:**

This paper proposes the FedSUM family designed to handle arbitrary client participation patterns without additional assumptions on data heterogeneity. The authors introduce two delay metrics, $\tau_{max}$ and $\tau_{avg}$, to quantify client availability variability and incorporate them into a unified convergence analysis. The theoretical results show that FedSUM achieves convergence rates comparable to or better than existing methods under diverse participation scenarios. Empirical experiments on MNIST, SVHN, and CIFAR-10 show that FedSUM variants outperform several baselines in both convergence speed and stability.

**Strengths:**

1. This paper is very theoretical, and it provides a unified treatment of arbitrary client participation, bridging several prior assumptions into one analytical framework.

2. The three FedSUM variants balance communication cost and performance trade-offs well.

3. The most related baselines are almost covered in this paper. Compared with the baselines, FedSUM shows better performance under different client participation patterns.

**Weaknesses:**

W1. While the paper claims that FedSUM-CR achieves reduced communication cost, it lacks quantitative analysis or empirical validation of this claim. The paper would be much stronger if it included explicit communication-per-round cost comparisons or more specific theoretical communication complexity analysis.

W2. A very related work should be included and compared in this paper. See Q1 below.

Minor:

Line 117: "objective *funtions*" -> "objective *functions*"

**Questions:**

Q1. What is the connection between FedSUM and FOCUS [1] algorithms (these two may be equivalent to some degree)?

Q2. For algorithms with the correction step, why is the bounded gradient heterogeneity still needed? (Scaffold or FOCUS [1] do not need that)

Q3. For uniform sampling, what is $\tau_{max}$?

[1] Ying, B., Li, Z., & Yang, H. (2025). Exact and Linear Convergence for Federated Learning under Arbitrary Client Participation is Attainable. arXiv preprint arXiv:2503.20117.

See weaknesses above.

---

> ### Author Response · Authors · 2025-11-23
>
> Thank you for your valuable comments. We address your concerns and questions below.
>
> - **Connection between FedSUM and FOCUS.** We notice that the Stochastic Uplink-Merge technique in FedSUM shares similarity with FOCUS, which allows the server to aggregate the (stochastic) gradients of the clients to obtain the update direction. This design mitigates the effect of data heterogeneity and also addesses the bias issue mentioned by Reviewer 7geH. We have included FOCUS in Section 3.1 when discussing the benefits of Stochastic Uplink-Merge.
>
>     However, the two algorithms are not equivalent from our observation. For instance, FedSUM introduces correction direction $y_i$ which incorporates global information from the server to update $x_i$, while in FOCUS, client $i$ does not receive such information from the server, and the update direction of $x_i$ depends only on local information (see line 7-10 in Algorithm 1 of [1]). The other settings, including the participation pattern, objective functions, stochastic gradients, are also different.
>
>     [1] Ying, B., Li, Z., & Yang, H. (2025). Exact and Linear Convergence for Federated Learning under Arbitrary Client Participation is Attainable. arXiv preprint arXiv:2503.20117.
>
> - **Bounded gradient heterogeneity.** FedSUM does not require the bounded gradient heterogeneity assumption, just like SCAFFOLD and FOCUS.
>
> - **$\tau_{\max}$ in uniform sampling.** As stated in Theorem 4.1 and Corollary 4.1 in Section 4, we characterize the average performance of the FedSUM family in terms of the expectations of $\tau_{\max}$ and $\tau_{\text{avg}}$ when client participation involves randomness. For uniform sampling, as described in Case 1 of Section 2.1, we have $\mathbb{E}[\tau_{\max}] \le \frac{N}{S} \log (NT)$, with the derivation provided in Lemma F.1 in Appendix F.
>
> In the latest version of the paper, we have adjusted the coefficients in the algorithm to ensure a fairer comparison among the algorithms. Specifically, we re-weight the local update direction by dividing by $N$, the number of clients, as shown in Line 9, Line 11, and Line 12 of both FedSUM and FedSUM-CR and reproduce all the experiments. This adjustment ensures that each local update has a similar scaling for the update direction, comparable to that of other algorithms. This change is equivalent to the original version, where we can derive the original versions by picking $\eta_l \leftarrow N\eta_l$ and $\eta_g \leftarrow \eta_g/N$ in the updated version.
>
> In the revision, we have further tested the performance of different algorithms by training ResNet-18 on the CIFAR-10 and CIFAR-100 datasets with 50 clients, as shown in Figure 12a and Figure 12b in Appendix H.5, respectively. Additionally, we fine-tuned a BERT model on the SST-2 dataset, as depicted in Appendix H.6, with the performance results shown in Figure 13 of Appendix H.6. Through the additional experiments, we show that the FedSUM family either converges faster or achieves competitive performance compared to all the related algorithms.

---

### Official Review · Reviewer_ew4X · 2025-10-30

**Soundness:** 3
**Presentation:** 3
**Contribution:** 2
**Rating:** 4
**Confidence:** 3

**Summary:**

The paper addresses the problem of arbitrary and unpredictable client participation in Federated Learning (FL). The core contribution is the FedSUM family of algorithms, including FedSUM-B, FedSUM, and FedSUM-CR. They leverage a Stochastic Uplink-Merge update framework to maintain meaningful global progress even when clients participate infrequently or irregularly. The paper provides unified convergence guarantees and shows that FedSUM variants either match or outperform prior methods under diverse participation patterns by introducing two participation variability metrics: maximum delay ($\tau_{max}$) and average delay ($\tau_{avg}$).

**Strengths:**

1. The arbitrary participation setting, characterized maximum delay ($\tau_{max}$) and average delay ($\tau_{avg}$), captures a wide range of realistic client behaviors, which significantly broadens applicability beyond common uniform sampling frameworks.

2. The convergence guarantees are presented in a unified manner for all three FedSUM variants, recovering known results in canonical special cases.

**Weaknesses:**

1. Datasets are small to medium scale (MNIST, SVHN, CIFAR-10).

2. Client participation patterns in experiment does not perfectly align with the four participation examples.

**Questions:**

1. What are the maximum delay ($\tau_{max}$) and average delay ($\tau_{avg}$) in practice or in experiment?

2. I am curious about the performance of sparse participation regimes. For example, in cross-device federated learning setting, there are so many clients and very small fraction of clients can participate in each round. This means $\tau_{max}$ and $\tau_{avg}$ would be very large. I suppose FedAvg works well in this case but not sure about the proposed methods.

3. I wonder if Case 4: Reshuffled Cyclic Participation corresponds to client-reshuffling-based FL (Malinovsky, Grigory, et al. "Federated learning with regularized client participation." arXiv preprint arXiv:2302.03662 (2023).).

---

> ### Author Response · Authors · 2025-11-23
>
> Thank you for your valuable comments. We address your concerns and questions below.
>
> - **Maximum and average delays.** The maximum and average delays, or their expectations when client participation involves randomness, can be determined or estimated under specific participation patterns. For example, in Case 1 (Uniform Random Sampling), we have $\mathbb{E}[\tau_{\max}] \le \frac{N}{S} \log (NT)$. In Case 2 (Probability-Based Independent Participation), we have $\mathbb{E}[\tau_{\max}] \le \frac{1}{\delta} \log (NT)$. In Case 3 (Deterministic Cyclic Participation), we have $\tau_{\max} \le \frac{N}{S}$. In Case 4 (Reshuffled Cyclic Participation), we have $\mathbb{E}[\tau_{\max}] \le \frac{N}{S}$. The derivations are provided in Lemmas F.1, F.2, F.3, and F.4 in Appendix F.
>
> - **Sparse participation regimes.** FedSUM works well in sparse participation regimes. Theoretically, under commonly considered participation patterns (Case 1 to 4 introduced in Section 2.1), the performance of FedSUM is comparable with the state-of-the-art results. For example, assuming $S$ out of $N$ clients are uniformly randomly sampled at every iteration, then the convergence rate of FedSUM becomes $\tilde{O}(\frac{1}{\sqrt{SKT}})$ in the dominant term, which slows down when $S$ decreases. This is similar to the performance of FedAvg or SCAFFOLD. In the experiments, we further consider probability-independent participation (i.e., Case 2) with varying values of $p$, as shown in Figure 10. It can be observed that FedSUM converges well even in sparse settings, for instance when $p = 0.05$. For $p = 0.01$, the chosen learning rate is too aggressive, and using smaller learning rates can lead to improved convergence.
>
> - **Case 4. Reshuffled Cyclic Participation.** Case 4 is indeed considered in the referenced client-reshuffling-based FL paper (Malinovsky, Grigory, et al. "Federated learning with regularized client participation." arXiv preprint arXiv:2302.03662 (2023)), where both the clients and their associated data are reshuffled, and the authors further develop the theoretical analysis under the assumption of strong convexity. We have cited this paper in the revised version (Section 2.1).
>
> In the latest version of the paper, we have adjusted the coefficients in the algorithm to ensure a fairer comparison among the algorithms. Specifically, we re-weight the local update direction by dividing by $N$, the number of clients, as shown in Line 9, Line 11, and Line 12 of both FedSUM and FedSUM-CR and reproduce all the experiments. This adjustment ensures that each local update has a similar scaling for the update direction, comparable to that of other algorithms. This change is equivalent to the original version, where we can derive the original versions by picking $\eta_l \leftarrow N\eta_l$ and $\eta_g \leftarrow \eta_g/N$ in the updated version.
>
> In the revision, we have further tested the performance of different algorithms by training ResNet-18 on the CIFAR-10 and CIFAR-100 datasets with 50 clients, as shown in Figure 12a and Figure 12b in Appendix H.5, respectively. Additionally, we fine-tuned a BERT model on the SST-2 dataset, as depicted in Appendix H.6, with the performance results shown in Figure 13 of Appendix H.6. Through the additional experiments, we show that the FedSUM family either converges faster or achieves competitive performance compared to all the related algorithms.

---

### Official Review · Reviewer_7duF · 2025-11-01

**Soundness:** 3
**Presentation:** 2
**Contribution:** 3
**Rating:** 4
**Confidence:** 4

**Summary:**

This paper proposes FedSUM, a family of three federated-learning algorithms (i.e., FedSUM-B, FedSUM, and FedSUM-CR), that are robust to arbitrary client participation patterns, including both controlled and uncontrolled, stochastic and deterministic, and homogeneous and heterogeneous schemes. The key idea is to capture participation variability through two delay metrics: maximum delay $\tau_{max}$ and average delay $\tau_{avg}$. These metrics enable a unified convergence analysis for the FedSUM algorithms without requiring restrictive assumptions on data heterogeneity.
The algorithms employ a Stochastic Uplink-Merge technique to address data heterogeneity among clients. Theoretical convergence bounds are provided under standard smoothness and bounded variance assumptions. Experiments on MNIST, SVHN, and CIFAR-10 under various participation patterns validate the effectiveness of FedSUM compared to the baseline algorithms.

**Strengths:**

1. The use of maximum delay ($\tau_{max}$) and average delay ($\tau_{avg}$) provides an elegant and intuitive way to quantify participation variability and enables a unified convergence analysis for the FedSUM algorithms.
2. The FedSUM family subsumes many client participation scenarios (random, cyclic, reshuffled, etc.) under a single analytical umbrella, offering broad applicability.
3. The convergence rates recover the known rates in special cases and are derived under standard assumptions (smoothness, bounded variance), avoiding unrealistic restrictions on data heterogeneity.

**Weaknesses:**

1. While the delay metrics are central to the theoretical framework, empirical results do not explicitly demonstrate how varying $\tau_{max}$ or $\tau_{avg}$ impacts performance or convergence speed. An ablation study varying delay levels across synthetic participation schemes would be beneficial.
2. While FedSUM-CR emphasizes communication savings and acknowledges additional memory overhead, neither the communication cost nor the memory overhead is quantitatively analyzed.
3. The evaluation is limited to vision tasks (MNIST, SVHN, CIFAR-10) using a single model architecture (CNN). Incorporating non-vision tasks (e.g., NLP) and additional model types (e.g., ResNet) would provide stronger evidence of the framework's generalizability.
4. The figures contain too many lines in a single plot, making them difficult to distinguish.

**Questions:**

Please refer to the Weaknesses.

---

> ### Author Response · Authors · 2025-11-23
>
> Thank you for your valuable comments. We address your concerns and questions below.
>
> - **Influence of varying delays.** To study the influence of carying delays on the algorithmic performance, in the revised version we have further considered probability-independent participation (i.e., Case 2) with varying values of $p$, as shown in Figure 10.
> It can be seen that larger $p$ (and thus smaller $\tau_{\text{avg}}$ and $\tau_{\max}$) generally leads to better performance, which agrees with the theoretical findings.
>
> - **Communication savings in FedSUM-CR.** The communication savings in FedSUM-CR are evident from its performance comparison with FedSUM in Figure 1b in the revision, which depicts the performance relative to the number of communicated variables, clearly highlighting the advantage of FedSUM-CR. Notably, the iteration-wise convergence results are comparable, with FedSUM-CR saving $1/3$ of the communication workload per iteration. However, it does incur an additional model-sized memory overhead compared to FedSUM.
>
> - **Enhanced experiments.** We have further tested the performance of FedSUM against competing algorithms by training ResNet-18 on the CIFAR-10 and CIFAR-100 datasets with 50 clients, as shown in Figure 12a and Figure 12b in Appendix H.5, respectively. Additionally, we fine-tuned a BERT model on the SST-2 dataset, as depicted in Appendix H.6, with the performance results shown in Figure 13 of Appendix H.6. Through the additional experiments, we show that FedSUM either converges faster or achieves competitive performance compared to all the related algorithms.
>
> - **Refined figures.** Thank you for your suggestions regarding the lines in the figures. First, we have changed the color of the lines in Figures 1a and 1b to make them more distinguishable. Additionally, we have plotted the comparison of the algorithms with respect to different participation patterns in Figures 5, 6, and 7, respectively. Furthermore, we have included the last 160 rounds of these algorithms in Figure 8 to provide a clearer comparison of the final stages.
>
> In the latest version of the paper, we have adjusted the coefficients in the algorithm to ensure a fairer comparison among the algorithms. Specifically, we re-weight the local update direction by dividing by $N$, the number of clients, as shown in Line 9, Line 11, and Line 12 of both FedSUM and FedSUM-CR and reproduce all the experiments. This adjustment ensures that each local update has a similar scaling for the update direction, comparable to that of other algorithms. This change is equivalent to the original version, where we can derive the original versions by picking $\eta_l \leftarrow N\eta_l$ and $\eta_g \leftarrow \eta_g/N$ in the updated version.

---

### Official Review · Reviewer_7geH · 2025-11-04

**Soundness:** 3
**Presentation:** 2
**Contribution:** 3
**Rating:** 4
**Confidence:** 4

**Summary:**

This paper proposes the FedSUM family of federated learning (FL) algorithms, addressing the challenge of arbitrary client participation, a common issue in real-world FL systems where clients may drop out or join unpredictably. A family of algorithm, FedSUM, is proposed followed by convergence analysis. Numerical simulations are provided to illustrate the performance of proposed algorithms.

**Strengths:**

1. By unifying several participation schemes into a single theoretical lens, the FedSUM family provides a flexible and general approach.

2. It (partially) addresses the problem of unpredictable participation from a theoretical perspective, bridging the gap between practical and theoretical studies of FL.

**Weaknesses:**

1. Evaluations are limited to standard vision datasets (MNIST, SVHN, CIFAR-10). No experiments on larger or non-IID real-world datasets (e.g., cross-device or cross-silo FL). Adding more realistic experiments would significantly improve the soundness of the paper.

2. The unified framework seems fail to capture time-dependent client participation. For example, shown in (Ribero et al., 2022, Sun et al, 2025), the participation pattern is Markovian, which do not fit in the proposed framework. Moreover, the bias issue discussed in (Ribero et al., 2022, Sun et al, 2025) is worthy to be analyzed and compared under the proposed framework.


[1]. Ribero, Mónica, Haris Vikalo, and Gustavo De Veciana. "Federated learning under intermittent client availability and time-varying communication constraints." IEEE Journal of Selected Topics in Signal Processing 17.1 (2022): 98-111.

[2]. Sun, Zhenyu, et al. "Debiasing Federated Learning with Correlated Client Participation." The Thirteenth International Conference on Learning Representations, 2025.

**Questions:**

1. Could authors add more experiments under more practical settings?

2. Could authors discuss whether the proposed framework can fit time-related participation? And how to address bias issue?

---

> ### Author Response · Authors · 2025-11-23
>
> Thank you for your valuable comments. We address your concerns and questions below.
>
> - **Enhanced experiments.** We have further tested the performance by training ResNet-18 on the CIFAR-10 and CIFAR-100 datasets with 50 clients, as shown in Figure 12a and Figure 12b in Appendix H.5, respectively. Additionally, we fine-tuned a BERT model on the SST-2 dataset, as depicted in Appendix H.6, with the performance results shown in Figure 13 of Appendix H.6. Through the additional experiments, we show that the FedSUM family either converges faster or achieves competitive performance compared to all the related algorithms.
>
> - **Time-related participation.** Since the considered participation sequence $\{ \mathcal{S}\_t , t=0,..., T-1\}$ is arbitrary, it also includes sequences generated by time-dependent (Markovian) stochastic processes. In order to capture the average performance of FedSUM under such scenarios, we can take full expectation with respect to $\{ \mathcal{S}\_t , t=0,..., T-1\}$ on both sides of the inequality in Theorem 4.1, leading to the following result.
>
>     **Corollary 4.1** Under the same setting as in Theorem 4.1, suppose the participation sequence $\{ \mathcal{S}\_t , t=0,..., T-1\}$ involves randomness.
>     Then all three algorithms achieve the following convergence rate:
>     $$
>         \frac{1}{T}\sum_{t=0}^{T-1} \mathbb{E}\bigl[\|\nabla f(x^{(t)})\|^2\bigr]
>         \le \frac{30\sqrt{(1+\mathbb{E}[\tau_{\text{avg}}])L\sigma^2\Delta_f}}{\sqrt{NKT}}
>         + \frac{20\mathbb{E}[\tau_{\max}](L\Delta_f + F_0)}{T}.
>     $$
>
>     Corollary 4.2 directly follows from Theorem 4.1 noting that $\mathbb{E}[\sqrt{(1+\tau_{\text{avg}})}]\le\sqrt{(1+\mathbb{E}[\tau_{\text{avg}}])}$. We state Corollary explicitly in the revision.
>
>     Note that $\mathbb{E}[\tau_{\text{avg}}]$ and $\mathbb{E}[\tau_{\max}]$ can be estimated by Monte Carlo simulations under specific Markovian stochastic processes.
>
> - **Bias issue.** The proposed methods do not suffer from the bias issue due to the Stochastic Uplink-Merge technique.
> In FedSUM, $y^{(t)}$ represents the sum of aggregated gradients from the most recent participation of each previously active client and serves as the server's update direction. This design allows the FedSUM family to address data heterogeneity as well as the bias issue by incorporating information from each client’s latest participation round. Similar idea has been explored in earlier works; see, e.g., [1, 2, 3]. To better convey the idea, we provide visualizations in Figure 2 and Figure 3 in Appendix A of the revised manuscript to demonstrate how the SUM technique mitigates the bias issue.
>
>     [1] Gu, X., Huang, K., Zhang, J., & Huang, L. (2021). Fast federated learning in the presence of arbitrary device unavailability. Advances in Neural Information Processing Systems, 34, 12052-12064.
>
>     [2] Yan, Y., Niu, C., Ding, Y., Zheng, Z., Tang, S., Li, Q., ... & Chen, G. (2024). Federated optimization under intermittent client availability. INFORMS Journal on Computing, 36(1), 185-202.
>
>     [3] Ying, B., Li, Z., & Yang, H. (2025). Exact and Linear Convergence for Federated Learning under Arbitrary Client Participation is Attainable. arXiv preprint arXiv:2503.20117.
>
>
> In the latest version of the paper, we have adjusted the coefficients in the algorithm to ensure a fairer comparison among the algorithms. Specifically, we re-weight the local update direction by dividing by $N$, the number of clients, as shown in Line 9, Line 11, and Line 12 of both FedSUM and FedSUM-CR and reproduce all the experiments. This adjustment ensures that each local update has a similar scaling for the update direction, comparable to that of other algorithms. This change is equivalent to the original version, where we can derive the original versions by picking $\eta_l \leftarrow N\eta_l$ and $\eta_g \leftarrow \eta_g/N$ in the updated version.

---

### Author Response · Authors · 2025-11-30

Dear Area Chair,

Thank you and all the reviewers for your time and valuable feedback on our submission, "FedSUM Family: Efficient Federated Learning Methods under Arbitrary Client Participation" (Submission 16500). We appreciate the reviewers' recognition of this paper's contribution, namely the development of a general framework for modeling arbitrary client participation in federated learning and the introduction of the FedSUM family of algorithms with unified convergence guarantees that recover well-established rates under specific participation patterns. In the rebuttal, we have carefully addressed all comments and revised the manuscript with substantially enhanced numerical experiments and improved clarity of presentation.

While we’ve responded to specific points in the rebuttal, we believe it’s helpful to summarize the key aspects. We have categorized the reviewers' comments into three areas: Delay Metrics and Client Participation Patterns, Experimental Analysis and Performance Comparisons, and Algorithm Design.

1. **Delay Metrics and Client Participation Patterns**

    - **Time-dependent Participation and Delays**

        Reviewer 7geH inquired about capturing time-dependent client participation, including Markovian patterns, while reviewers ew4X and Z1o5 inquired about delay estimation in practical or specific participation patterns.

        We clarify that the participation sequence $\mathcal{S}\_t, t=0,...,T-1$ can include time-dependent (Markovian) stochastic processes. Our framework allows expectation-based analysis for these scenarios, as detailed in **Theorem 4.1** and **Corollary 4.1**, which provides a convergence rate based on **the expectations of $\tau_{\max}$ and $\tau_{\text{avg}}$** when participation involves randomness. Furthermore, $\mathbb{E}[\tau_{\text{avg}}]$ and $\mathbb{E}[\tau_{\max}]$ can be estimated using Monte Carlo simulations for specific Markovian processes.

        We also provide delay estimations for the four client participation patterns discussed in Section 2.1 (Cases 1 to 4), with detailed derivations in Appendix F.

    - **Sparse Participation Regimes**

        FedSUM performs well in sparse participation regimes, with performance comparable to state-of-the-art methods. For instance, when $S$ clients are uniformly randomly sampled in each round, the convergence rate of FedSUM becomes $\tilde{O}(\frac{1}{\sqrt{SKT}})$, which is comparable to FedAvg and SCAFFOLD. Experiments demonstrate strong performance even in extremely sparse settings such as when $p = 0.05$ (see Figure 10).

2. **Experimental Analysis and Performance Comparisons**

    - **Enhanced Experiments**

        In response to Reviewer 7geH and 7duF’s requests, we have expanded the experimental evaluation of FedSUM against competing algorithms. Specifically, we additionally trained a ResNet-18 model on the CIFAR-10 and CIFAR-100 datasets with 50 clients, as shown in **Figures 12a** and **12b** in Appendix H.5, respectively. Moreover, we fine-tuned a BERT model on the SST-2 dataset, as shown in **Figure 13** of Appendix H.6. These experiments show that FedSUM converges faster or performs competitively to other algorithms.

    - **Communication Savings in FedSUM-CR**

        Reviewer 7duF raised concerns about the communication cost and memory overhead in FedSUM-CR. We clarified that FedSUM-CR saves 1/3 of the communication workload per iteration but incurs additional memory overhead compared to FedSUM (**Figure 1b**).

    - **Impact of Varying Delays**

        In response to Reviewer 7duF’s suggestion, we explored the effect of varying delays $\tau_{\text{avg}}$ and $\tau_{\max}$, showing that larger values of $p$ (smaller delays) lead to better performance (see **Figure 10**).


3. **Algorithm Design**

    - **Bias Issue**

        Reviewer 7geH raised the concern about the bias issue. We clarified that FedSUM does not suffer from this issue, as the Stochastic Uplink-Merge technique mitigates bias. Visualizations in **Figures 2** and **3** and empirical results in **Figure 14** (Appendix H.8) confirm that FedSUM remains unaffected by bias.

    - **Connection between FedSUM and FOCUS**

        Reviewer Z1o5 questioned the connection between FedSUM and FOCUS. We clarify that, FedSUM utilizes the SUM technique to mitigate data heterogeneity and bias similarly to FOCUS. However, the two algorithms differ in their update mechanisms, the participation pattern, objective functions, and stochastic gradients (discussed in Section 3.1).

We have updated the manuscript to address these findings and enhance clarity. We believe these revisions strengthen the paper’s contribution and address the gaps noted by reviewers. Thank you for considering this additional context during your final assessment.

Thank you for your time and consideration.

Best regards,
The Authors of Submission 16500

---

### Meta-Review · Area_Chair_jfvq · 2026-01-06

**Summary:**

This paper addresses federated learning with arbitrary and irregular client participation. It proposes the FedSUM family of algorithms, which use a stochastic uplink-merge framework to maintain global progress under participation variability. The paper provides unified convergence guarantees and introduces maximum and average delay metrics to characterize client participation. Experiments show that FedSUM variants match or outperform existing methods across diverse participation patterns.

**Reviewer Concerns:**

Below, I summarize the reviewers' concerns and how the authors addressed them.

**Reviewer 7geH.**

1. The experiments are limited to small, standard vision datasets and do not reflect realistic or large-scale FL settings. The authors added new experiments on larger and more diverse models and tasks, including ResNet-18 on CIFAR-10/100 with 50 clients and BERT fine-tuning on SST-2, demonstrating competitive or faster convergence.

2. The framework does not capture time-dependent or correlated (e.g., Markovian) client participation. The authors clarified that their framework allows arbitrary participation sequences, including Markovian processes, and extended their theoretical analysis by taking expectations over stochastic participation, resulting in a new convergence corollary.

3. The paper does not analyze bias introduced by correlated client participation as discussed in prior work. The authors argued that the Stochastic Uplink-Merge design inherently mitigates bias by aggregating the latest gradients from previously active clients, supported by discussion, citations to related work, and new visualizations.

**Reviewer 7duF.**

1. The experiments do not directly demonstrate the impact of the proposed delay metrics on convergence or performance. The authors added experiments with probability-independent participation and varying participation rates, showing that smaller average and maximum delays lead to improved performance consistent with theory.

2. Communication and memory overheads are not quantitatively evaluated, despite being a key motivation for FedSUM-CR. The authors provided explicit comparisons demonstrating that FedSUM-CR achieves similar convergence while reducing communication by roughly one-third per iteration, while acknowledging the additional model-sized memory overhead.

3. The experimental scope is narrow, focusing only on vision tasks with a single model architecture. The authors expanded experiments to include deeper vision models (ResNet-18 on CIFAR-10/100) and a non-vision task (BERT fine-tuning on SST-2), showing competitive or faster convergence.

4. Several figures are overcrowded, which hinders result interpretability. The authors redesigned plots with more distinguishable lines, separated comparisons across figures, and added focused views of the final training stages.

**Reviewer ew4X.**

1. Limited experimental evaluation on small- and medium-scale datasets, raising concerns about scalability. The authors added new experiments on larger and more realistic models and datasets, including ResNet-18 on CIFAR-10/100 and BERT fine-tuning on SST-2, demonstrating competitive or improved performance.

2. Mismatch between the participation patterns used in experiments and the theoretical participation cases. The authors clarified the participation cases and adjusted algorithm coefficients to ensure fair comparisons across methods, re-running all experiments with consistent scaling.

3. Lack of clarity on the practical meaning and typical values of the delay metrics ($\tau_{\max}$ and $\tau_{avg}$). The authors provided explicit bounds and expectations for $\tau_{\max}$ under different participation patterns, with formal derivations included in the appendix.

4. Unclear performance in highly sparse participation regimes (e.g., cross-device FL), particularly relative to FedAvg. The authors analyzed sparse settings theoretically and empirically, showing that FedSUM behaves similarly to FedAvg/Scaffold in terms of convergence rate and performs well with appropriate learning-rate tuning.

5. Insufficient clarification of the relationship between the reshuffled cyclic participation setting and prior client reshuffling-based FL work. The authors clarified that the reshuffled cyclic participation case aligns with prior client reshuffling work, added the missing citation, and explained the conceptual connection in the revised paper.

**Reviewer Z1o5.**

1. Claims on reduced communication cost lack quantitative or theoretical support. The authors didn't respond to this concern, as far as I see.

2. Related work (FOCUS / Ying et al., 2025) is insufficiently discussed and compared. The authors clarified the distinction between FedSUM and FOCUS, emphasizing non-equivalence due to server-side correction and different update mechanisms.

3. Some theoretical assumptions (e.g., bounded gradient heterogeneity) need better justification. The authors clarified that FedSUM does not rely on a bounded gradient heterogeneity assumption, similar to SCAFFOLD and FOCUS.

4. Key definitions (e.g., $\tau_{\max}$ under uniform sampling) need clarification. The authors provided a formal definition and bound for $\tau_{\max}$ under uniform sampling, with derivations in the appendix.

**Reviewer Scores:**

The rebuttal helps clarify several aspects of the paper and addresses a number of the reviewers’ questions. However, some concerns remain regarding the strength of the empirical validation and how well the evaluation reflects practical settings.

As a result, it is difficult to assess whether the reviewers would significantly increase their scores based on the rebuttal alone. Overall, while the work is technically sound, it does not appear sufficiently strong across all dimensions to support acceptance at this time.

---

### Decision · Program_Chairs · 2026-01-26

Reject